# Position: A Dynamical Systems Perspective is Needed to Advance Time Series Modeling

**Daniel Durstewitz** [* 1 2 3]  **Christoph Jürgen Hemmer** [* 3 1 2]  **Florian Hess** [1 2]  **Charlotte Ricarda Doll** [1 2]  **Lukas Eisenmann** [1 2]

## Abstract

Time series (TS) modeling has come a long way from early statistical, mainly linear, approaches to the current trend in TS foundation models. With a lot of hype and industrial demand in this field, it is not always clear how much progress there really is. To advance TS forecasting and analysis to the next level, here we argue that the field needs a *dynamical systems (DS)* perspective. TS of observations from natural or engineered systems almost always originate from some underlying DS, and arguably access to its governing equations would yield theoretically optimal forecasts. This is the promise of *DS reconstruction (DSR)*, a class of ML/AI approaches that aim to infer *surrogate models* of the underlying DS from data. But models based on DS principles offer other profound advantages: Beyond short-term forecasts, they enable to predict the *long-term statistics* of an observed system, which in many practical scenarios may be the more relevant quantities. DS theory furthermore provides domain-independent *theoretical insight into mechanisms* underlying TS generation, and thereby will inform us, e.g., about upper bounds on performance of *any* TS model, generalization into unseen regimes as in tipping points, or potential control strategies. After reviewing some of the central concepts, methods, measures, and models in DS theory and DSR, we will discuss how insights from this field can advance TS modeling in crucial ways, enabling better forecasting with much lower computational and memory footprints. We conclude with a num-
ber of specific suggestions for translating insights from DSR into TS modeling.

[1]Dept. of Theoretical Neuroscience, Central Institute of Mental Health, Mannheim, Germany [2]Faculty of Physics & Astronomy, Heidelberg Univ., Heidelberg, Germany [3]Interdisciplinary Center for Scientific Computing, Heidelberg, Germany. Correspondence to: Daniel Durstewitz, Christoph Hemmer <{daniel.durstewitz,christoph.hemmer}@zi-mannheim.de>.

*Proceedings of the 43$^{rd}$ International Conference on Machine Learning*, Seoul, South Korea. PMLR 306, 2026. Copyright 2026 by the author(s).

## 1. Introduction: The DS Perspective

Any system that evolves in time, where this temporal evolution can be described by a set of rules in either discrete time (as a recursive map) or in continuous time (as a set of differential equations), is a dynamical system (DS).[1] DS theory is the mathematical theory dealing with the behavior of such systems. Almost any time series (TS) from natural or human-made systems originates from some underlying DS. Whether we consider chemical and molecular systems (Bhalla & Iyengar, 1999), weather (Kalnay, 2003), climate (Tziperman et al., 1997), the brain (Izhikevich, 2007), physiological processes (Govindan et al., 1998), ecosystems and animal populations (Turchin & Taylor, 1992; Mumby et al., 2007), disease and epidemics (Dehning et al., 2020), behavior (Staddon, 2001), psychiatric conditions (Durstewitz et al., 2020), societal processes like opinion dynamics (Hołyst et al., 2001), or the economy (Mandelbrot & Hudson, 2007), all of them can be understood and formulated as DS at some level. Also the machine learning (ML) tools used to analyze and forecast TS, from simple statistical auto-regressive moving-average (ARMA) models to recurrent neural networks (RNNs), as well as the gradient-based techniques we use to train them, are DS in a strict formal sense. Hence, it appears intuitively clear that a mathematical theory dealing with the temporal behavior of such systems should be able to inform TS analysis (TSA) and forecasting (TSF), and help to improve training and modeling.

But how specifically? This is the question we are going to explore in this position paper, calling for a paradigm shift:

> **Position: A DS perspective is necessary for making further progress in TSF/A and solving some of its most challenging generalization problems.**

We will first review central concepts, measures, and models in DS theory and *DS reconstruction (DSR)*. In DSR the

---

[1]See Tab. 3 for a list of acronyms.

goal is to learn a *generative surrogate model* of the DS that produced the observed TS. Since DSR models are trained to approximate the underlying governing equations, they can also be used for TSF. We will then explore important lessons for TSF gained from the DS perspective. We will also examine the *alternative view* that DS theory is of little practical use for most real-world TSF tasks, before we conclude with a couple of specific action items and recommendations.

## 2. Overview of key concepts in DS theory

Here we provide a short informal overview of key ideas in DS theory important for understanding our lines of argument; see Appx. A for a more formal introduction. DS theory is the mathematical theory which examines the temporal behavior of systems that are described by sets of (coupled) differential equations, $dx/dt \equiv \dot{x} = f(x)$ with *vector field* $f(x)$, or recursive maps, $x_t = F(x_{t-1})$, $x \in E \subseteq \mathbb{R}^M$ (Guckenheimer & Holmes, 1983; Alligood et al., 1996; Perko, 2001; Strogatz, 2024). When a DS *explicitly* depends on time, i.e. $\dot{x} = f(x,t)$ or $x_t = F(x_{t-1}, s_t)$, it is called *non-autonomous*, otherwise we call it *autonomous*. Hence, in an autonomous DS the vector field $f(x)$ itself is static (Fig. 1), while in a non-autonomous DS the vector field changes across time (Fig. 2) because of time-dependent parameters or external drivers. The set $E \subseteq \mathbb{R}^M$ in which a DS lives, i.e. the set of all possible states $x(t)$ or $x_t$ can assume, is called its *state space*. More generally, a DS is defined as a (semi-)group $(T, E, \Phi)$ with a set of times $T$ (typically $T = \mathbb{R}$ in continuous time, and $T = \mathbb{Z}$ in discrete time) and *flow* or evolution operator $\Phi(t, x_0) \equiv \Phi_t(x_0), t \in T, x_0 \in E$, which describes how the DS evolves in state space $E$ across time $t$ starting from some initial condition $x_0$. In continuous time, $\Phi$ may be thought of as providing the solution $\Phi_t(x_0) = x_0 + \int_0^t f(x(s))ds$ to the set of differential equations $\dot{x} = f(x)$ as obtained by integrating the vector field along the curve starting from $x_0$. In discrete time, $\Phi$ is simply given by the map $F$.

The idea of a state space, illustrated in Fig. 1, is central to DS theory. A *trajectory* (or orbit) $\Omega_{x_0} = \{x(t) \mid \exists t : x(t) = \Phi_t(x_0)\}$ is the *unique* curve in state space that leads through point $x_0$ following the vector field $f(x)$. Two trajectories in this space may never intersect, since if they would, the vector field would not be uniquely defined at the intersection point and the state space would not be complete, as we mathematically require. *Observed* TS $\{y_\tau\}, \tau = 1 \ldots T$, correspond to trajectories $x(t)$ in state space, assessed through some *measurement* or *observation function* $y_\tau = g(x(\tau))$ at discrete time points $\tau$. **A key point about state spaces is that they introduce *geometric* and *topological* concepts into the analysis of temporal behavior.** It is the topological and geometric properties of

the state space which determine the fate of trajectories, and hence the nature of the associated TS. If we know the state space of a DS, we can predict its behavior starting from any initial condition $x_0 \in E$ for all time. The *limiting* behavior of a TS is determined by the limiting behavior of the corresponding trajectory in state space, which will follow the vector field and may ultimately converge into specific topological sets called *attractors* (Appx. A.1). An attractor may just be a single point (an *equilibrium* or *fixed point*; Fig. 1), a closed orbit called a *limit cycle* (Fig. 1), or may have a more complex, *fractal* geometry, a *chaotic* attractor (Fig. 3). Hence, an attractor is a set in state space toward which trajectories starting from a set of initial conditions surrounding the attractor, its so-called *basin of attraction*, will ultimately converge (Fig. 1a). Trajectories *within* the attractor will–if unperturbed–never leave it, making an attracting set $\mathcal{A}$ *invariant* under the flow ($\Phi_t(\mathcal{A}) \subseteq \mathcal{A} \ \forall t$). A more formal definition is provided in Def. A.2.

An important insight from DS theory is that many of such attractors, each with its own basin of attraction, can *co-exist* in the same DS for the very same set of parameters, so-called *multistability*, an example of which is provided in Fig. 1. In such systems, noise may push the system's state from one basin of attraction into another, leading to an abrupt change in dynamical regime, also called *N-tipping* (Fig. 1b), one type of tipping point (Ashwin et al., 2012). Multistability is not a curiosity, but actually *the rule* in high-dimensional complex systems such as the brain (Izhikevich, 2007) or climate systems (Lohmann et al., 2024). Another type of tipping point, so-called *B-tipping*, can occur when a slowly changing control parameter of a DS drives it across a *bifurcation*, which denotes a *qualitative, topological* change in the system's state space, pushing it abruptly into a different dynamical regime (Fig. 2; Appx. A.3). By 'topological change' we mean that the vector fields of the DS before and after the bifurcation point are no longer *topologically equivalent*. Loosely, topological equivalence says there is a *bijective* map $h : A \to B$ mapping between trajectories in state spaces $A$ and $B$ that preserves the direction of flow; see Def. A.4 for a precise definition.

Fixed points (equilibria), limit cycles and chaotic sets may not be stable, i.e. might not be attractors towards which states converge from all directions for $t \to \infty$. They could also be *unstable* or *half-stable*, with states *diverging* along one or more directions which span their *unstable manifold* (Fig. 1). An important quantity for characterizing the degree of convergence (contraction) vs. divergence (expansion) in different directions as we move along trajectories is the *Lyapunov spectrum*, given for discrete-time DS by

$$\lambda_i := \lim_{T \to \infty} \frac{1}{T} \log \sigma_i \left( \prod_{t=0}^{T-1} J_{T-t} \right), \qquad (1)$$

where $\sigma_i$ is the $i$-th singular value of the product of Ja-

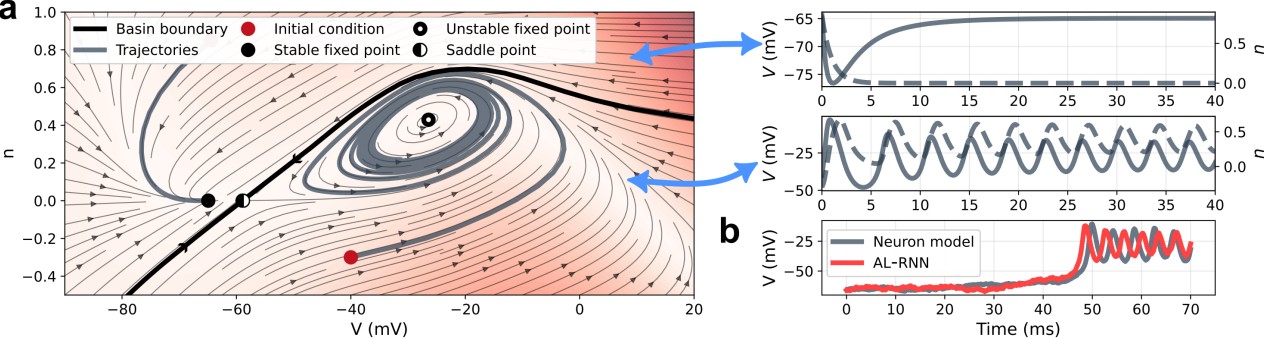

*Figure 1.* **a**) State space of a bistable neuron model with co-existing point (left basin) and limit cycle (right basin) attractor (see Appx. D.2 for details). The two basins are separated by the *stable manifold* of a *saddle node*. Trajectories follow the system's vector field, indicated by arrows with red shading indicating flow velocity (darker = faster). The corresponding TS (right) of model variables $V(t)$ (solid) and $n(t)$ (dashed) converge either to a point attractor (top) or a stable limit cycle (bottom) depending on the basin in which the trajectory was initialized (as indicated by the blue arrows). **b**) N-tipping in the true neuron model (gray) and in an AL-RNN (red) trained on trajectories from both basins. Noise was added in both models and eventually drives them across the basin boundary into cyclic activity (cf. Fig. 9).

cobians $\boldsymbol{J}_t := \frac{\partial F(\boldsymbol{x}_{t-1})}{\partial \boldsymbol{x}_{t-1}}$. This can be defined analogously for continuous-time DS via the flow map $\Phi$. For a point attractor we have all $\lambda_i < 0$, for a stable limit cycle $\lambda_{max} := \max(\lambda_i) = 0$, and for a *chaotic* attractor $\lambda_{max} > 0$. In fact, at least one positive Lyapunov exponent is *defining* for chaos (together with the condition that a chaotic set $\mathcal{A}$ must be bounded; Alligood et al. (1996)). Another characteristic feature of chaos is that TS are *irregular*, *aperiodic*: Unlike a limit cycle (nonlinear oscillation), trajectories never close up and the system never returns to the same precise value, even in the complete absence of noise. See Appx. A for further discussion of DS concepts.

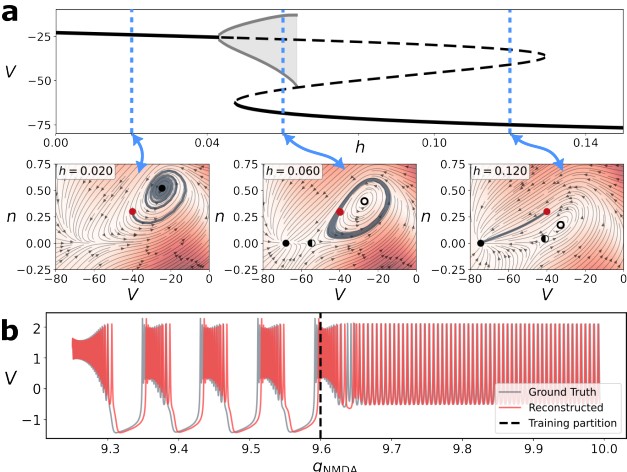

*Figure 2.* **a**) Bifurcation diagram of a spiking neuron model depending on a control parameter $h$ (see Appx. D.2; Durstewitz (2009)). Stable fixed points are indicated by solid black lines, unstable ones by dashed black lines. The gray-shaded area corresponds to a stable limit cycle. Graphs below give snapshots of the state space for different values of $h$. **b**) B-tipping from a 'bursting' into a 'spiking' regime in the full simulated neuron model (gray) is successfully predicted by an AL-RNN (red) trained only on TS data up to the black dashed line.

## 3. Dynamical Systems Reconstruction (DSR)

In DSR, we aim to learn a surrogate model from TS data that, after training, behaves equivalently to the observed system in a DS sense (Fig. 7). Formally, what we are aiming for is that the reconstructed DS is *topologically equivalent*, or even *topologically conjugate*, to the observed DS on the whole of state space according to Def. A.4 and Def. B.1. This is challenging (Göring et al., 2024), and custom-trained approaches often content with topological conjugacy on a single basin of attraction $\mathcal{B}$ (cf. Figs. 1 & 10). While topological properties are often considered most important from a theoretical perspective, we commonly would also like to preserve geometric and temporal properties of the underlying DS. In particular, *a proper DSR model should have the same behavior in the limit* $t \to \infty$, i.e. should converge to the same attractor objects with the same *geometric* and *temporal* structure. For that reason, the focus in DSR is often on training algorithms and loss functions that incentivize the correct long-term behavior (Appx. C; Mikhaeil et al. (2022); Hess et al. (2023); Platt et al. (2022; 2023); Jiang et al. (2023); Schiff et al. (2024)), and evaluation is done with measures that assess the *long-term ("climate") statistics* (Fig. 3).

Common measures to assess overlap in *attractor geometry* compare true and model-generated trajectory distributions in *state space*, e.g. based on the Kullback-Leibler divergence (denoted $D_{stsp}$ below) or Wasserstein distance (Brenner et al., 2022; Göring et al., 2024; Park et al., 2024; Fumagalli et al., 2025), see Appx. B for details. An attractor's geometric structure can also be quantified through its *box-counting* or *correlation dimension*, or the *Kaplan-Yorke dimension* computed from the Lyapunov spectrum as as an upper bound (see Appx. B.1; Alligood et al. (1996); Kantz & Schreiber (2004)). These specifically capture the *fractal geometry* of the object. To compute these measures for empirical data,

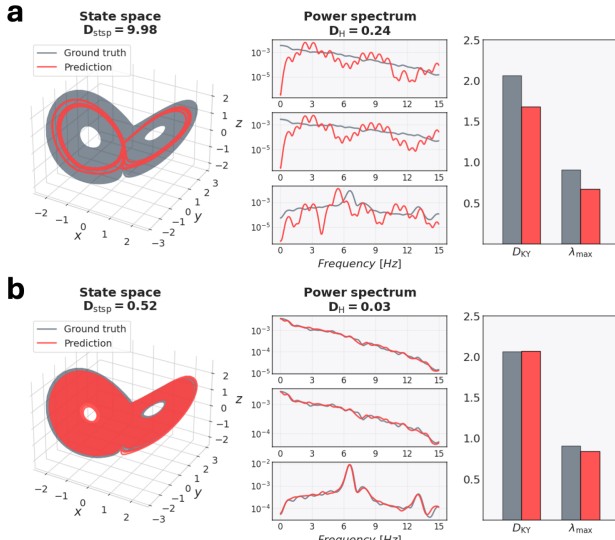

*Figure 3.* DSR measures: Comparison of geometric (dis)agreement ($D_{\text{stsp}}$), power spectral distance ($D_{\text{H}}$), Kaplan-Yorke fractal dimension ($D_{\text{KY}}$), and max. Lyapunov exponent ($\lambda_{\text{max}}$) on **a**) a poor and **b**) a good reconstruction of the chaotic Lorenz-63 system by an AL-RNN. See also Fig. 8.

since we usually have only access to lower-dimensional observations from a much higher-dimensional DS, *temporal delay embedding* techniques are commonly used to lift the empirical observations into a higher-dimensional space where trajectories and the vector field along them smoothly map $1 : 1$ onto those of the true underlying system according to the delay embedding theorems (Takens (1981); Sauer et al. (1991); see Appx. A.2). To assess agreement in the *long-term temporal structure* of true and generated trajectories, measures based on the system's auto-correlation function (Wood, 2010; Brenner et al., 2024b) or power spectrum (Mikhaeil et al., 2022; Brenner et al., 2022) could be used, for instance the Hellinger distance between appropriately smoothed power spectra (Mikhaeil et al., 2022; Hess et al., 2023). *It is important to note that most of these measures only make sense in the ergodic long-term limit $t \to \infty$. A transient trajectory is not informative about attractor geometry*, nor can a power spectrum or Lyapunov exponents be reasonably computed from just short, potentially transient (non-stationary) trajectory bits (see Appx. B for more details). Thus, commonly time series of length $T \geq 1000$ (depending on scale), often $T \geq 10^4$, are simulated from the trained model, and initial transients are cut off in estimation.

DSR models have been formulated based on various types of RNNs (Vlachas et al., 2018; Brenner et al., 2022; 2024a; Hess et al., 2023; Brändle et al., 2026), reservoir computers (RCs; Pathak et al. (2017); Verzelli et al. (2021); Platt et al. (2022; 2023)), library-based methods like SINDy (Brunton et al., 2016; Champion et al., 2019), Koopman operator theory (Otto & Rowley, 2019; Brunton et al., 2022; Naiman

& Azencot, 2021; Azencot et al., 2020; Wang et al., 2022; Lusch et al., 2018), or continuous-time models such as Neural ODEs (Chen et al., 2018; Karlsson & Svanström, 2019; Alvarez et al., 2020; Ko et al., 2023); see Appx. E. There are also DSR models which explicitly account for *process/dynamical noise*, based on stochastic differential equations (Neural SDEs; Tzen & Raginsky (2019); Haussmann et al. (2021); O'Leary et al. (2022); Issa et al. (2023)) in continuous time, or some form of nonlinear Kalman filters or variational inference in discrete time (Koppe et al., 2019; Kramer et al., 2022; Brenner et al., 2024b; Pals et al., 2024). While these may be preferable if uncertainty estimates in latent states or model parameters are required, e.g. in clinical settings or weather forecasts, they are often more brittle and computationally expensive to train (Issa et al., 2023; Pals et al., 2024), and scale less well. Indeed, they commonly lag in DSR performance behind their deterministic counterparts, *even when the true DS is dynamically stochastic*, with some of the underlying reasons identified in Herz et al. (2026). Lastly, beyond Neural ODEs, DSR models have also been formulated specifically for spatiotemporal (Raissi et al., 2019; Li et al., 2021; Chen et al., 2021) and irregularly sampled (Brändle et al., 2026) TS data.

Transformer-based architectures, in contrast, are uncommon in this field, *mainly because they lack a natural representation of time as required for approximating a flow map or its integral* (Zhai et al., 2026). In fact, in sharp contrast to the TS modeling field, we are not aware of any transformer-based model which is able to reconstruct DS in the sense, and according to the measures, defined above (see Tables 1 & 2 and Figs. 14 & 15).

As already indicated, far more important than the architecture itself is the training procedure: To encourage the correct long-term behavior and combat exploding gradients, which are – mathematically – an inevitable consequence of training on chaotic DS (Mikhaeil et al., 2022), control-theoretic training methods like *sparse teacher forcing (STF)* (Mikhaeil et al., 2022; Brenner et al., 2022) and *generalized teacher forcing (GTF)* (Hess et al., 2023; Sağtekin et al., 2025; Doya, 1992) have been devised (not to be confused with conventional TF), see Appx. C for details. These enable long rollouts, allowing model-generated trajectories to 'explore the future' whilst training, either by replacing model-generated by data-inferred states $\hat{\boldsymbol{z}}_{t+k\tau} = g^{-1}(\boldsymbol{x}_{t+k\tau})$ at time lags $\tau$ optimally chosen according to the system's maximal Lyapunov exponent (STF), or by striking an optimal balance between model-generated and data-inferred states at each time step, $\hat{\boldsymbol{z}}_t = (1-\alpha)\boldsymbol{z}_t + \alpha g^{-1}(\boldsymbol{x}_t)$ (GTF; see Hess et al. (2026) for an extension to particularly long roll-outs). Other common procedures explicitly include longer-term forecasts (Platt et al., 2022; Vlachas & Koumoutsakos, 2024; Lusch et al., 2018) and/or invariant statistics like the maximal Lyapunov exponent or an estimate of fractal dimensionality

(Platt et al., 2023), or invariant measures (Jiang et al., 2023; Schiff et al., 2024), in the loss function.

A final important aspect about DSR models that may also impact the TSA field concerns their *interpretability and mathematical tractability*. DSR models are not mere prediction tools, they are supposed to provide insight into the *dynamical mechanisms* underlying the data, as relevant in scientific or medical applications (Durstewitz et al., 2023; Fechtelpeter et al., 2025). We would like to be able to analyze the topological and geometric properties of their state spaces to learn how the underlying system works, and to understand its behavior *even outside of the immediate data regime*, as illustrated in Figs. 10 & 11. Having access to an approximate flow operator or vector field also enables to derive *optimal control strategies*, e.g. for designing optimal interventions in medical settings (Fechtelpeter et al., 2025). For that reason, many important DSR models are *piecewise-linear* (De Feo & Storace (2007); Storace & De Feo (2004); Hess et al. (2023); Brenner et al. (2024a); Linderman et al. (2016); see Appx. E for a more detailed overview of model classes): For *linear* DS we have a complete analytical understanding, but they cannot produce many important DS phenomena such as stable oscillations, multistability, or chaos (Perko, 2001). Piecewise-linear models are the next-best alternative which can reconstruct arbitrary DS (Brenner et al., 2024a; Linderman et al., 2016), yet still allow for semi-analytical computation of many of their topological and geometric properties (Eisenmann et al., 2023; 2026). For that reason, piecewise-linear models have been popular in engineering (Bemporad et al., 2000; Carmona et al., 2002; Juloski et al., 2005) and DS theory (Alligood et al., 1996; Avrutin et al., 2019; Coombes et al., 2024; Simpson, 2023) for many decades.

## 4. Lessons for TSF from the DSR angle

DS theory and DSR, unlike pure TS modeling, provide us with an *understanding of the processes* that led to the TS observed, and allow to infer properties of the system beyond what a pure TS model could provide. This understanding can help predicting previously unobserved phenomena or devising control strategies. It is also a type of understanding that does not require domain knowledge about the system at hand: DS principles and phenomena, such a multistability, chaos, or bifurcations, are *universal*, they can be observed the same way in arbitrary systems from interacting particles to social interactions (Strogatz, 2024). Let us now examine in more detail the implications for TS models & TSA.

**Extrapolating beyond the data regime** TS models focus on prediction, while DSR models explicitly aim to approximate the underlying governing equations or flow operator. If these dynamical laws were precisely known, then this would enable to forecast the system's future temporal evolution naturally better than would any other way of modeling the system's temporal structure less directly (Gneiting & Raftery, 2007; Kaszás & Haller, 2020). In fact, the system's behavior could be predicted *anywhere* across its state and parameter space (Göring et al., 2024). In practice, of course, there is no guarantee a good approximation will indeed be achieved (as is true as well for any human-created scientific theory). However, DSR-specific training techniques and assessment criteria (cf. Sect. 3) make this outcome more likely. In fact, current SOTA DSR models usually can generalize to new (unobserved) initial conditions within a given basin of attraction and reconstruct the full attractor geometry from just a short trajectory snippet, as shown in Fig. 10. They may also infer additional geometric properties of the state spaces, like fixed points (Fig. 10) or their un-/stable manifolds (Eisenmann et al., 2026), even though not explicitly represented in the training data, and reproduce properties of the vector field surrounding the training regime (Fig. 11). These are capabilities that standard TS models fundamentally lack, and that require some sort of inference beyond the immediate data regime. They enable to analyze the observed DS in depth, gain some causal and mechanistic understanding, and explore new empirical scenarios, perturbations, and suitable control strategies by model simulation (Fechtelpeter et al., 2025).

**Tipping points and abrupt transitions in dynamical regime** A particularly challenging form of prediction beyond the training domain is a change in dynamical regime. For instance, a good biophysical model of the nervous system would be able to predict transitions into epileptic seizures under some parameter changes (Jirsa et al., 2014), even if only knowledge about healthy tissue was used in constructing the model. It has been argued in TSA for a while that such abrupt changes in the behavior of a TS are unpredictable (i.e., usually not inferrable just from historical records) unless the *underlying process* is known or modeled (Pesaran et al., 2006). There are indeed many important cases where a DS perspective provides hope. These are situations where the system under study undergoes a *tipping point* and enters a new dynamical regime. For multistable DS, for instance, we might be able to predict the likelihood of transitions among different regimes at different times (N-tipping), as illustrated in Fig. 1b. Granted, of course, access to the system's state space, as DSR models promise to provide. Many DS undergo tipping points as a consequence of a slowly changing control parameter (B-tipping), like climate systems due to rising levels of greenhouse gases, or sepsis (blood poisoning) in a patient due to buildup of bacterial load. A DSR model trained on a non-stationary TS or stationary snapshots sampled across different dynamical regimes (cf. Fig. 12) may be able to pick up such driving forces and predict a bifurcation in the underlying DS (Kong

et al., 2023; Patel & Ott, 2023; Köglmayr & Räth, 2024; Huh et al., 2025; Van Tegelen et al., 2025), an example of which is shown in Fig. 2b. Forecasting tipping points and post-tipping dynamics is still a hard problem. It is a type of *out-of-domain (OOD) generalization* which is much harder than that imposed by a 'mere' distribution shift, because it represents a *topological shift* in dynamics (Göring et al., 2024) (cf. Def. A.4). In fact, in its most general form, the topological OOD problem is *intractable* (Göring et al., 2024). However, exploiting topological constraints and functional dependencies within a DS (Huh et al., 2025), pre-training on related systems (Brenner et al., 2025; Hemmer & Durstewitz, 2025), physics-informed priors (Raissi et al., 2019; Lim et al., 2020) or other inductive biases (Fig. 13), or methods for inferring relevant control parameters jointly with the dynamics (Huh et al., 2025), start to show promise. This is another active research area in DSR from which TS modeling may hugely benefit.

**Chaos and the limits of predictability** Chaos is fundamentally relevant for TSF since most complex systems encountered in nature and society are almost inevitably chaotic (Sivakumar, 2004; Durstewitz & Gabriel, 2007; Govindan et al., 1998; Turchin & Taylor, 1992; Field et al., 1972), a natural consequence if many highly nonlinear and diverse elements are combined in a network (van Vreeswijk & Sompolinsky, 1996; Ispolatov et al., 2015). By virtue of its positive Lyapunov exponent, chaos imposes principle limits on the forecast horizon: It implies that minuscule differences in initial conditions grow exponentially fast, meaning that – depending on the exact size of $\lambda_{\max}$ – two TS from the *very same* chaotic DS will quickly and inevitably be completely misaligned after only a short time (Fig. 4). This also has implications for *evaluation*: Standard statistical quantities like MSE or MAPE, commonly used to evaluate TS model performance, become meaningless after some finite time (Wood, 2010). In the presence of noise (Fig. 4), this becomes a *principle* limitation, nothing we can really ameliorate that much by improving knowledge about the initial condition or the model design. Claims, therefore, about models being able to forecast noisy chaotic systems for more than one Lyapunov time (Liu et al., 2025) should be treated with great care! We may instead focus on statistical properties as determined by the DS' ergodic distribution.

**Short- and long-term forecasts** While DSR models are intended for understanding dynamical mechanisms behind TS generation, we may nevertheless ask whether the principles by which they are designed and trained also improve TSF. This is examined in Table 1 for a set of SOTA custom-trained DSR and TS models, and in Table 2 for DSR and TS foundation models (FMs); see also Fig. 5. For the custom-trained TS models, we chose AutoARIMA as a simple statistical model, PatchTST as a transformer-based (Nie

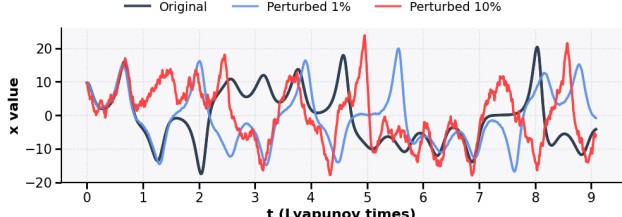

*Figure 4.* The limits of predictability: Three TS from the same Lorenz-63 system (same parameters) started at the same initial condition quickly diverge even with just 1% of noise, while with 10% noise prediction beyond 1 Lyapunov time becomes hopeless.

et al., 2023), and NBEATS as MLP-based model (Oreshkin et al., 2020). For DSR, we used the shPLRNN trained by GTF (Hess et al., 2023), AL-RNN trained by STF (Brenner et al., 2024a) and an RC for DSR (Platt et al., 2023). We compare both short-term prediction using MASE, and prediction of long-term statistical properties using $D_{\text{stsp}}$ and $D_{\text{H}}$ (see Appx. B). We trained all models on a variety of real-world data, including energy, weather, and traffic data, and physiological signals such as human fMRI and EEG.

As expected, DSR models have a clear edge over pure TS models in terms of correct reproduction of long-term properties (Fig. 5a). This is also vividly illustrated in Figs. 5c, 14 & 15, which show that in the long-term typical TS models – in contrast to DSR models – either diverge or converge to a fixed point (flat line), retaining none of the true long-term dynamics. While expected, since this is what DSR models are optimized for, it is in itself an important observation: *There are many empirical scenarios where the long-term properties are the more important prediction target.* For instance, in a patient with long-Covid (Thaweethai et al., 2025) or mental health issues (Fechtelpeter et al., 2025) we are less interested in the daily fluctuations in well-being but more in the long-term trend, e.g. in order to evaluate whether a specific pharmaceutical or behavioral intervention is effective. Similarly, in climate research we are primarily concerned with long-term predictions (Kirtman et al., 2013; Patel & Ott, 2023). More surprisingly, however, Fig. 5b reveals that SOTA DSR models can also outperform SOTA TS models in *short-term prediction* on at least some of the datasets (see also Fig. 28b). Thus, there seems to be something inherent in the training or design of DSR models that could also improve short-term prediction, although not their primary target.

These observations carry over to comparisons between DSR and TS FMs, as shown in Fig. 5 & Table 2, where *zero-shot* (no fine-tuning) short- and long-term forecasts are compared. Here we included Chronos (Ansari et al., 2024), Chronos-2 (Ansari et al., 2025), Panda (Lai et al., 2025), TiRex (Auer et al., 2025) as TS FMs, and DynaMix as the, to our knowledge, so far only true DSR FM trained by DSR

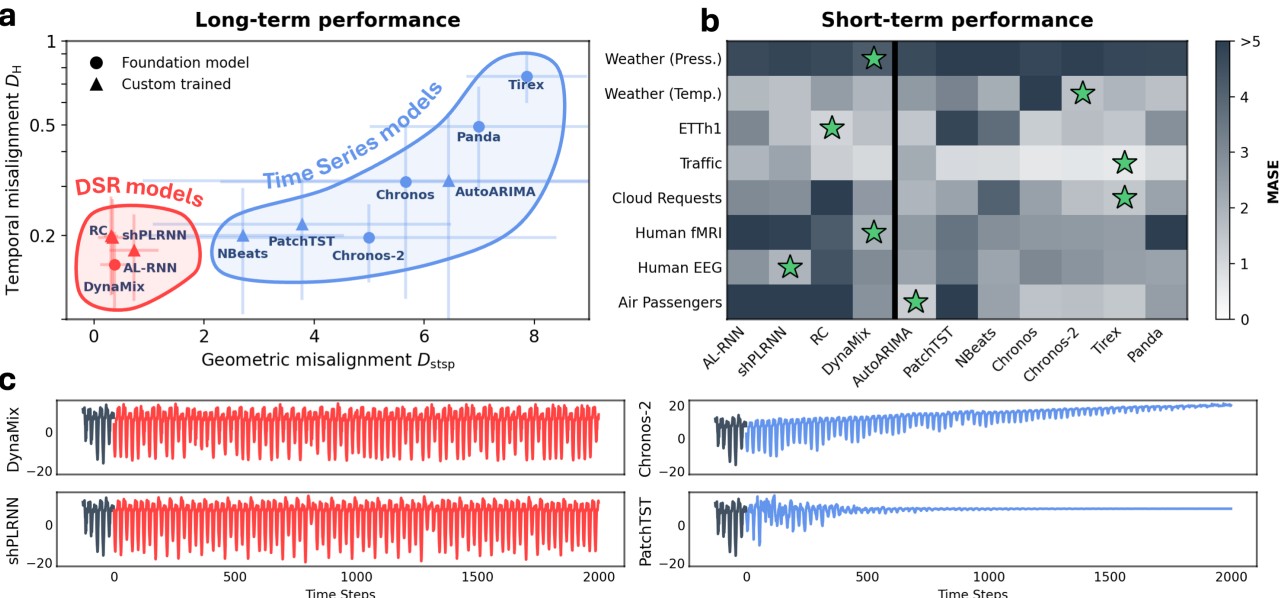

*Figure 5.* **a**): Long-term geometric ($D_{\text{stsp}}$) and temporal ($D_{\text{H}}$) forecast accuracy (cf. Appx. B) as median $\pm$ MAD, comparing DSR and TS models, evaluated on $10,000$-step roll-outs. Lower = better. Results for both custom-trained (triangles) and foundation models (dots) are shown. **b**): Short-term forecasting accuracy (MASE); lighter colors = better, green stars = best MASE. **c**): Example long-term forecasts of ETTh1 data for DSR and TS models, exposing the failure of TS models to capture long-term behavior. See also Figs. 14 & 15.

principles (see Appx. E for further details on all models).[2] As evidenced in Table 2 and further illustrated in Fig. 5 (and as already reported in Hemmer & Durstewitz (2025)), DynaMix is the *only* FM which can correctly reproduce attractor geometries and the system's long-term behavior. All other models converge to either simple fixed points or limit cycles in the long-term limit, not capturing any of the hallmark features of a chaotic attractor. For instance, *context parroting* is a phenomenon commonly observed in TS FMs like Chronos (Zhang & Gilpin, 2025). Context parroting leads to (nearly) periodic activity (Hemmer & Durstewitz (2025); see Fig. 16), and thus to not just geometrically but *topologically incorrect* long-term behavior on chaotic systems, failing to reconstruct the underlying DS. Furthermore, as above for the custom-trained models, we observe that DynaMix often even outperforms the TS FMs on short-term forecasts.

**The surprising simplicity of DSR models** Another remarkable observation is that DSR models achieve performance on par in short-term prediction with those of even the most sophisticated TS models despite their much simpler architectural design and lower parameter and training costs, see Tables 1 & 2. DynaMix, for instance, has only $0.01\%$ the number of trainable parameters that Chronos has, hence also $\approx 100\times$ faster inference times, and a minimal

training corpus of only $34$ different DS. The AL-RNN, as another example, one of the SOTA DSR models included in Table 1, features a simple piecewise-linear structure (Appx. E), essentially just a 1-layer RNN with a small number of ReLU nonlinearities. This reinforces a point we were making earlier: *The actual training algorithms used may be far more important than the network architecture!* TS models, on the other hand, may often tend to be overly complex. For FMs, besides the training algorithm, there may be another key feature that leads to the superior performance of DSR models, and that is their *training corpus*. While Chronos and many other TS FMs are largely trained on artificially assembled TS, constructed from concatenations of more basic processes, DynaMix is trained on TS from *simulated chaotic DS*. On the one hand side this may lead to a training corpus that is much more natural in a sense, much closer to real-world processes, which, as noted above, are often chaotic. On the other hand, many chaotic attractors are known to harbor an *infinity of unstable periodic orbits of any period*, with smeared out, long-tailed power spectra (Guckenheimer & Holmes, 1983; Alligood et al., 1996). Despite the much smaller training corpus, this may thus endow models like DynaMix with a much richer repertoire of experienced time scales and temporal patterns. A more compact, DS-inspired architecture and training corpus may also lead to a more natural integration of the different datasets. For Transformer-based FMs, in contrast, it has been reported that different training datasets tend to engage different task-relevant subnetworks within the model (Zhao

---

[2]While Panda, similar to DynaMix, was trained purely on TS from simulated DS, we still group it with the TS FMs here since it was optimized for short-term prediction, not DSR.

et al., 2025). That is, unlike DynaMix, these models do not learn an integrated representation, but more or less a collection of separate subnetworks, different for each type of dataset.

# 5. Alternative Views

Most of DS theory deals with low-dimensional ($\leq 3d$), deterministic, autonomous DS, assuming access to all the system's dynamical variables (Perko, 2001; Alligood et al., 1996; Strogatz, 2024). The most common benchmark scenarios for testing DSR performance are also of this type. In contrast, most TS of empirical relevance come from presumably much higher-dimensional underlying DS, which are only partially observed, and exhibit (highly) stochastic, non-autonomous and non-stationary dynamics (see Appx. A.4 for formal definition). Sometimes knowledge about external events or inputs may be more relevant in forecasting a TS than the history of the TS itself (Williams et al., 2025). Hence, one may argue that most of DS theory is of little relevance in practical, real-world settings, and that consideration of statistical properties, external regressors/ covariates, and context knowledge about the system in question (Williams et al., 2025) is far more important. For instance, integrating TSF into LLMs which enhance the forecasting process through their background and world knowledge may be a more rewarding future direction (Cho et al., 2025).

First of all, although less strongly developed than for autonomous, deterministic DS, there is some theory also for stochastic and non-autonomous DS (Kloeden & Yang, 2020). In the simplest case, one may see (and analyze) external inputs and noise as perturbations that drive the system's state around in its state space (Fig. 9), such that the deterministic, autonomous analysis can still help to understand the DS' behavior. In other cases we may have to evoke more advanced concepts like that of a *pullback attractor* defined through the limit $t_0 \to -\infty$ (Kloeden & Yang, 2020). More importantly, essentially all DSR models naturally can handle external inputs, important theorems still apply in that case (Mikhaeil et al., 2022), and many DSR approaches that explicitly incorporate process (dynamical) noise have been advanced (Kramer et al., 2022; Koppe et al., 2019; Pals et al., 2024; Li et al., 2020; Oh et al., 2024).

Second, most of the limitations noted above concern more the *analysis* of trained models rather than their application. The training algorithms developed in this field (Mikhaeil et al., 2022; Hess et al., 2023; Platt et al., 2023), in contrast, are largely indifferent w.r.t. the system's dimensionality or noise level. Partial observations can be taken care of by delay-embedding the data (cf. Appx. A.2; Takens (1981); Sauer et al. (1991)), and generally do not decimate DSR performance that much (Fig. 28a). DSR models have also been observed to *implicitly delay-embed* TS in their latent space (Hart, 2025). After all, DSR models have in fact been applied to high-dimensional, complex, noisy, partially observed empirical data such as physiological recordings or temperature records (Mikhaeil et al., 2022; Hess et al., 2023; Hemmer & Durstewitz, 2025) (see Tables 1 & 2, Fig. 5, & Appx. F), and may even outperform TS models as noise levels increase (Fig. 28b). *Analysis* of high-dimensional DS is also an area of active research, with several algorithms advanced in recent years to dissect the topological structure of high-dimensional models and to detect fixed points, cycles, or un-/stable manifolds, for instance (Sussillo & Barak, 2013; Eisenmann et al., 2023; Pals et al., 2024; Eisenmann et al., 2026; Dabholkar & Barak, 2025). The different approaches are also not mutually exclusive. *Ergodic theory* (Appx. A.4), for instance, uses probability measures to characterize chaotic sets (Katok et al., 1995), and thus may build a natural bridge to statistical approaches and stochastic DS. Or DSR models may also be integrated into (fine-tuned jointly with) LLMs to inform the DSR process by external events and driving functions based on the LLM's world and context knowledge.

Some practitioners may argue that they are only interested in short-term prediction, and neither in long-term properties nor understanding or extrapolation to unobserved regimes. While, in principle, this may be a valid stance, we point out that many DSR models, despite their much more parsimonious structure and lower parameter load, are at least on par with most TS models (Fig. 5, Tab. 1 & 2), and in some circumstances may even outperform them (Fig. 28b). Fig. 28c further explicitly demonstrates that DSR training techniques may improve short-term prediction as well.

Finally, are there any TS-generating systems or TS data not accessible to a DSR approach at all? Given that almost all natural or engineered systems evolve according to some dynamical rule, the only exceptions we can think of are systems which are either fully stochastic and thus genuinely unpredictable for any TS model beyond the distributional moments, or data generated by some artificial construction rule, like sequential MNIST (Fig. 29, however, shows that DSR approaches may even work for such cases). Generally, there are also statistical tests, like phase-randomized bootstraps (Kantz & Schreiber, 2004), one could use to probe for nonlinear-deterministic structure in observed TS (although, pragmatically, one may just try out different TS and DSR models on the TS at hand). Nevertheless, it is likely that not for all types of TS data the DSR approach is equally well suited. While DSR methods have been developed also for count, ordinal, or categorical data (Kramer et al., 2022; Brenner et al., 2024b), such data types are certainly more challenging, especially when the TS are rather short (Volkmann et al., 2024). Other data constraints may even impose principle limits on the feasibility of DSR (Göring et al., 2024; Park et al., 2024; Shumaylov et al., 2025).

# 6. Conclusions and Call to Action

What are the major take homes of the preceding discussion for the TS analysis field? Here we summarize our major recommendations for the further development of the field:

### ① Incorporate DSR training techniques

First, as our analyses in Figs. 5 & 28b,c suggest, *incorporating training techniques from the DSR field* into TS modeling may pay off, *even if the goal is just short-term prediction* (most explicitly shown in Fig. 28c). Methods like STF (Mikhaeil et al., 2022) and GTF (Hess et al., 2023), or regularization criteria based on invariant measures (Platt et al., 2023; Jiang et al., 2023), are general and do not depend much on the specific architecture used. For instance, as shown in Fig. 30, GTF may also improve the performance of Mamba (Gu & Dao, 2024). These techniques do not impose much of an additional burden in training, and allow for predicting *longer-term horizons*, in particular *statistical long-term properties*, and potentially other features. Better training may also enable to *simplify models* and *reduce the computational resources* required.

### ② Pre-train FMs on simulations from DS models

Another important aspect concerns the training data used: Instead of artificially assembled TS that might lack many of the features present in TS produced by real-world DS, we suggest *to employ DS simulation data* in TS FMs as in DynaMix (Hemmer & Durstewitz, 2025) or Panda (Lai et al., 2025), which emulate realistic time courses and patterns in TS data. Chaotic DS might be of particular advantage here, since many of them naturally express an infinity of timescales and temporal patterns, *injecting sufficient temporal variety* into the training process. Fig. 12 provides some examples of what such a training corpus may look like.

### ③ Move back to modern RNN variants for TSF

The field may want to rethink the current transformer dominance. While transformers certainly were 'transformative' for NLP and many other areas of AI, they may be less suited for TS modeling (Zeng et al., 2023; Tan et al., 2024; Hewamalage et al., 2023). They lack a natural representation of time and hence cannot easily model truly dynamical processes in the data, which are defined through recursion in continuous or discrete time. Rather, they operate on the principle of temporal pattern recognition, as evidenced by the failure modes of models such as Chronos or Panda (Hemmer & Durstewitz, 2025). In our minds the field should *put a stronger emphasis on (discrete- or continuous-*

*time) RNNs*, for which GPU-efficient techniques as in Mamba (Gu & Dao, 2024; Dao & Gu, 2024), xLSTM (Beck et al., 2024), or RNNs more generally (Lim et al., 2024; Gonzalez et al., 2024; Hess et al., 2026), make them also computationally increasingly attractive alternatives to transformers. RNNs are further dynamically universal and can approximate arbitrary DS (Hanson & Raginsky, 2020; Aguiar et al., 2023). Another option might be to integrate RNNs into transformers, as has been done in recent foundation models for physiological data (Ma et al., 2026; Wang et al., 2026).

### ④ Address the hard problems: topological shifts

DS theory also provides a more nuanced view on the issue of OOD generalization in TS modeling (Göring et al., 2024), and may help to crack problems (tipping points) that the TS field felt were largely out of scope so far. The hardest problems are those where there is not just a distribution shift from training to test, or even *within* a given non-stationary TS, but where the TS-generating DS undergoes a *topological shift*, entering a new dynamical regime. Understanding and reconstructing the underlying DS from data and approximating its governing equations provides potential solutions to such forecasting problems. Currently there are three major ways we see for addressing this: First, *simulations from DS which undergo various types of tipping points* may be explicitly included in the training data (as in Fig. 12). Second, methods may be devised for *extracting control parameters from TS* and using them to extrapolate into unseen dynamical regimes (Huh et al., 2025; Brenner et al., 2025). Third, generic inductive biases (not dependent on the specific physical system) like separation of timescales (Schmidt et al., 2021) or skew-product forms (Kloeden & Yang, 2020) may be used to separate slowly evolving control parameters from the faster state dynamics, as illustrated in Fig. 13.

### ⑤ Focus on mechanisms underlying TS generation

Finally, and more generally, we encourage to adopt a DS perspective on TS processes as it provides a kind of *mechanistic understanding* of TS generation, without requiring specific domain knowledge in the area the TS come from. Such an understanding may help to forecast the behavior of a system under unseen, novel, and unexpected conditions, e.g. when external events such as natural disasters or changes in governmental policies impact the observed process. Integrating DSR models into LLMs for informing DSRs by specific context or physical domain knowledge may also be a fruitful direction (Williams et al., 2025).

## Acknowledgements

This work was funded by the German Research Foundation (DFG) through individual grants Du 354/15-1 (project no. 502196519) & Du 354/18-1 (project no. 567025973) to DD, and via Germany's Excellence Strategy EXC 2181/1 – 390900948 (STRUCTURES). We also would like to thank Konstantin Dibbern for providing the code and model for producing Fig. 2b.

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

# A. Core Concepts in Dynamical Systems Theory

DS theory is the mathematical theory which examines the temporal behavior of systems that are described by sets of (coupled) differential equations, $d\boldsymbol{x}/dt \equiv \dot{\boldsymbol{x}} = f(\boldsymbol{x})$ with *vector field* $f(\boldsymbol{x})$, or recursive maps, $\boldsymbol{x}_t = F(\boldsymbol{x}_{t-1})$, $\boldsymbol{x} \in E \subseteq \mathbb{R}^M$ (Guckenheimer & Holmes, 1983; Alligood et al., 1996; Perko, 2001; Strogatz, 2024). When a DS *explicitly* depends on time, i.e. $\dot{\boldsymbol{x}} = f(\boldsymbol{x}, t)$ or $\boldsymbol{x}_t = F(\boldsymbol{x}_{t-1}, \boldsymbol{s}_t)$, it is called *non-autonomous*, otherwise we call it *autonomous*. In theory, a non-autonomous DS can always be recast as an autonomous DS by introducing an additional variable $x_{M+1} = t$, $\dot{x}_{M+1} = 1$. This is useful for some mathematical purposes, but may be deceptive in other cases as it tends to 'hide' the truly non-converging behavior of many non-autonomous DS. A DS is generally defined as a (semi-)group $(T, E, \Phi)$ with a set of times $T$ (typically $T = \mathbb{R}$ in continuous time, and $T = \mathbb{Z}$ in discrete time), a set of states $E \subseteq \mathbb{R}^M$ (the state space), and a *flow* or evolution law $\Phi(t, \boldsymbol{x}_0) \equiv \Phi_t(\boldsymbol{x}_0), t \in T, \boldsymbol{x}_0 \in E$, with the property $\Phi_{s+t} = \Phi_s \circ \Phi_t = \Phi_t \circ \Phi_s$.

## A.1. Limit Sets and Attractors

Consider a DS defined by $(\mathbb{R}, E, \Phi_t)$. For $t \to -\infty$ and $t \to \infty$, $\Phi_t(\boldsymbol{x}_0)$ will often converge to certain sets for an $\boldsymbol{x}_0 \in E$, called the $\alpha$ and $\omega$ limit sets, respectively (Guckenheimer & Holmes, 1983):

**Definition A.1.** For a flow $\Phi_t : E \to E$ and initial condition $\mathbf{x}_0 \in E$, the $\omega$-limit set (forward limit set) of $\mathbf{x}_0$ is defined as

$$\omega(\mathbf{x}_0) = \bigcap_{T>0} \overline{\{\Phi_t(\mathbf{x}_0) : t \geq T\}}, \tag{2}$$

and the $\alpha$-limit set (backward limit set) as

$$\alpha(\mathbf{x}_0) = \bigcap_{T>0} \overline{\{\Phi_t(\mathbf{x}_0) : t \leq -T\}}, \tag{3}$$

where the overline denotes closure of the set.

Often these sets are isolated points (equilibria), closed orbits (limit cycles), quasi-periodic sets densely filling a torus (i.e., $\alpha(\mathbf{x}_0), \omega(\mathbf{x}_0) = \mathbb{T}^k$), or chaotic sets, although there are several other possibilities like manifolds of dense (non-isolated) equilibria or cycles, homoclinic orbits (connecting an equilibrium to itself), heteroclinic orbits (connecting two different equilibria), or the entire state space, for instance. An *attractor* is a special type of $\omega$-limit set with the following properties:

**Definition A.2 (Attractor and basin of attraction).** Let $(\mathbb{R}, E, \Phi)$ be a DS with state space $E \subseteq \mathbb{R}^M$ and flow $\Phi_t(\boldsymbol{x})$. An attractor $\mathcal{A} \subset \mathcal{B} \subseteq E$ is a closed set with the following properties:
1. $\Phi_t(\mathcal{A}) \subseteq \mathcal{A} \quad \forall t$ (invariance under the flow)
2. $\forall \boldsymbol{x}_0 \in \mathcal{B}, \ t \geq 0 : \lim_{t \to \infty} d(\Phi_t(\boldsymbol{x}_0), \mathcal{A}) = 0$ (convergence from an open set $\mathcal{B}$, the *basin of attraction*)
3. $\mathcal{A}$ is the minimal set with such properties, while $\mathcal{B}$ is the maximal such set.

An equilibrium of a continuous-time DS $\dot{\boldsymbol{x}} = f(\boldsymbol{x})$ is defined by the condition $f(\boldsymbol{x}) = 0$, while a fixed point of a recursive map $F$ is a point $\boldsymbol{x}^*$ for which we have $\boldsymbol{x}^* = F(\boldsymbol{x}^*)$. Equilibria (and likewise fixed points) of DS can be classified according to the eigenvalue spectrum of the local Jacobians at those points. Consider a *linear* continuous-time DS $\dot{\boldsymbol{x}} = f(\boldsymbol{x}) = \boldsymbol{A}\boldsymbol{x}$, which has the general solution $\boldsymbol{x}(t) = \exp(\boldsymbol{A}t)\boldsymbol{x}_0$ (Perko, 2001). Such a system cannot exhibit any limit cycles (stable oscillations; only dense sets of closed orbits given by purely trigonometric forms), multistability, or chaos. Define $\lambda_i := \text{eig}_i(\boldsymbol{A}), i = 1 \ldots M$, as the eigenvalues of $\boldsymbol{A}$, with real and imaginary parts $\alpha_i = \text{Re}(\lambda_i)$ and $\omega_i = \text{Im}(\lambda_i)$, respectively. Then a *node* is a point for which $\forall i : \omega_i = 0$, and a *spiral* a point for which $\exists i : \omega_i \neq 0$ (in the latter case there are at least two eigenvalues with imaginary parts, since they always come as conjugate pairs). For a spiral point, we have damped or growing oscillations in its vicinity. An equilibrium is called *stable* (making it a point attractor) if $\forall i : \alpha_i < 0$, *unstable* (a repeller) if $\forall i : \alpha_i > 0$, and a *saddle* if $\exists i : \alpha_i < 0 \ \wedge \ \exists j : \alpha_j > 0$ (analogous definitions exist for discrete-time DS defined by recursive maps). Importantly, these concepts carry over to *nonlinear* DS by virtue of the Hartman-Grobman theorem (Perko, 2001), which states that for hyperbolic DS (see below) in a neighborhood $N_\epsilon(\boldsymbol{x}_0)$ of an equilibrium $\boldsymbol{x}_0$, the nonlinear DS is *topologically conjugate* (see Def. A.4) to a linear DS with the Jacobian $\boldsymbol{J}_{\boldsymbol{x}_0} := \frac{\partial f(\boldsymbol{x})}{\partial \boldsymbol{x}}|_{\boldsymbol{x}_0}$ as its evolution operator.

If there exists one $\alpha_i = 0$ (making the system *non-hyperbolic*), then we have a dense, continuous set of equilibria or cycles along those directions. If there is either no motion or convergence along all other directions (i.e., $\forall j \neq i : \alpha_j \leq 0$), the set is called *marginally* stable (sometimes called a *manifold attractor*) and has some interesting properties. Manifold attractors are

interesting from an ML and a TSA perspective, since they form perfect integrators of perturbations into the system and can be exploited for long-range sequential problems and for learning DS with arbitrary timescales (Schmidt et al., 2021).

A *limit cycle* is defined as an *isolated* (not dense) closed orbit $\Omega$ in state space (i.e., for which we have $\Phi_t(\boldsymbol{x}_0) = \Phi_{t+\tau}(\boldsymbol{x}_0)$ for a fixed period $\tau > 0$ and all $\boldsymbol{x}_0 \in \Omega$). Like equilibria, limit cycles may be stable (attractors), unstable, or of the saddle type. An analogous concept for a discrete-time DS $F$ is that of a $k$-cycle, which is an orbit $\Omega = \{\boldsymbol{x}_1^*, \boldsymbol{x}_2^*, \ldots, \boldsymbol{x}_k^*\}$ such that all $\boldsymbol{x}_i^*, i = 1 \ldots k$, are *distinct* and $\forall i : \boldsymbol{x}_i^* = F^k(\boldsymbol{x}_i^*)$. Hence, for a discrete-time DS each cyclic point is a *fixed point* of the $k$-times iterated map $F$, and its stability can thus be determined from the Jacobians of $F^k(\boldsymbol{x}_i^*)$.

Unlike on a limit cycle, trajectories on a *chaotic attractor*, characterized by high sensitivity to initial conditions due to the maximum Lyapunov exponent $> 0$ causing exponential divergence, *never* close up (i.e., there is no $\tau > 0$ for which $\Phi_t(\boldsymbol{x}_0) = \Phi_{t+\tau}(\boldsymbol{x}_0)$) and hence are *aperiodic* and *irregular*. While for a chaotic attractor we have $\lambda_{\max} > 0$, for the object to be an attractor, we still need $\sum \lambda_i < 0$.

A final remark concerns the relation between discrete and continuous time DS, which may be connected through the idea of the flow map. Consider, for convenience, a linear continuous-time DS $\dot{\boldsymbol{x}} = \boldsymbol{A}\boldsymbol{x}$ with solution $\boldsymbol{x}(t) = \exp(\boldsymbol{A}t)\boldsymbol{x}_0$. Then assuming we sample the continuous-time DS at fixed intervals $k\Delta t, k \in \mathbb{Z}$, and defining $\tilde{\boldsymbol{A}} := \exp(\boldsymbol{A}\Delta t)$, the discrete-time DS defined by $\boldsymbol{x}_k = \tilde{\boldsymbol{A}}\boldsymbol{x}_{k-1}$ produces outputs *equivalent* to those of the continuous-time DS on the temporal domain it is defined, and provided it is started in the same initial condition. This idea can be extended to define an equivalence between discrete and continuous time ReLU-based RNNs (Monfared & Durstewitz, 2020), and applied at least approximately to translating between arbitrary nonlinear continuous- and discrete-time DS (Ozaki, 2012). Another important tool to connect continuous to discrete time DS is the *Poincaré map* (Perko, 2001): In a Poincaré map, an $M-1$-dimensional manifold $\Sigma$ transversal to the flow of an $M$-dimensional continuous time DS is inserted into its state space, and consecutive intersections $\boldsymbol{x}_k = \Phi_t(\boldsymbol{x}_0) \cap \Sigma$ from one direction are recorded and formulated as a map $\boldsymbol{x}_k = F(\boldsymbol{x}_{k-1})$. These close formal connections between discrete and continuous-time DS can often been exploited for DS analysis; e.g., we can analyze the stability of a limit cycle through the stability of a cyclic point in a corresponding map, which is usually a simpler problem.

### A.2. Temporal Delay Embedding

Say we have observed an underlying DS $\dot{\boldsymbol{x}} = f(\boldsymbol{x})$, $\boldsymbol{x} \in \mathbb{R}^M$, through some generic measurement function $y_\tau = g(\boldsymbol{x}(\tau))$, $\boldsymbol{y} \in \mathbb{R}^N$, at time points $\tau = 1 \ldots T$, yielding an observed TS $\{y_\tau\}$. Let us consider the case $N = 1 << M$, i.e. a scalar TS $\{y_\tau\}$. Is there any hope of recovering the attractors of the original DS from just these scalar measurements? One of the most surprising and fundamental results as one moves from DS theory to empirical data is the delay embedding theorems, which state that in general the answer is 'yes' under quite generic conditions (Takens, 1981; Sauer et al., 1991). Loosely speaking, if all degrees of freedom in the underlying DS are coupled (i.e., our observations are not just from a subsystem which is completely isolated from another subsystem we are interested in), we can replace the missing observations from other dynamical variables through *time-lagged* versions $y_{\tau-k\Delta\tau}, k \in \mathbb{N}$. Intuitively, a single observation $y_\tau$ won't allow us to predict the future evolution of the underlying system because it is ambiguous, but the more past observations $y_{\tau-\Delta\tau}, y_{\tau-2\Delta\tau}, \ldots$, we consider, the more constrained the potential future course of a trajectory becomes. Specifically, we build a *delay coordinate vector* from the scalar observations as follows:

$$\boldsymbol{y}_t := \begin{bmatrix} y_t, & y_{t-\Delta t}, & \ldots, & y_{t-(m-1)\Delta t} \end{bmatrix}. \tag{4}$$

This is also called a *delay coordinate map*, let us denote it by $H$. How should we choose the time lag $\Delta t$ and the *embedding dimension* $m$? The answer for $m$ is given by the *Fractal Delay Embedding Prevalence Theorem*, which we loosely state here following (Sauer et al., 1991):

**Theorem A.3.** *Let $\Phi : E \to E$ be a $C^r$ ($r \geq 2$) flow on an open set $E \subseteq \mathbb{R}^M$. Let $A \subseteq E$ be a compact set invariant under $\Phi$ (i.e., $\Phi(A) \subseteq A$) with box-counting dimension $d_{box}$. Let $g : \mathbb{R}^M \to \mathbb{R}$ be a smooth measurement function and $H_{\Delta t} : E \to \mathbb{R}^m$ be a delay coordinate map with time lag $\Delta t$, defined on the measurements as in Eq. 4. Then, if $m > 2d_{box}$, for almost every (in the sense of prevalence) choice of $g$ and $\Delta t$ the map $H_{\Delta t}$ restricted to $A$ is*

1. *one-to-one on $A$ (injective),*
2. *an immersion on $A$, i.e. the derivative map $DH$ has full rank at every point in $A$.*

*Therefore, $H$ restricted to $A$ is a diffeomorphism onto its image $H(A)$.*

By 'prevalent' measurement function $g$ we mean a proper $g$ is dense in the space of functions, i.e. with probability $\to 1$ $g$ will have the required properties. By 'almost every' $\Delta t$ we mean that there are only specific, measure-0 choices of $\Delta t$

that don't work, e.g. $\Delta t$ should not correspond to the exact multiple of the period of a limit cycle we aim to reconstruct. Intuitively, the condition $m > 2d_{\text{box}}$ guarantees that trajectories cannot intersect in the embedding space (two sets $A$ and $B$ with dimensions $m$ and $n$, respectively, will intersect with probability $\to 0$ in a $d > m + n$ dimensional space), and that there are no discontinuities in the embedded vector field.

Practically, we can determine $m$ by the technique of false-nearest neighbors (Kennel et al., 1992): False nearest neighbors are defined as points between which the distance suddenly grows over-proportionally as we move from an $m$- to a $m + 1$-dimensional embedding space, indicating that in the $m$-dimensional space trajectories were not yet properly disentangled (but close to forming an intersection). The optimal time lag $\Delta t$, on the other hand, while nearly irrelevant theoretically, matters practically in noisy settings: If $\Delta t$ is too small, the geometric structure will not be sufficiently resolved as data points tend to cluster in lower-dimensional subspaces due to high auto-correlations among immediate temporal neighbors. If, on the other hand, $\Delta t$ is too large, points in the embedding space will tend to erratically jump around as all correlations are lost. Hence, we seek an intermediate choice of $\Delta t$, often determined close to the first trough of the auto-correlation function or auto-mutual information (Kantz & Schreiber, 2004).

More recent methods extend the principle of delay embeddings to multivariate TS observations and non-uniform lags between time points (Krämer et al., 2021). Delay embedding theorems have also been advanced for *non-continuous* observations like point processes (Sauer, 1994; 1995). Most DSR models, in principle, can perform 'intrinsic delay embeddings' in their latent space (Duan et al., 2023), but practically an explicit prior delay embedding of the data may often help and ease the task for the DSR algorithm, providing it direct access to temporal context at each time step.

### A.3. Bifurcations and Tipping Points

#### A.3.1. BIFURCATIONS

Consider a continuous-time dynamical system $\dot{\boldsymbol{x}} = f(\boldsymbol{x}, \boldsymbol{c})$ with $\boldsymbol{x} \in \mathbb{R}^M$ and $\boldsymbol{c} \in \mathbb{R}^{d_c}$ a vector of control parameters. A bifurcation is a *topological change* in the structure or stability of attractors and their basins that occurs when a control parameter passes a critical value called the bifurcation point (Kuznetsov, 1998). Formally, by topological change we mean that the vector fields (state spaces) before and after the bifurcation are no longer *topologically equivalent* according to the following definition:

**Definition A.4** (**Topological equivalence and conjugacy**). Let $(\mathbb{R}, A, \Phi)$ and $(\mathbb{R}, B, \Psi)$ be two DS. These are said to be *topologically equivalent* if there is a homeomorphism $h : A \to B$ such that for each $\boldsymbol{x}_0 \in A$ we have $h(\Phi_t(\boldsymbol{x}_0)) = \Psi_\tau(h(\boldsymbol{x}_0))$ with $\frac{d\tau(t, \boldsymbol{x}_0)}{dt} > 0$ everywhere (i.e., if $h$ preserves the direction of flow). They are said to be *topologically conjugate*, if $\tau(t, \boldsymbol{x}_0) = t$ (i.e., if the parameterization by time is the same).

Different types of bifurcations can be distinguished, depending on whether they depend on one (codimension 1) or more (codimension 2 etc.) control parameters, on whether they are local (at a fixed point) or global, and depending on which types of objects (fixed points, cycles, etc.) and what types of stability changes they involve. Like all other DS phenomena, bifurcations are *universal* phenomena, i.e. can be observed in equivalent form in many different systems from interacting molecules to interacting individuals in a society (Strogatz, 2024).

Local bifurcations are defined by their universal normal forms, simplified canonical equations that reproduce the dynamics in a neighborhood of such bifurcation points to which the original system can be reduced by a suitable change of variables (Guckenheimer & Holmes, 1983), examples of which are provided in Fig. 6. The minimal state space dimension required for a particular bifurcation is thus given by the dimension of its normal form (Kuznetsov, 1998).

In practice, a small number of bifurcations already captures many qualitative regime changes observed in real systems. A prominent example in one dimension is the saddle-node bifurcation, whose normal form describes the creation (or annihilation) of a pair of equilibria, one stable and one unstable (Fig. 6). Another common example is the pitchfork bifurcation, often observed in systems with certain symmetries, which gives rise to two symmetric branches of equilibria as one equilibrium in the center changes its stability. Saddle node and pitchfork bifurcations can, for instance, be observed in a single neural unit with sigmoid activation function as the bias parameter (saddle) or the weight and thus slope of the sigmoid (pitchfork) is varied (Doya, 1992). In two dimensions, the Hopf bifurcation is the most common local mechanism for the emergence or disappearance of a limit cycle around a spiral point, a stable limit cycle in the supercritical case and an unstable one in the subcritical case, with growing amplitude as the control parameter changes. Hopf bifurcations can give rise, for instance, to spiking activity in real neurons (Izhikevich, 2007).

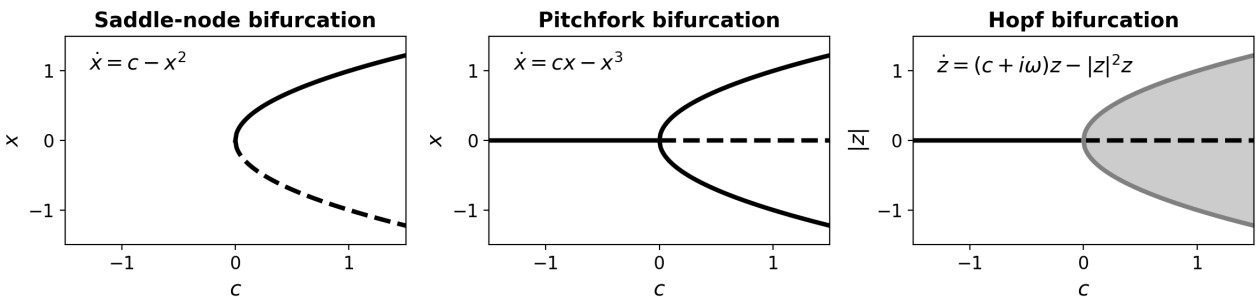

*Figure 6.* Normal forms and bifurcation diagrams for saddle-node and supercritical pitchfork and Hopf bifurcations. Shown are the stable and unstable equilibria as solid and dashed black lines, respectively, and the limit cycle as shaded region between its minimum–maximum branches (gray lines), as the control parameter $c$ in the respective normal form is varied.

Beyond the mere onset of oscillations, other bifurcations of periodic orbits can provide standard routes to chaos. Period-doubling bifurcations occur when a periodic orbit loses stability and is replaced by a new orbit of twice the period, and repeated iterations of this local mechanism can produce a cascade of doublings that accumulates until the onset of chaotic dynamics with a backbone of infinitely many unstable periodic orbits (Guckenheimer & Holmes, 1983). Another classical, but global, route to chaos is via homoclinic bifurcations, where the stable and unstable manifolds of a saddle reconnect to form a homoclinic loop, in the vicinity of which trajectory flows become highly sensitive to minuscule perturbations (Strogatz, 2024). Homoclinic orbits lead to 'reinjection' of trajectories into the same region of space, one of the mechanisms by which fractal structure of a chaotic attractor can be created.

### A.3.2. TIPPING POINTS

Tipping points are abrupt changes in the dynamical behavior of a DS, corresponding to qualitative (topological) changes (cf. Def. A.4) in the effective attractor governing the dynamics. While most commonly tipping points in TS may be induced by bifurcations (called B-tipping), at least two other types of tipping mechanisms have been identified, termed R-tipping (rate-induced, caused by rapid changes in external forcing) and N-tipping (noise-induced, driven by fluctuations) (Ashwin et al., 2012).

**B-tipping** As control parameters vary, attractors of a DS can undergo bifurcations, leading to modifications of vector field topology, and thus changes in the behavior of observed trajectories. From a modeling perspective, predicting such B-tipping events as well as post-tipping dynamics requires learning not only the state evolution but also its dependence on the underlying control parameter. In the DSR context, this can, for example, be implemented by adding an explicitly time-dependent forcing function to the reconstruction model (Kong et al., 2021; Köglmayr & Räth, 2024; Van Tegelen et al., 2025), or by using a hierarchical scheme in which a given or learnable parameter controls the parametrization of model instances (Brenner et al., 2025; Huh et al., 2025). In Fig. 2b we gave an example of B-tipping in a spiking neuron model (Appx. D.2). For this, the model parameter $g_{NMDA}$ (Eq. 35) was linearly varied over time, and an AL-RNN was trained only on a piece of trajectory *prior* to the bifurcation into the regular (simple) spiking regime (without any knowledge of the control parameter).

**R-tipping** Non-autonomous DS with time-dependent control parameters $c(t)$ may also exhibit tipping in the absence of bifurcations, if the rate of change, $r = dc/dt$, is too fast for trajectories to adiabatically follow the evolving attractor. In such cases, the system will undergo a qualitative shift in its dynamic behavior, as trajectories are pulled out from one quasi-static stable set associated with the current parameter value. In multistable DS this may cause the state to tip across a basin boundary, pushing it towards a different attractor (Ashwin et al., 2012). Recent deep learning approaches aimed at estimating tipping probabilities in systems undergoing R-tipping (Huang et al., 2024), but so far, to our knowledge, no attempt has been made to capture the phenomenon with a proper DSR model.

**N-tipping** In multistable systems with fixed parameters (as in Fig. 1), transitions between coexisting stable states can be triggered by stochastic fluctuations that push the state across the boundary of one basin of attraction into another basin (see Def. A.2), leading to an abrupt regime shift in absence of a bifurcation (Ashwin et al., 2012). One way of approaching such scenarios in multiscale systems is to infer an effective fast driving process from observations of the slow dynamics and to

model this forcing separately for prediction (Lim et al., 2020). Figure 1b (see also Fig. 9) gives an example of N-tipping where noise pushes the system from the point attractor into the limit cycle attractor regime. To create this specific example, an AL-RNN was trained on trajectories from both attractor basins, and both the AL-RNN after training and the neuron model were run with additive Gaussian noise.

### A.4. Ergodic Theory and Stationarity

Strictly, stationarity is not a DS but a *statistical* concept that is fundamental to the understanding of some of the most challenging problems in TSA. It is, however, intimately related to important concepts and phenomena in DS theory, specifically to *ergodic theory* through Birkhoff's ergodic theorem and the idea of measure-preserving maps (Billingsley, 2017). For instance, if a process is ergodic, it is also strictly stationary, and if it is strictly stationary, it is measure-preserving (i.e., the measure is invariant under the map describing the process). Most chaotic attractors have the property that the underlying process on the attractor is *topologically mixing*, defined as (Katok et al., 1995; Lind & Marcus, 2021)

**Definition A.5 (Topologically mixing).** Let $f \in C^1(E)$ be a vector field defined on an open set $E$, with associated flow map $\Phi_t$. The DS given by $(\mathbb{R}, E, \Phi_t)$ is said to be *topologically mixing* if for any two open nonempty subsets $A, B \subseteq E$ there is a $t_0 > 0$ such that $\forall t \geq t_0 : \Phi_t(A) \cap B \neq \emptyset$.

For *measure-theoretic* mixing we require $\lim_{t \to \infty} \mu(\Phi_t(A) \cap B) = \mu(A)\mu(B)$. This mixing condition implies ergodicity, hence also strict (strong-sense) stationarity. If a dynamical process crosses a tipping point, a *topological shift* is induced, and the process is no longer ergodic or stationary.

In mathematical terms, *strong sense stationarity* is defined by the condition that all moments of the joint probability distribution across TS observations $\boldsymbol{X} = \{\boldsymbol{x}_t\}$ are constant in time, i.e. it requires $p(\{\boldsymbol{x}_t\}|t_0 \leq t \leq t_1) = p(\{\boldsymbol{x}_t\}|t_0 + \Delta t \leq t \leq t_1 + \Delta t) \, \forall t_0 \leq t_1, \Delta t \geq 0$. It is important to note that this definition is assumed to be across an *ensemble* of TS drawn from many different random realizations of that process (Durstewitz, 2017); this is the reason why an oscillatory process is still considered stationary, because if we draw many TS realizations from that process with random initial phase, *across* the ensemble of these realizations the stationarity conditions will be fulfilled. A process is called *weak sense stationary* if the conditions above hold only for the first two moments of $p(\boldsymbol{X})$, i.e. if $\mathbb{E}(\boldsymbol{x}_t) = const. \, \forall t$ and $\mathrm{Cov}(\boldsymbol{x}_t, \boldsymbol{x}_{t+\Delta t}) = \mathrm{Cov}(\Delta t) \, \forall t, \Delta t$, i.e. if the mean and autocorrelation are constant across time and for all temporal lags.

## B. DSR Measures

In DSR, we are primarily concerned with the topology, geometry, and temporal structure of attractors, and hence with the behavior of the system in the ergodic limit $t \to \infty$. Topologically, a DSR may be defined as follows (Durstewitz et al., 2023):

**Definition B.1 (Topological DSR).** Let $D = (T, E, \Phi)$ be a true data-generating DS and $D^* = (T^*, E^*, \Phi^*)$ a reconstructed (modeled) DS, with $E, E^* \subseteq \mathbb{R}^N$ open sets, and $h : E \to E^*$ and $\tau : T \to T^*$ homeomorphisms. Let $A \subset B \subseteq E$ be an attractor of $\Phi$ with $B$ its basin of attraction. We say that $D^*$ is a *partial DSR* of $D$ on $B$ if $\Phi^*$ on $B^*$, with $\tilde{h} : B \to B^*$ the restriction of $h$ to $B$, is *topologically conjugate* (see Def. A.4) to $D$ on $B$. If the topological conjugacy holds for the whole of state space $E$, $D^*$ is a *full DSR* of $D$ on $E$.

This may be difficult to establish in practice, and, moreover, beyond topological agreement, we are also often interested in preserving *geometric* and precise *temporal* (same time scales) properties of the original DS. This can be assessed through the measures introduced below. For calculating any of these, it is *mandatory to draw sufficiently long trajectories*, which in practice means generating long autoregressive roll-outs from the trained model (of, depending on the timescales in the data, at least $T = 1,000 - 10,000$ time steps), from which the initial transients are discarded (unless the model is already initialized on the attractor).

### B.1. Characterizing Attractor Geometry

B.1.1. FRACTAL GEOMETRY

Attractors of chaotic DS famously have a *fractal geometry*, characterized by self-similarity across all spatial scales (Eckmann & Ruelle, 1985; Alligood et al., 1996; Kantz & Schreiber, 2004). This is a consequence of the divergent yet bounded nature of trajectories on chaotic attractors, which get 'reinjected' into the same region of space infinitely many times without ever closing up (Strogatz, 2024). The resulting structures are topologically often well characterized by the Smale horseshoe

map, which describes the infinitely repeated stretching of a set along one and contraction along another direction, together with the folding of the set onto itself (Guckenheimer & Holmes, 1983), resulting in a $2d$ Cantor set (which has the peculiar property of being of length (measure) 0, yet consisting of uncountable infinitely many points).

One idea to compute the dimension of such a chaotic set is to cover it with boxes of edge length $\epsilon$, and then determine the number $N_\epsilon$ of boxes needed to cover the whole object in the limit $\epsilon \to 0$. This gives rise to the so-called *box-counting dimension* defined as (Eckmann & Ruelle, 1985; Alligood et al., 1996)

$$d_{\text{box}} := \lim_{\epsilon \to 0} \frac{\log N_\epsilon}{\log 1/\epsilon} \ . \tag{5}$$

If this concept is applied to a classical $1d$, $2d$, $3d$ etc. geometric object like a line segment, disc, or cube, $d_{\text{box}}$ will be an integer number (i.e., $1, 2, 3$, respectively). For Cantor sets and chaotic attractors, however, $d_{\text{box}}$ will be a fraction, e.g. $d_{\text{box}} \approx 2.06$ for the Lorenz-63 attractor in Fig. 3. Practically, we can determine $d_{\text{box}}$ as the (negative) slope of a line fitted to the graph of $\log N_\epsilon$ vs. $\log \epsilon$.

Another, empirically in higher dimensions easier to apply estimate of the fractal dimension is the *correlation dimension* (Eckmann & Ruelle, 1985; Kantz & Schreiber, 2004). Here the idea is to place balls of radius $\epsilon$ on the points $\boldsymbol{x}_t$ along observed trajectories $S_T = \{\boldsymbol{x}_1, \boldsymbol{x}_2, \ldots, \boldsymbol{x}_T\}$, and count the number $C(\epsilon)$ of neighbors within those balls:

$$C(\epsilon) := \lim_{T \to \infty} \frac{\#\{(\boldsymbol{x}_i, \boldsymbol{x}_j) | \|\boldsymbol{x}_i - \boldsymbol{x}_j\|_2^2 < \epsilon\}}{\#\{(\boldsymbol{x}_i, \boldsymbol{x}_j) | \boldsymbol{x}_i, \boldsymbol{x}_j \in S_T\}} \ . \tag{6}$$

$C(\epsilon)$ is also called the correlation integral. Considering the limit $\epsilon \to 0$, we then obtain the definition

$$d_{\text{corr}} := \lim_{\epsilon \to 0} \frac{\log C(\epsilon)}{\log \epsilon} \ . \tag{7}$$

In **implementing these measures**, besides the fact that we require sufficiently long trajectories for the convergence of these measures, there are two practical considerations (Kantz & Schreiber, 2004): First, we need to remove purely temporal neighbors in the calculation of $C(\epsilon)$ and $d_{\text{box}}$, i.e. pairs $(\boldsymbol{x}_t, \boldsymbol{x}_{t'})$ for which $|t - t'| < \Delta t$, since we intend to capture the *spatial geometry* of the object independent of auto-correlations in the data. Second, for small $\epsilon$ the estimates will be conflated by noise, which tends to expand in all directions equally, so we are aiming for a reasonably linear region in the log-log plots after a potentially initially steep rise.

Another estimate of fractal dimensionality, the *Kaplan-Yorke* dimension $D_{\text{KY}}$ as provided in Fig. 3, an upper bound to the information dimension and often close to the box-counting and correlation dimensions (Frederickson et al., 1983; Eckmann & Ruelle, 1985; Kantz & Schreiber, 2004), is based on the DS' Lyapunov spectrum as defined in Eq. 1:

$$D_{\text{KY}} := j + \frac{\sum_{i=1}^{j} \lambda_i}{|\lambda_{j+1}|} \ , \tag{8}$$

where we have sorted Lyapunov exponents $\lambda_1 \geq \lambda_i \geq \lambda_j \geq \cdots \geq \lambda_M$ in descending order, and $j$ is the largest integer such that $\sum_{i=1}^{j} \lambda_i \geq 0$ and $\sum_{i=1}^{j+1} \lambda_i < 0$.

### B.1.2. DISTRIBUTIONAL MEASURES

Other measures are based on probability measures defined on the attractor (Eckmann & Ruelle, 1985; Kantz & Schreiber, 2004). Consider the *normalized occupation measure*, which 'measures the average amount of time' a trajectory starting in $\boldsymbol{x}_0$ across finite time $T$ spends in some region (set) $A$ of state space, defined by

$$\mu_{\boldsymbol{x}_0}(A, T) := \frac{1}{T} \int_0^T \mathbb{I}[\Phi_t(\boldsymbol{x}_0) \in A] dt \ , \tag{9}$$

where $\mathbb{I}$ is the indicator function and $\Phi_t$ the flow map. In the ergodic limit, $t \to \infty$, this converges to the so-called *invariant measure*, a probability measure defined on the attractor and invariant under the flow $\Phi_t$ (Eckmann & Ruelle, 1985). Common quantities used to compare geometries between true (observed) and reconstructed attractors are based on this normalized occupation measure.

**Kullback-Leibler divergence**   Specifically, consider for $T \to \infty$ and $\epsilon \to 0$ the probability densities $p(\boldsymbol{x})$ and $q(\boldsymbol{x})$ for true and model-generated distributions of trajectory points $\boldsymbol{x}$ across *state space* (potentially obtained via delay embedding), respectively, then one way to compare these distributions is the Kullback-Leibler divergence

$$D_{\text{stsp}} := D_{\text{KL}}\left(p(\boldsymbol{x}) \,\|\, q(\boldsymbol{x})\right) = \int_{\boldsymbol{x} \in \mathbb{R}^N} p(\boldsymbol{x}) \log \frac{p(\boldsymbol{x})}{q(\boldsymbol{x})} d\boldsymbol{x} \,. \tag{10}$$

For low-dimensional state spaces, we may just bin the space into $K$ disjoint sets $A_k$ which completely cover the attractors, and approximate $p(\boldsymbol{x})$ and $q(\boldsymbol{x})$ through the relative frequencies $\hat{p}_k(\boldsymbol{x}) = \frac{\#\{\boldsymbol{x}_t \in A_k\}}{T}$ and likewise for $\hat{q}_k(\boldsymbol{x})$ (Koppe et al., 2019; Mikhaeil et al., 2022; Hess et al., 2023), giving rise to the estimate

$$\hat{D}_{\text{stsp}} = D_{\text{KL}}\left(\hat{p}(\boldsymbol{x}) \,\|\, \hat{q}(\boldsymbol{x})\right) = \sum_{k=1}^{K} \hat{p}_k(\boldsymbol{x}) \log \frac{\hat{p}_k(\boldsymbol{x})}{\hat{q}_k(\boldsymbol{x})} \tag{11}$$

where $K = m^N$ is the total number of bins, with $m$ the number of bins per dimension, $\hat{p}_k(\boldsymbol{x}) \approx \mu_{\boldsymbol{x}_1}(A_k, T)$ (a finite, discrete time approximation) for the ground truth orbits $\boldsymbol{X} = \{\boldsymbol{x}_1, \dots, \boldsymbol{x}_T\}$ and $\hat{q}_k(\boldsymbol{x})$ likewise for the model-generated orbits.

For high-dimensional systems, where the binning approach becomes computationally prohibitive, the densities $p(\boldsymbol{x})$ and $q(\boldsymbol{x})$ may be approximated through Gaussian Mixture Models (GMMs) with Gaussians placed along orbits (Koppe et al., 2019; Brenner et al., 2022), i.e. $\hat{p}(\boldsymbol{x}) = 1/T \sum_{t=1}^{T} \mathcal{N}(\boldsymbol{x}; \boldsymbol{x}_t, \boldsymbol{\Sigma})$ and $\hat{q}(\boldsymbol{x}) = 1/T \sum_{t=1}^{T} \mathcal{N}(\boldsymbol{x}; \hat{\boldsymbol{x}}_t, \boldsymbol{\Sigma})$, where $\boldsymbol{x}_t$ and $\hat{\boldsymbol{x}}_t$ are observed and generated states, respectively, $\mathcal{N}(\boldsymbol{x}; \boldsymbol{x}_t, \boldsymbol{\Sigma})$ is a multivariate Gaussian with mean vector $\boldsymbol{x}_t$ and covariance matrix $\boldsymbol{\Sigma}$, where usually $\boldsymbol{\Sigma} = \sigma^2 \mathbf{I}_{N \times N}$ for standardized data, and $T$ is the orbit length. There is no closed-form analytical expression for the KL divergence between two GMMs, but different approximations based on variational principles or sampling are provided in Hershey & Olsen (2007). A Monte Carlo approximation is given by

$$D_{\text{stsp}} = D_{\text{KL}}\left(\hat{p}(\boldsymbol{x}) \,\|\, \hat{q}(\boldsymbol{x})\right) \approx \frac{1}{n} \sum_{i=1}^{n} \log \frac{\hat{p}(\boldsymbol{x}^{(i)})}{\hat{q}(\boldsymbol{x}^{(i)})} \,, \tag{12}$$

with $n$ Monte Carlo samples $\boldsymbol{x}^{(i)} \sim \hat{p}(\boldsymbol{x})$ drawn from the GMM approximating the empirical density. The variational approximation does not require sampling and has been shown to produce results similar to the Monte Carlo approximation (Koppe et al., 2019):

$$D_{\text{stsp}} = D_{\text{KL}}\left(\hat{p}(\boldsymbol{x}) \,\|\, \hat{q}(\boldsymbol{x})\right) \approx \frac{1}{T} \sum_{t=1}^{T} \log \frac{\sum_{j=1}^{T} e^{-D_{\text{KL}}(\mathcal{N}(\boldsymbol{x}; \boldsymbol{x}_t, \boldsymbol{\Sigma}) \,\|\, \mathcal{N}(\boldsymbol{x}; \boldsymbol{x}_j, \boldsymbol{\Sigma}))}}{\sum_{k=1}^{T} e^{-D_{\text{KL}}(\mathcal{N}(\boldsymbol{x}; \boldsymbol{x}_t, \boldsymbol{\Sigma}) \,\|\, \mathcal{N}(\boldsymbol{x}; \hat{\boldsymbol{x}}_k, \boldsymbol{\Sigma}))}} \,, \tag{13}$$

where arguments of the exponentials are KL divergencies between the individual Gaussians of the GMMs, for which we have closed-form analytical expressions.

One limitation of $D_{\text{stsp}}$ is its dependence on hyperparameters, the number of bins $m$ and the GMM covariance $\boldsymbol{\Sigma}$, both of which can critically influence the sensitivity of the measure. For example, if $m$ is chosen too small, the resulting spatial binning becomes overly coarse, preventing the accurate resolution of fine geometric details as important for the assessment of chaotic attractors (see Appx. of Brenner et al. (2022) for illustration). In contrast, a large $m$ produces mostly empty bins, may increase the impact of noise, and comes with high computational costs. In addition, arbitrary selection of these parameters will impede comparability of results across studies. To address such issues, here we suggest to adopt statistically principled ways of selecting these parameters based on ideas in optimal multivariate kernel density estimation (Scott, 2015), for instance bootstrap estimators (Faraway & Jhun, 1990; Sain, 2002) or estimators based on the sample (co-)variance (Scott, 2015; Silverman, 2018).

**Wasserstein distance**   Instead of the KL divergence, several authors based comparisons between trajectory distributions on the Wasserstein distance (Patel & Ott, 2023; Park et al., 2024; Göring et al., 2024). For instance, Göring et al. (2024) define a statistical error through the sliced Wasserstein-1 distance (SW$_1$; Bonneel et al. (2015)) between the occupation measures $\mu_{\boldsymbol{x},T}^{\Phi}$ of the ground-truth DS $\Phi$ and $\mu_{\boldsymbol{x},T}^{\Phi_R}$ of the DSR model $\Phi_R$:

$$\text{SW}_1(\mu_{\boldsymbol{x},T}^{\Phi}, \mu_{\boldsymbol{x},T}^{\Phi_R}) = \mathbb{E}_{\xi \sim \mathcal{U}(\mathbb{S}^{N-1})} \left[ W_1(g_{\boldsymbol{\xi}} \sharp \mu_{\boldsymbol{x},T}^{\Phi}, g_{\boldsymbol{\xi}} \sharp \mu_{\boldsymbol{x},T}^{\Phi_R}) \right], \tag{14}$$

where $\mathbb{S}^{N-1} := \{\boldsymbol{\xi} \in \mathbb{R}^N \mid \|\boldsymbol{\xi}\|_2^2 = 1\}$ is the unit hyper-sphere, $g_{\boldsymbol{\xi}} \sharp \mu$ denotes the pushforward of $\mu$, $g_{\boldsymbol{\xi}}(\boldsymbol{x}) = \boldsymbol{\xi}^T \boldsymbol{x}$ is the one-dimensional slice projection, and $W_1$ the Wasserstein-1 distance (Villani et al., 2009).

Eq. (14) is defined across distributions generated by single trajectories, typically to assess agreement between attractor geometries. The measure can be generalized, however, to compare the geometric behavior across entire subsets $U$ of state space $E$ by considering trajectories from a whole subset of initial conditions (Göring et al., 2024):

$$\mathcal{E}_{\text{stat}}^U(\Phi, \Phi_R) = \int_{U \subseteq E} \text{SW}_1(\mu_{\boldsymbol{x},T}^{\Phi}, \mu_{\boldsymbol{x},T}^{\Phi_R}) \, d\boldsymbol{x}, \tag{15}$$

where practically integration is performed across (initial conditions from) $U$. This allows for a more general comparison between vector field topologies, beyond the attracting limit set, but – unfortunately – is usually not computable for real-world datasets as neither $\Phi$ nor trajectories generated by $\Phi_t(\boldsymbol{x}_0)$ for all $\boldsymbol{x}_0 \in U$ are available.

Following Göring et al. (2024), Eq. (14) can be calculated using a Monte-Carlo approximation of the expectation

$$\text{SW}_1(\mu_{\boldsymbol{x},T}^{\Phi}, \ \mu_{\boldsymbol{x},T}^{\Phi_R}) \approx \frac{1}{L} \sum_{l=1}^{L} W_1(g_{\boldsymbol{\xi}^{(l)}} \sharp \mu_{\boldsymbol{x},T}^{\Phi}, \ g_{\boldsymbol{\xi}^{(l)}} \sharp \mu_{\boldsymbol{x},T}^{\Phi_R}), \tag{16}$$

where projection vectors $\boldsymbol{\xi}^{(l)} \sim \mathcal{U}(\mathbb{S}^{N-1})$ are drawn uniformly across the unit hypersphere embedded in $\mathbb{R}^N$. The sliced Wasserstein-1 distance is computed across trajectories (empirical distributions) of the ground-truth flow $\Phi$ and the reconstructed flow $\Phi_R$. Trajectories are drawn by evolving the respective system for $T$ time units from initial conditions $\boldsymbol{x} \in \mathbb{R}^N$. Between two 1D distributions (1D slices), the Wasserstein-1 distance can then efficiently be computed as

$$W_1(\mu, \nu) = \int_0^1 \left| F_\mu^{-1}(q) - F_\nu^{-1}(q) \right| dq, \tag{17}$$

where $F_\bullet^{-1}$ is the inverse cumulative distribution function. The integral is approximated by evaluating the CDFs at some finite resolution, e.g. $\Delta q = 10^{-3}$, and using, for instance, $L = 1000$ projections.

A major benefit of Wasserstein-based measures compared, e.g., to the Kullback-Leibler divergence, is the fact that differences in the support of the underlying probability measures are naturally resolved. Because it is an optimal transport metric, it is fundamentally less sensitive to shifts in the precise attractor location in state space or other rigid transformations than measures, like the KL divergence, that depend on the same binning grid and thus also on hyperparameters like the bin size or GMM covariance $\boldsymbol{\Sigma}$.

Various other distributional measures, e.g. based on the generalized correlation integral, have been defined in the literature (Fumagalli et al., 2025).

### B.2. Characterizing Agreement in Long-Term Temporal Structure

In a famous paper, Wood (2010) discussed the problem that for chaotic systems likelihood landscapes will be rough, often fractal, and thus straightforward maximum likelihood or MSE estimation of model parameters will fail (see also Voss et al. (2004) for a related discussion). This is related to the high sensitivity to precise initial conditions and the exponential divergence of trajectories in chaotic DS that we had illustrated in Fig. 4. Wood (2010) therefore suggested to use summary statistics, like the coefficients of the auto-correlation function, in a surrogate likelihood, an approach followed in, e.g., Brenner et al. (2024b) to assess the agreement in true and reconstructed long-term temporal structure.

Alternatively, one may compare the power spectra between true and reconstructed DS (recall that for stationary processes the power spectrum and auto-correlation are strictly related through the Wiener-Khinchin theorem; Papoulis & Pillai (2002)). One distance measure that has often been used in this context is the *Hellinger distance* (Mikhaeil et al., 2022; Hess et al., 2023; Pals et al., 2024). Specifically, given normalized power spectra $f_i(\omega)$ and $g_i(\omega)$ of the $i$-th dynamical variable of the observed and generated time series $\boldsymbol{X}$ and $\hat{\boldsymbol{X}}$, respectively, where $\int_{-\infty}^{\infty} f_i(\omega) d\omega = 1$ and $\int_{-\infty}^{\infty} g_i(\omega) d\omega = 1$, the Hellinger distance is given by

$$H(f_i(\omega), g_i(\omega)) = \sqrt{1 - \int_{-\infty}^{\infty} \sqrt{f_i(\omega) g_i(\omega)} \, d\omega} \tag{18}$$

The Hellinger distance is a scalar $0 \leq H \leq 1$ which measures the discrepancy in frequency content, where 0 indicates perfect agreement.

In practice, power spectra are computed using the Fast Fourier Transform (Cooley & Tukey, 1965), yielding $\hat{\boldsymbol{f}}_i = |\mathcal{F}x_{i,1:T}|^2$ and $\hat{\boldsymbol{g}}_i = |\mathcal{F}\hat{x}_{i,1:T}|^2$, with vectors $\hat{\boldsymbol{f}}_i$ and $\hat{\boldsymbol{g}}_i$ discrete power spectra of ground truth TS $x_{i,1:T}$ and model generated TS $\hat{x}_{i,1:T}$. Since raw power spectra tend to be quite noisy, some smoothing is usually applied using a Gaussian filter with standard deviation $\sigma_s$. The resulting spectra are normalized to fulfill $\sum_\omega \hat{f}_{i,\omega} = 1$ and $\sum_\omega \hat{g}_{i,\omega} = 1$. $H$ (18) is then computed as

$$H(\hat{\boldsymbol{f}}_i, \hat{\boldsymbol{g}}_i) = \frac{1}{\sqrt{2}} \left\| \sqrt{\hat{\boldsymbol{f}}_i} - \sqrt{\hat{\boldsymbol{g}}_i} \right\|_2, \tag{19}$$

where the square root is applied elementwise. The final measure $D_H$ is obtained by averaging $H$ across all dimensions:

$$D_H = \frac{1}{N} \sum_{i=1}^{N} H(\hat{\boldsymbol{f}}_i, \hat{\boldsymbol{g}}_i). \tag{20}$$

The smoothing $\sigma_s$ is commonly treated as a hyperparameter, set individually for each dataset at hand. Unfortunately, as for $D_{\text{stsp}}$, the performance of $D_H$ depends on the choice of $\sigma_s$, which is in particular a problem if different studies are to be compared. Choosing it too large leads to oversmoothing of the spectra such that important details might be lost, while without any smoothing irrelevant and minuscule shifts in frequency peaks will be overemphasized. Here we therefore suggest some potential improvements: 1) use of statistically optimal smoothing estimators for power spectra, for instance based on plug-in unbiased risk estimation (Lee, 2001) or minimizing an MSE estimate over Thomson's multitaper spectrum (Haley & Anitescu, 2017); 2) calculating the Hellinger distance between *logarithmized* power spectra to level the importance of low- and high-frequency components; 3) replacing the Hellinger distance with the Wasserstein distance (e.g. $W_1$, see Eq. (17)), which – as indicated above – is more robust to shifts in the curves as long as their overall shapes match well. A caveat here is that the Wasserstein distance is not bounded from above, and hence more difficult to interpret than a normalized quantity like the Hellinger distance.

### B.3. Valid Prediction Time

A common measure to assess *short-term* forecasts, especially when dealing with chaotic dynamics, is the Valid Prediction Time (VPT, Vlachas et al. (2020); Platt et al. (2022); Fumagalli et al. (2025)), which measures the time it takes for data and model forecasts to diverge as determined by some arbitrary error criterion $\epsilon$. Given a pointwise forecast error $\mathcal{E}(\boldsymbol{x}_t, \hat{\boldsymbol{x}}_t)$ with data $\boldsymbol{x}_t$ and corresponding model forecasts $\hat{\boldsymbol{x}}_t$, the VPT is defined by

$$\text{VPT} = \arg\max_t \{ t \mid \mathcal{E}(\boldsymbol{x}_t, \hat{\boldsymbol{x}}_t) < \epsilon \}. \tag{21}$$

The VPT is usually reported in units of the underlying DS' Lyapunov time $\tau_{Lyap} := 1 / \lambda_{max}$, if (an estimate of) it is available. Common choices for the forecast error are NRMSE or sMAPE combined with an arbitrarily chosen $\epsilon \in [0.3, 0.5]$ (Vlachas et al., 2020; Platt et al., 2022; Gilpin, 2023).

For chaotic systems, a VPT approaching one Lyapunov time is a desirable outcome for any forecasting model. Sometimes cases with VPTs of multiple Lyapunov times (up to $> 10 \cdot \tau_{Lyap}$) were reported, but these are always very special scenarios where models were trained and evaluated on *completely noise-free* benchmark systems (Pathak et al., 2018; Platt et al., 2022; Gilpin, 2023; Vlachas & Koumoutsakos, 2024; Hurley & Shaheen, 2025), hence not representative of any empirical setting. Since the VPT is highly dependent on the choice of $\epsilon$ and thus an author's subjective judgment of what constitutes a 'valid' forecast, evaluations based on the VPT should be treated with great care!

## C. Training Methods for DSR

The biggest difference between training methods for DSR vs. those for 'standard' TSF models is the emphasis on capturing the long-term dynamics of the observed DS. It is by now common knowledge that performing multi-step ahead prediction, or *unrolling* autoregressive models such as RNNs and Neural Operators into the future, improves long-term forecasts (Lusch et al., 2018; Platt et al., 2022; Mikhaeil et al., 2022; Brenner et al., 2022; Hess et al., 2023; Teutsch & Mäder, 2022; Vlachas & Koumoutsakos, 2024; Schiff et al., 2024; Park et al., 2024; Zhou & Cheng, 2025). Most autoregressive models are

trained using backpropagation through time (BPTT, Werbos (1990)) to compute gradients of a loss function, which sums up contributions of individual time steps, w.r.t. parameters of the model, followed by a gradient descent update in parameter space. Assume the model map is given by $z_t = F_{\boldsymbol{\theta}}(z_{t-1}, s_t)$ with states $z_t$ and optional external inputs $s_t$. Denote the loss as a function of parameters by $\mathcal{L} = \sum_t \mathcal{L}_t(\boldsymbol{\theta})$, then BPTT involves calculating gradients of parameters $\theta \in \boldsymbol{\theta}$ w.r.t. the loss function $\mathcal{L}$:

$$\frac{\partial \mathcal{L}}{\partial \theta} = \sum_t \frac{\partial \mathcal{L}_t}{\partial \theta} \quad \text{with} \quad \frac{\partial \mathcal{L}_t}{\partial \theta} = \sum_{r=1}^{t} \frac{\partial \mathcal{L}_t}{\partial z_t} \frac{\partial z_t}{\partial z_r} \frac{\partial^+ z_r}{\partial \theta}, \tag{22}$$

where $\partial^+$ denotes the immediate derivative (Pascanu et al., 2013). The center term in the loss gradient for each individual time step can be further unwrapped by recursive application of the chain rule into a product series of Jacobians as

$$\frac{\partial z_t}{\partial z_r} = \frac{\partial z_t}{\partial z_{t-1}} \frac{\partial z_{t-1}}{\partial z_{t-2}} \cdots \frac{\partial z_{r+1}}{\partial z_r} = \prod_{k=0}^{t-r-1} \frac{\partial z_{t-k}}{\partial z_{t-k-1}} = \prod_{k=0}^{t-r-1} \boldsymbol{J}_{t-k}, \tag{23}$$

where $\boldsymbol{J}_t = \frac{\partial z_t}{\partial z_{t-1}}$ is the model Jacobian at time step $t$. This Jacobian product during the backward pass in training exposes the famous exploding and/or vanishing gradient problem (Bengio et al., 1994; Pascanu et al., 2013): If in the geometric mean, maximum singular values of the individual Jacobian terms $\sigma_{max}(\boldsymbol{J}_i)$ are larger than 1, the Jacobian product will diverge and hence the respective gradient $\partial \mathcal{L}_t / \partial \theta$. Mikhaeil et al. (2022) showed that gradient explosion is inevitable when the model is trained to reproduce chaotic dynamics. Recall (cf. Eq. 1) that the maximum Lyapunov exponent $\lambda_{max}$ for a discrete map is defined as

$$\lambda_{max} := \lim_{T \to \infty} \frac{1}{T} \log \sigma_{max} \left( \prod_{t=0}^{T-1} \boldsymbol{J}_{T-t} \right), \tag{24}$$

where $T$ is the orbit length. Comparing Eq. (24) and Eq. (23), it is evident that the same Jacobian product series occurs both in the loss gradients as well as in the definition of the maximum Lyapunov exponent. Since for reconstructing chaotic dynamics we must have $\lambda_{max} > 0$, this product series will inevitably diverge for $T \to \infty$. This is therefore a severe issue for training on real world data, since almost all complex systems, from the physical to the human world, are chaotic (Sivakumar, 2004; Durstewitz & Gabriel, 2007; Govindan et al., 1998; Turchin & Taylor, 1992; Field et al., 1972).

Since long roll-outs in training are needed to properly capture the DS' long-term statistics, major training methods for DSR aim to mitigate the problem of chaos-induced exploding gradients through mainly control-theoretic techniques (Abarbanel, 2013). Alternatively, long-term statistics could be built directly into the loss criterion, another line of training methods for DSR discussed below.

### C.1. Control-Theoretic Training Through Variants of Teacher Forcing

Teacher forcing (TF) methods date back to work by Pineda (1988); Williams & Zipser (1989); Pearlmutter (1989); Doya (1992), who suggested that autoregressive models benefit from guidance by data to better learn the task at hand. Here we discuss two TF variants specifically designed to improve DSR.

**Sparse Teacher Forcing (STF)**   STF seeks to mitigate chaos-induced exploding gradients by periodically replacing the model-generated states $z$ by data-inferred states $\hat{z}$ (Mikhaeil et al., 2022):

$$z_{t+1} = \begin{cases} F(\hat{z}_t) & \text{if } t \in \mathcal{T} = \{t \mid t = n\tau, \ n \in \mathbb{N}_0\} \\ F(z_t) & \text{else} \end{cases} \tag{25}$$

where $\tau$ is the forcing interval and $\mathcal{T}$ the set of forcing times. The forcing signal is produced by either inversion of the observation/decoder model $\hat{z} = g^{-1}(x_t)$ (Mikhaeil et al., 2022; Hess et al., 2023), or by an encoder model $E(x_t)$ in a (variational) auto-encoder setting (Brenner et al., 2024b). For example, if $g$ is given by a linear mapping $\boldsymbol{B} z_t$ which maps from a higher-dimensional latent space to the observations, one can estimate $\hat{z}_t = \boldsymbol{B}^+ x_t$, where $\boldsymbol{B}^+$ denotes the Moore-Penrose pseudo-inverse. When the model is trained *directly* on data, the forcing signals may be given straight by $\hat{z}_t = x_t$. STF has two implications: first, it realigns model-generated trajectories with trajectories of the underlying DS, pulling them 'back on track'; second, it truncates gradients by virtue of the forcing with data-inferred values as $\boldsymbol{J}_{n\tau+1} = \frac{\partial F(\hat{z}_t)}{\partial z_t} = 0$. In this approach, $\tau$ is a crucial hyperparameter that needs to be chosen in an optimal way: If it is too short, the system's long-term properties won't be captured, while if it is too large, gradients will diverge. Based

on the observation that the rate of gradient divergence is determined by $\lambda_{\max}$, Mikhaeil et al. (2022) show that using the *predictability time* $\tau_{\text{pred}} = \frac{\ln 2}{\lambda_{max}}$ provides a good heuristic for most simulated and empirical systems.

Brenner et al. (2022; 2024a;b); Hemmer & Durstewitz (2025) use a variant of STF where the state of the model is only partially forced (Tsung & Cottrell, 1994). Let $z \in \mathbb{R}^M$, then partially forcing the first $k < M$ units results in $z_{t+1} = F([\hat{z}_1, \ldots, \hat{z}_k, z_{k+1}, \ldots z_M]^\top)$. This leaves a $M - k$-dimensional subspace through which gradients can flow freely.

**Generalized Teacher Forcing (GTF)**  Instead of replacing the entire state at specific intervals $\tau$ as in STF, GTF replaces states at each time step by a linear interpolation between model-iterated state and forcing signal, weighted by a forcing strength $0 \leq \alpha \leq 1$ (Doya, 1992; Hess et al., 2023):

$$
\begin{aligned}
z_{t+1} &= F(\ (1-\alpha)z_t + \alpha\hat{z}_t\ ) \\
&= F(\tilde{z}_t),
\end{aligned}
\tag{26}
$$

where $\tilde{z}$ denotes the forced state. By virtue of this forcing, the model Jacobian decomposes during training as

$$
J_{t+1} = \frac{\partial F(\tilde{z}_t)}{\partial z_t} = (1-\alpha)\frac{\partial F(\tilde{z}_t)}{\partial \tilde{z}_t}\ .
\tag{27}
$$

Assuming $\frac{\partial F(\tilde{z}_t)}{\partial \tilde{z}_t} \approx \frac{\partial F(z_t)}{\partial z_t}$, choosing $\alpha = 1 - \frac{1}{\sigma_{max}}$ bounds the norm of the Jacobian product in Eq. (23) from above by 1 (Hess et al., 2023), where $\sigma_{max}$ is the maximum singular value across all Jacobians evaluated along the sequence. While fully mitigating exploding gradients, too high settings of $\alpha$ can exacerbate vanishing gradients. Hence, best results are often achieved by treating $\alpha$ as a hyperparameter (Hess et al., 2023; Volkmann et al., 2024), or by curriculum learning strategies that adaptively estimate an optimal $\alpha$ using computationally efficient proxies to Jacobian products evaluated on the current training batch (Hess et al., 2023).

In contrast, conventional, likelihood-based TF methods (Bengio et al., 2015; Goodfellow et al., 2016), where instead of *replacing* the state teacher signals are fed through the input layer of the RNN, have mainly tried to address the exposure bias problem (Ranzato et al., 2016). Exposure bias refers to the discrepancy between training and testing of autoregressive models, where models are trained to predict the next state based on the previous ground truth (TF) but are tasked to generate predictions in the absence of any forcing signals during test time. To address this issue, mostly curriculum learning strategies have been developed (Bengio et al., 2015; Lamb et al., 2016; Teutsch & Mäder, 2022) and applied in DSR to improve long-term forecasts (Vlachas & Koumoutsakos, 2024). However, these methods do not address chaos-induced exploding gradients (Mikhaeil et al., 2022) and hence often fall short in solving the long-term issues for noisy real-world data (Mikhaeil et al., 2022; Brenner et al., 2022; Hess et al., 2023).

Although STF/GTF suffer similarly from exposure bias, their respective forcing parameters $\tau$ and $\alpha$ can control its severity and it comes down to a min-max game: Ideally, forcing should be as minimal as possible (high $\tau$, small $\alpha$) to reduce exposure bias, yet large enough (small $\tau$, high $\alpha$) to avoid diverging trajectories and exploding gradients (Mikhaeil et al., 2022; Hess et al., 2023). Techniques like GTF also lead to smooth, often unimodal if optimally adjusted, loss landscapes (Voss et al., 2004; Abarbanel, 2013; 2022; Hess et al., 2023; Brenner et al., 2024b).

### C.2. Multiple Shooting

An older method in the DS literature, originally introduced to solve boundary value problems and similar in spirit to STF, has been termed 'multiple shooting' (MS; Bock & Plitt (1984); Voss et al. (2004)). MS divides time into multiple sub-intervals, for each of which a separate initial value problem is to be solved. Continuity across interval boundaries is then enforced through a regularization term in the loss function (Voss et al., 2004). Thus, instead of state estimation through observation/decoder model inversion or through an encoder network as in STF/GTF, initial conditions for each sub-interval are trainable parameters which are jointly optimized with the model dynamics.

Given an observed TS $X = \{x_t\}$ of length $T$, MS works by first dividing the TS into $n = \lfloor \frac{T}{L} \rfloor$ subsequences of (roughly) equal length $L$, such that the training data is given as a set of subsequences $\{X^i\}_{i=1}^n$. For each subsequence $X^i$, a separate initial condition $\mu_0^i \in \mathbb{R}^M$ is learned, where $M$ is the dimensionality of the RNN. During training, the model freely generates trajectories of length $L$ from initial conditions $\mu_0^i$ for $X^i$ sampled from the training set, and a continuity regularization term

is added to the (usually) MSE loss:

$$\mathcal{L} = \mathcal{L}_{MSE} + \lambda_{MS} \sum_{i=1}^{n-1} \|F_{\boldsymbol{\theta}}(\boldsymbol{z}_L^i) - \boldsymbol{\mu}_0^{i+1}\|_2^2 \,, \tag{28}$$

where $F_{\boldsymbol{\theta}}$ is the DSR model with latent states $\boldsymbol{z}_t$, and $\lambda_{MS}$ is a regularization parameter (note that, as the model evolves freely for $L$ time steps, $\boldsymbol{z}_L^i = F^L(\boldsymbol{\mu}_0^i)$).

The sequence length $L$ plays a similar role here as the forcing interval $\tau$ in STF (Brenner et al., 2024b) and determines the time after which gradients are truncated. However, MS scales less favorably than STF, since the number of learnable initial conditions $\boldsymbol{\mu}_0$ grows linearly with dataset size, and there is no inbuilt mechanism to start the model from *arbitrary* initial conditions.

### C.3. Specific DSR Loss Functions

Another idea is to replace or augment the standard MSE loss by a loss function that directly reflects the long-term objectives of DSR training, for instance by incorporating dynamical invariants into the loss through regularization. Along these lines, Platt et al. (2022; 2023) employ a 'macro-scale' loss function which measures the discrepancy between data- and model-derived dynamical invariant(s) $\boldsymbol{C}_{\text{data}}$ and $\boldsymbol{C}_{\text{model}}$, respectively:

$$\mathcal{L}_{\text{macro}} = \epsilon_1 \|\boldsymbol{C}_{\text{data}} - \boldsymbol{C}_{\text{model}}\|^2 + \epsilon_2 \sum_{k=1}^{n} \sum_{t=t_i^{(k)}}^{t_f^{(k)}} \|\hat{\boldsymbol{x}}_t^{(k)} - \boldsymbol{x}_t^{(k)}\|^2 \exp\left(-\frac{t - t_i^{(k)}}{t_f^{(k)} - t_i^{(k)}}\right), \quad t \in \mathbb{Z} \,, \tag{29}$$

where $n$ is the number of individual trajectory bits $\{\boldsymbol{x}_{t_i^{(k)}:t_f^{(k)}}\}_{k=1}^n$ drawn i.i.d. from the total training data pool, $\epsilon_i$ are regularization parameters, $\hat{\boldsymbol{x}}_t$ are freely generated forecasts of the model (a reservoir computer in their case) and $\boldsymbol{x}_t$ the respective (observed) training data. $t_i$ and $t_f$ are initial and final forecast times. Thus, the first term punishes deviations in dynamical invariants, while the second term is an MSE loss with exponentially decaying weighting (with forecast horizon length) which accounts for trajectory divergence when modeling chaotic dynamics. For dynamical invariants $\boldsymbol{C}$, Platt et al. (2023) suggest the use of quantities that can in principle be *estimated* from empirical data, such as the maximum Lyapunov exponent or the fractal dimension through the correlation dimension (see Appx. B.1.1). The macro-loss is then minimized in conjunction with a vanilla MSE loss using a two-step optimization process, where Eq. (29) is optimized through Bayesian techniques or evolutionary methods (Platt et al., 2022; 2023). Platt et al. (2023) show that including just a single Lyapunov exponent in the macro-loss can significantly improve long-term forecasts of reservoir computers.

Jiang et al. (2023) designed a more general loss function for neural operators that aims to preserve invariant measures (probability measures invariant under the flow) for accurate long-term forecasts using optimal transport and contrastive learning. If prior knowledge in form of $n_s$ summary statistics $\boldsymbol{s}(\boldsymbol{x}) = [s^{(1)}(\boldsymbol{x}), \ldots s^{(n_s)}(\boldsymbol{x})]$, $\boldsymbol{s}_i := \boldsymbol{s}(\boldsymbol{x}_i)$, that can be computed from trajectories is available, an optimal transport loss may be defined as

$$\mathcal{L}_{OT}(\boldsymbol{S}, \hat{\boldsymbol{S}}) = \frac{1}{2}\left(W^{\gamma}(\boldsymbol{S}, \hat{\boldsymbol{S}})^2 - \frac{W^{\gamma}(\boldsymbol{S}, \boldsymbol{S})^2 + W^{\gamma}(\hat{\boldsymbol{S}}, \hat{\boldsymbol{S}})^2}{2}\right) \,, \tag{30}$$

where $W^{\gamma}$ denotes the entropy-regularized Wasserstein distance with regularization parameter $\gamma$, and $\boldsymbol{S} = \{\boldsymbol{s}_i\}_{i=1}^K$, $\hat{\boldsymbol{S}} = \{\hat{\boldsymbol{s}}_i\}_{i=1}^K$ are summary statistics obtained from data and model generated distributions, respectively. When no such prior knowledge is available, Jiang et al. (2023) propose to instead add a contrastive learning loss to the total loss that encourages similarity between features predicted from data and model trajectories through a separate encoder network $f_{\psi}$:

$$\mathcal{L}_{CL}(\boldsymbol{x}_{t:t+L}, \hat{\boldsymbol{x}}_{t:t+L}; f_{\psi}) = \sum_i D_C\left(f_{\psi}^i(\boldsymbol{x}_{t:t+L}), f_{\psi}^i(\hat{\boldsymbol{x}}_{t:t+L})\right) \,, \tag{31}$$

where $\boldsymbol{x}_{t:t+L}, \hat{\boldsymbol{x}}_{t:t+L}$ are subsequences of data and model trajectories, $D_C$ is the cosine distance and $f_{\psi}^i$ denotes the output of the $i$-th layer of the encoder network. The encoder network is pretrained using contrastive learning on a larger pool of trajectories, where sequences $\boldsymbol{x}_{t:t+L}$ from the same trajectory are treated as positive pairs, while sequences from different trajectories are treated as negative pairs. Hence, the encoder is trained to produce features that distinguish trajectories that

follow different invariant measures. These loss terms, Eq. (30) & (31), targeting long-term dynamics are then paired with a straightforward relative root MSE (RMSE) error between sequences

$$\mathcal{L}_{\text{RMSE}}\left(\boldsymbol{x}_{t:t+L}, \hat{\boldsymbol{x}}_{t:t+L}\right) = \frac{1}{L}\sum_{i=t}^{L}\frac{\|\boldsymbol{x}_i - \hat{\boldsymbol{x}}_i\|_2}{\|\boldsymbol{x}_i\|_2}. \tag{32}$$

When prior knowledge is available the optimal transport loss is used, $\mathcal{L} = \mathcal{L}_{\text{RMSE}} + \lambda\mathcal{L}_{OT}$, where $\lambda$ is a regularization parameter. Otherwise, informative features are learned and matched using the contrastive learning loss, i.e. $\mathcal{L} = \mathcal{L}_{\text{RMSE}} + \lambda\mathcal{L}_{CL}$.

Relatedly, Schiff et al. (2024) introduce the Dynamics Stable Learning by Invariant Measure (DySLIM) objective

$$\mathcal{L}_\lambda^{\text{D}}(\boldsymbol{\theta}) = \mathcal{L}_{\text{obj}}(\boldsymbol{\theta}) + \lambda_1\widehat{\text{D}}\left(\mu^*, (F_{\boldsymbol{\theta}}^n)_\sharp\mu^*\right) + \lambda_2\widehat{\text{D}}\left((F^n)_\sharp\mu^*, (F_{\boldsymbol{\theta}}^n)_\sharp\mu^*\right), \tag{33}$$

where $\lambda_i$ are regularization parameters, $\mathcal{L}_{\text{obj}}(\boldsymbol{\theta})$ is an ($n$-step) autoregressive roll-out error (e.g. MSE), $F_{\boldsymbol{\theta}}$ is the model map, $F$ is the ground truth DS, $\mu^*$ is the invariant measure given by the data, $(F_{\boldsymbol{\theta}}^n)_\sharp\mu^*$ denotes the pushforward of measure $\mu^*$ under the $n$-times iterated map, $\boldsymbol{x}_{t+n} = F_{\boldsymbol{\theta}}^n(\boldsymbol{x}_t)$, and $D$ is a distance measure between distributions, such as the Maximum Mean Discrepancy (Schiff et al., 2024). As in Jiang et al. (2023), this loss function encourages the model to preserve the invariant measure by treating data and model generated states as samples from the respective invariant measures and minimizing their distance.

As observed by many other groups (Platt et al., 2021; Mikhaeil et al., 2022), Park et al. (2024) also demonstrate that using mere 1-step-ahead predictions to train NN based DSR models is insufficient for capturing the long-term statistics of DS. Their strategy is to augment the loss by a Jacobian term that aims to match local derivatives along trajectories:

$$\mathcal{L} = \frac{1}{T-1}\sum_{t=1}^{T-1}\|F_{\boldsymbol{\theta}}(\boldsymbol{x}_t) - F(\boldsymbol{x}_t)\|^2 + \lambda\|\boldsymbol{J}_{\boldsymbol{\theta},t} - \boldsymbol{J}_t\|^2, \tag{34}$$

where $\boldsymbol{J}_t$ and $\boldsymbol{J}_{\boldsymbol{\theta},t}$ are the Jacobians of the ground-truth DS $F$ underlying the data and of the DSR model $F_{\boldsymbol{\theta}}$, respectively. Park et al. (2024) show that this enables DSR models to learn the invariant measure of the true DS without the need to unroll dynamics over several steps during training. However, an obvious caveat here is that full temporally resolved Jacobians are not available (or extremely hard to reliably estimate) for real empirical data.

## D. Simulation Models and Datasets

### D.1. Lorenz-63 Model of Atmospheric Convection

The *Lorenz-63 system* (Lorenz, 1963) is a continuous-time DS that was originally proposed as a minimal, low-order Galerkin model of atmospheric convection. It is given by nonlinear ODEs for three state variables governing the temporal evolution of convective overturning, horizontal temperature contrast, and vertical temperature distortion as

$$\begin{aligned}\frac{dx_1}{dt} &= \sigma(x_2 - x_1),\\ \frac{dx_2}{dt} &= x_1(\rho - x_3) - x_2,\\ \frac{dx_3}{dt} &= x_1 x_2 - \beta x_3,\end{aligned}$$

We chose the standard parameters $\sigma = 10$, $\rho = 28$, and $\beta = \frac{8}{3}$ that place the system into a chaotic regime.

### D.2. Spiking Neuron Model

We used a simplified Hodgkin-Huxley-type $3d$ biophysical neuron model describing the temporal evolution of the membrane potential $V$ and two gating variables $n$ and $h$ which control the opening of fast and slow potassium currents, respectively (Durstewitz, 2009):

$$\dot{V} = \frac{1}{C}\Big[I - g_L(V - E_L) - g_{\mathrm{Na}}\, m_\infty(V)\,(V - E_{\mathrm{Na}}) - g_K\, n\,(V - E_K) \tag{35}$$
$$- g_M\, h\,(V - E_K) - g_{\mathrm{NMDA}}\, s_\infty(V)\,(V - E_{NMDA})\Big],$$

$$\dot{n} = \frac{n_\infty(V) - n}{\tau_n}, \tag{36}$$
$$\dot{h} = \frac{h_\infty(V) - h}{\tau_h}, \tag{37}$$

with

$$m_\infty(V) = \frac{1}{1 + \exp\big((V_{h,\mathrm{Na}} - V)/k_{\mathrm{Na}}\big)}, \tag{38}$$

$$n_\infty(V) = \frac{1}{1 + \exp\big((V_{h,K} - V)/k_K\big)}, \tag{39}$$

$$h_\infty(V) = \frac{1}{1 + \exp\big((V_{h,M} - V)/k_M\big)}, \tag{40}$$

$$s_\infty(V) = \frac{1}{1 + 0.33\,\exp\big(-0.0625\,V\big)}. \tag{41}$$

Chosen model parameters are provided in the Table below.

For creating Figs. 1 & 2a, the $h$ variable was fixed to certain values ($h = 0.05$ in Fig. 1), reducing the model to 2 dimensions, while Fig. 2b shows the dynamics of the full $3d$ model with the parameter $g_{NMDA}$ linearly increasing across time (making the DS non-autonomous).

Neuron model parameter settings

| $I$ | $C$ | $g_L$ | $E_L$ | $g_{\mathrm{Na}}$ | $E_{\mathrm{Na}}$ | $V_{h,\mathrm{Na}}$ | $k_{\mathrm{Na}}$ | $g_K$ | $E_K$ | $V_{h,K}$ | $k_K$ | $\tau_n$ | $g_M$ | $V_{h,M}$ | $k_M$ | $\tau_h$ | $g_{\mathrm{NMDA}}$ | $E_{\mathrm{NMDA}}$ |
|---|---|---|---|---|---|---|---|---|---|---|---|---|---|---|---|---|---|---|
| 0 | 6 | 8 | $-80$ | 20 | 60 | $-20$ | 15 | 10 | $-90$ | $-25$ | 5 | 1 | 25 | $-15$ | 5 | 200 | 10.2 | 0 |

### D.3. Real-World Datasets

Here we give a brief description of the real-world datasets used to compare DSR and TS models. The *traffic data* consist of hourly measurements of the number of vehicles passing through road junctions (https://www.kaggle.com/datasets/fedesoriano/traffic-prediction-dataset/data). The *cloud data* are publicly available from Huawei Cloud (https://github.com/sir-lab/data-release) and contain records of function requests from Huawei's serverless cloud platform. The *weather data* comprise daily soil temperature and air pressure readings from a city in Germany provided by the German Weather Service, and can be obtained from https://www.dwd.de/EN/ourservices/cdc/cdc_ueberblick-klimadaten_en.html. The *functional magnetic resonance imaging (fMRI) data* comes from human participants performing various cognitive tasks and is publicly available on GitHub (Kramer et al., 2022). Kramer et al. (2022) report a positive maximum Lyapunov exponent for models reconstructed from these time series, suggesting that the underlying dynamics is chaotic (see also (Volkmann et al., 2024)). The *ETTh1 dataset* belongs to the Electricity Transformer Temperature (ETT) benchmark, a standard benchmark for TS forecasting. It comprises hourly measurements from a power transformer station (Zhou et al., 2021) and is available at https://github.com/zhouhaoyi/ETDataset. The *electroencephalogram (EEG) data* are from a study by Schalk et al. (2000) and contain recordings from human subjects engaged in various motor and motor imagery tasks. In line with Brenner et al. (2022), the EEG signals were smoothed using a Hann window of length 15. The non-stationary *Air Passengers* dataset, consisting of airline passenger counts, is available at https://www.kaggle.com/datasets/chirag19/air-passengers.

# E. Methodological Details on DSR and TS Models

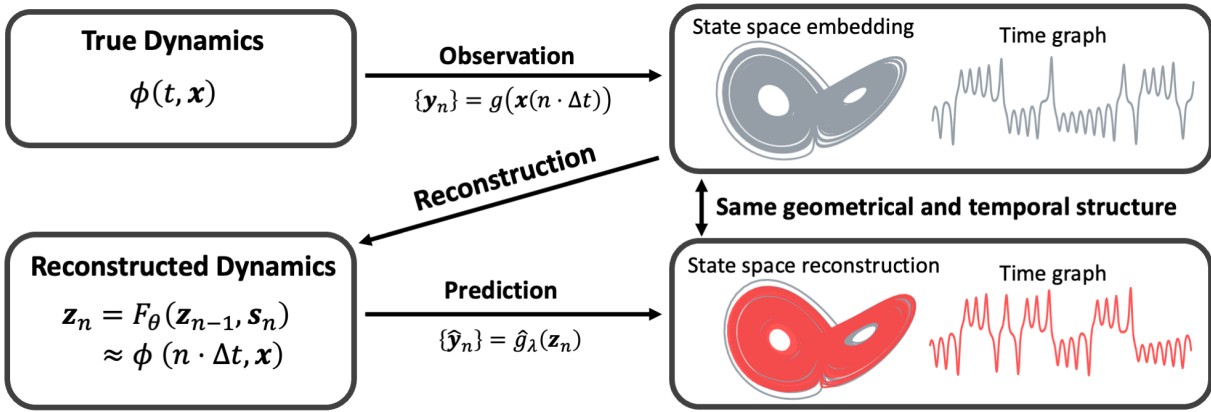

*Figure 7.* Illustration of DSR process. "$\approx$" is meant here in the sense of topological conjugacy (cf. Def. A.4) and (potentially) geometric & temporal invariants (like the Lyapunov spectrum or fractal dimension).

Fig. 7 provides the general layout of DSR. For all models described in this section, implementations from the original code repositories as indicated below in the respective subsections were used for the comparisons shown in Fig. 5 and Tables 1 & 2. Optimal hyperparameters were determined for all models by following the authors' original methodology, and by conducting hyperparameter tuning through either grid search or Bayesian optimization.

## E.1. Custom Trained DSR Models

Most DSR models are more or less standard architectures (like RNNs or Neural ODEs) with, however, often an emphasis on either physical (Brunton et al., 2016; Champion et al., 2019; Raissi et al., 2019) or mathematical (Linderman et al., 2017; Lusch et al., 2018; Brenner et al., 2022; 2024a; Bolager et al., 2025) interpretability, since these models are commonly employed with scientific or medical applications in mind. Comparatively easy accessibility of mathematical (topological/geometric) properties using DS theoretical tools is thus an important criterion in model design. Apart from this, as discussed in the main text, the more important aspect are the training techniques used to achieve proper DSRs (cf. Appx. C). Below we therefore only review the models used for the comparisons in Fig. 5 and Table 1. Generally, most DSR models (like all RNNs essentially) aim to learn an approximation to the flow operator $\Phi_{\Delta t}(\boldsymbol{x})$, while some continuous-time models (like Neural ODEs or SINDy) aim to approximate the vector field $f(\boldsymbol{x})$.

**AL-RNN** The Almost-Linear RNN (AL-RNN; Brenner et al. (2024a)) aims, as the name suggests, to reduce the number of nonlinearities to a bare minimum in order to achieve maximal mathematical tractability. Nevertheless it has been shown to be on par with other SOTA DSR models when properly trained by techniques as reviewed in Appx. C, such as STF and GTF. It also induces a natural symbolic encoding (Lind & Marcus, 2021) that preserves key topological properties of the underlying DS. The AL-RNN combines linear units and ReLUs according to

$$\boldsymbol{z}_t = \boldsymbol{A}\,\boldsymbol{z}_{t-1} + \boldsymbol{W}\,\Phi^*(\boldsymbol{z}_{t-1}) + \boldsymbol{h}, \tag{42}$$

where $\boldsymbol{z}_t$ is the model's latent state, $\boldsymbol{A} \in \mathbb{R}^{M \times M}$ is a purely linear (diagonal) term, $\boldsymbol{W} \in \mathbb{R}^{M \times M}$, and bias vector $\boldsymbol{h} \in \mathbb{R}^M$. The function $\Phi^* : \mathbb{R}^M \to \mathbb{R}^M$ applies a ReLU nonlinearity only to a subset of $P$ neurons:

$$\Phi^*(\boldsymbol{z}) = \big[z_1, \ldots, z_{M-P}, \max(0, z_{M-P+1}), \ldots, \max(0, z_M)\big]^\top. \tag{43}$$

This divides the state space into $2^P \ll 2^M$ distinct regions with linear dynamics, easing model analysis. For evaluation, we used the code provided by https://github.com/DurstewitzLab/ALRNN-DSR.

**shPLRNN** The shallow PLRNN (shPLRNN; Hess et al. (2023)) is another SOTA DSR model with a piecewise-linear structure to allow for deeper mathematical analysis (Eisenmann et al., 2023). It is based on a one hidden layer RNN with ReLU activation, mapping the latent states into a higher dimensional hidden space, thus enhancing expressivity, and is defined by

$$\boldsymbol{z}_t = \boldsymbol{A}\,\boldsymbol{z}_{t-1} + \boldsymbol{W}_1\,\phi\big(\boldsymbol{W}_2\,\boldsymbol{z}_{t-1} + \boldsymbol{h}_2\big) + \boldsymbol{h}_1, \tag{44}$$

where $\boldsymbol{A} \in \mathbb{R}^{M \times M}$ is a diagonal matrix, $\boldsymbol{W}_1 \in \mathbb{R}^{M \times H}$ and $\boldsymbol{W}_2 \in \mathbb{R}^{H \times M}$ are weight matrices, with usually $H > M$, $\boldsymbol{h}_2 \in \mathbb{R}^H$ and $\boldsymbol{h}_1 \in \mathbb{R}^M$ are bias vectors, and $\phi(\cdot)$ is the ReLU activation. The model is trained by GTF (see Appx. C). For evaluation, we used the implementation given by `https://github.com/DurstewitzLab/GTF-shPLRNN`.

**Reservoir Computing**   Reservoir Computers (RC) are a type of RNN with a huge pool of recurrently coupled nonlinear units (the 'reservoir') and a commonly linear readout layer that maps activity in the reservoir to a much smaller set of outputs (Jaeger & Haas, 2004). They are one of the most popular DSR models (Platt et al., 2022; Pathak et al., 2017; Verzelli et al., 2021; Patel & Ott, 2023). A key property of RCs is that only the weights of the linear output layer are trained, while those in the reservoir are kept fixed. The central idea in RCs is that the reservoir provides a huge but fixed repertoire of (basis for) potential dynamics (a bit similar in spirit to SVMs) that can be harvested to approximate any DS. While various architectures exist, a common formulation is given by (Patel & Ott, 2023):

$$\begin{aligned} \boldsymbol{r}_t &= \alpha \boldsymbol{r}_{t-1} + (1 - \alpha) \tanh\left(\boldsymbol{W}\boldsymbol{r}_{t-1} + \boldsymbol{W}_{in}\boldsymbol{u}_t + \boldsymbol{b}\right) \\ \hat{\boldsymbol{x}}_t &= \boldsymbol{W}_{out}\boldsymbol{r}_t, \end{aligned} \tag{45}$$

where $\boldsymbol{r}_t \in \mathbb{R}^M$ is the reservoir state, $\alpha \in \mathbb{R}$ a leakage parameter, $\boldsymbol{W} \in \mathbb{R}^{M \times M}$ the *fixed* reservoir connectivity matrix, $\boldsymbol{W}_{in} \in \mathbb{R}^{M \times N}$ a likewise *fixed* input-to-reservoir matrix weighing inputs $\boldsymbol{u}_t \in \mathbb{R}^N$, $\boldsymbol{b} \in \mathbb{R}^M$ a fixed bias vector, and $\boldsymbol{W}_{out} \in \mathbb{R}^{N \times M}$ the only trainable matrix mapping reservoir states to the observed data. Since parameter $\{\boldsymbol{W}, \boldsymbol{W}_{in}, \boldsymbol{b}\}$ are fixed and not trained, training an RC becomes particularly easy and fast, since it boils down to a simple linear regression problem, making this approach highly attractive. Specifically, given data $\boldsymbol{X} = [\boldsymbol{x}_1, \ldots, \boldsymbol{x}_T] \in \mathbb{R}^{N \times T}$ and the reservoir trajectory $\boldsymbol{R} = [\boldsymbol{r}_1, \ldots, \boldsymbol{r}_T] \in \mathbb{R}^{M \times T}$ obtained by driving the model through the inputs ($\boldsymbol{u}_t = \boldsymbol{x}_{t-1}$ during training and $\boldsymbol{u}_t = \hat{\boldsymbol{x}}_{t-1} = \boldsymbol{W}_{out}\boldsymbol{r}_{t-1}$ at test time), the loss function

$$\mathcal{L} = \|\boldsymbol{X} - \boldsymbol{W}_{out}\boldsymbol{R}\|_F^2 + \lambda\|\boldsymbol{W}_{out}\|_F^2 \tag{46}$$

results in a straightforward ridge regression problem with closed form solution

$$\boldsymbol{W}_{out} = \boldsymbol{X}\boldsymbol{R}^T\left(\boldsymbol{R}\boldsymbol{R}^T + \lambda\boldsymbol{I}\right)^{-1}. \tag{47}$$

However, since parameters of the reservoir are fixed after initialization, RCs generally depend on a large (and thus less tractable) reservoir ($M \geq 500$) and carefully designed initialization schemes (Patel & Ott, 2023). For our evaluation, we employed code adapted from Patel & Ott (2023), which we obtained upon request.

### E.2. Custom trained TS models

**AutoARIMA**   Autoregressive Integrated Moving Average models with seasonal components, ARIMA$(p,d,q)(P,D,Q)_m$ for short, are classical linear TSF tools that have been in use for decades (Box et al., 2015), given in compact notation by

$$\Psi_P(B^m)\psi_p(B)(1 - B^m)^D(1 - B)^d y_t = c + \Theta_Q(B^m)\theta_q(B)\epsilon_t , \tag{48}$$

where $c$ is a constant offset, $\{\epsilon_t\}$ comes from a white noise process with zero mean and variance $\sigma^2$, $B$ denotes the backshift operator, $B^k y_t = y_{t-k}$, and a seasonal component with period $m$. The non-seasonal AR and MA parts are in the polynomials $\psi(B)$ and $\theta(B)$ of orders $p$ and $q$, respectively, while the seasonal components are given by the polynomials $\Psi(B^m)$ and $\Theta(B^m)$ of orders $P$ and $Q$ in the seasonal backshift $B^m$. The factors $(1 - B)^d$ and $(1 - B^m)^D$ apply non-seasonal and seasonal differencing for trend removal and seasonal variation.

For our comparisons, we used an automatic ARIMA model selection procedure (AutoARIMA) (Hyndman & Khandakar, 2008) and its implementation in the StatsForecast package (Garza et al., 2022). AutoARIMA searches over a predefined hyperparameter space and selects the ARIMA specification that optimizes an information criterion, thereby automatically determining the autoregressive (AR), differencing (I), and moving average (MA) orders as well as potential seasonal terms. Depending on the selected orders, the resulting model can effectively reduce to pure AR, MA, ARMA, or non-seasonal ARIMA. Unlike the order parameters $p, d, q, P, D, Q$, the seasonality $m$ is not optimized by AutoARIMA and therefore required a small grid search around plausible values.

**NBEATS**   Neural basis expansion analysis for interpretable time series forecasting (N-BEATS), introduced in (Oreshkin et al., 2020), is a deep learning architecture for univariate TSF built from stacks of fully connected MLP blocks with ReLU

activations and no recurrent, convolutional, or transformer components. It uses both backward and forward residual links between the MLP blocks to form the final prediction. Model parameters are learned by minimizing the sMAPE forecasting loss between predictions and ground truth. In addition to the generic configuration, N-BEATS also offers a variant that enforces trend and seasonality structure via polynomial and harmonic bases, enabling partial interpretability through this decomposition. In this work, AutoNBEATS from the neuralforecast library (Olivares et al., 2022) is used for model training and prediction.

**PatchTST**   PatchTST (Nie et al., 2023) is a Transformer-based architecture for multivariate TSF. A key component of PatchTST is its patching mechanism, which segments an input time series $\mathbf{x} \in \mathbb{R}^{1 \times T}$ into $N_P$ (typically overlapping) subseries, or patches, $\mathbf{x}_p \in \mathbb{R}^{L_P \times N_P}$ of length $L_P < T$. These patches serve as input tokens to the encoder-only Transformer, enabling the model to capture richer context information than point-wise time step representations while simultaneously reducing the number of input tokens. Another core feature of PatchTST is channel independence: for multivariate forecasting, each time series dimension is treated as an independent univariate series, while sharing the same Transformer embeddings and weights across all $N$ channels. The model is trained using MSE loss between the forecasts and ground truth values. In this work, we employ the AutoPatchTST implementation from the NeuralForecast library (Olivares et al., 2022) for training and prediction.

### E.3. Foundation models

**DynaMix**   DynaMix (Hemmer & Durstewitz, 2025) is, to our knowledge, the only foundation model so far which achieves zero-shot DS reconstructions, i.e. performs a type of in-context generalization without parameter fine-tuning. It is based on a mixture-of-experts architecture employing $J$ different AL-RNNs (cf. eq. 42) as experts. Their individual next-state predictions are combined into a single prediction through a weighted sum $\mathbf{z}_{t+1} = \sum_{j=1}^{J} w_{j,t}^{exp} \cdot \mathbf{z}_{t+1}^j$, with weights $w_{j,t}^{exp}$ obtained from a gating network

$$\boldsymbol{w}_t^{exp} = \sigma \left( \frac{\mathrm{MLP}(\tilde{\boldsymbol{C}} \boldsymbol{w}_t^{att}, \boldsymbol{z}_t)}{\tau_{\exp}} \right) \in \mathbb{R}^J \quad , \tag{49}$$

which takes as input a context signal $\boldsymbol{C} \in \mathbb{R}^{N \times T_C}$ transformed into a set of temporal features $\tilde{\boldsymbol{C}} = \mathrm{CNN}(\boldsymbol{C})$ by a CNN, which are in turn weighted by time-dependent attention weights $\boldsymbol{w}_t^{att}$ through a state-attention mechanism defined as

$$\boldsymbol{w}_t^{att} = \sigma \left( \frac{\left| \boldsymbol{C} - (\boldsymbol{D} \boldsymbol{z}_t + \boldsymbol{\epsilon}) \, \mathbf{1}_{T_C}^\top \right|^\top \mathbf{1}_N}{\tau_{\mathrm{att}}} \right) \in \mathbb{R}^{T_C} \quad . \tag{50}$$

$\tau_{\exp}$ and $\tau_{\mathrm{att}}$ are temperature parameters in these equations, $\boldsymbol{D}$ is a learnable matrix, and $\mathbf{1}_{\{N, T_C\}}$ denotes column vectors of ones. This fully recurrent foundation model was trained following DSR principles by STF, where the training corpus consisted of $\approx 6 \cdot 10^5$ trajectories drawn from 34 distinct cyclic or chaotic DS. For evaluation we used the implementation from https://github.com/DurstewitzLab/DynaMix-python.

**Chronos**   Chronos is a transformer-based foundation model for probabilistic time series forecasting that repurposes LLM architectures for temporal data (Ansari et al., 2024). It converts real-valued time series into discrete token sequences through scaling and quantization, enabling autoregressive forecasting using a standard transformer architecture. The model is pretrained on large-scale datasets consisting of both synthetic TS created using Gaussian processes and a broad collection of real-world TS (Ansari et al., 2024; Godahewa et al., 2021), including traffic, weather/climate, electricity, and web data. This broad pretraining enables Chronos to perform zero-shot forecasts on a wide variety of TS without task-specific fine-tuning. For our evaluation we used the standard pipeline as described in https://github.com/amazon-science/chronos-forecasting.

**Chronos-2**   Chronos-2 is a recently proposed multivariate extension of the original Chronos model (Ansari et al., 2025). It retains the T5 encoder architecture (Raffel et al., 2020) used in the initial Chronos design, and augments it with a novel group-attention mechanism. This group-attention mechanism enables sharing information across related groups of TS, such as different variables or covariates. Through this the model can, in contrast to the original Chronos, also handle multivariate or covariate-informed TS in a zero-shot manner. The model was trained in both univariate and multivariate configurations. For the univariate case, the Chronos training corpus was employed together with GIFT-EVAL for pretraining. In the multivariate case, the model was trained exclusively on synthetic data generated using methods

such as KernelSynth (Ansari et al., 2024). With this training scheme, the model not only supports multivariate forecasting but also often outperforms the original Chronos. For our evaluation we followed the standard pipeline as described in `https://github.com/amazon-science/chronos-forecasting`.

**TiRex**   TiRex is a recently introduced foundation model for zero-shot time series forecasting based on the LSTM (Hochreiter & Schmidhuber, 1997). Specifically, it uses the xLSTM architecture (Auer et al., 2025), an enhanced LSTM variant that combines explicit state tracking with improved in-context learning capabilities. TiRex adopts a decoder-only recurrent design for autoregressive multi-step forecasting. The model is pre-trained on the same data as in (Ansari et al., 2024), comprising real-world time series and synthetic data generated using the KernelSynth method. In addition, subsets of the GiftEVAL pre-training dataset are included, resulting in a total of 47.5 million training TS. Overall, TiRex achieves SOTA zero-shot forecasting performance on benchmarks such as GiftEVAL while using fewer parameters than competing transformer-based models. Evaluation is performed using the pipeline given in `https://github.com/NX-AI/tirex`.

**Panda**   Panda is a recently proposed foundation model for zero-shot short-term forecasting of DS (Lai et al., 2025). Like Chronos, it is based on a transformer architecture that operates on patched representations of TS. Unlike Chronos, however, the model is trained on simulation data from $2 \cdot 10^4$ different DS created by combining base DS from Gilpin (2022) in so-called skew-product form, where one dependent system is driven by the other independently evolving system. Evaluation is performed as in `https://github.com/abao1999/panda`.

# F. Further Results and Illustrations

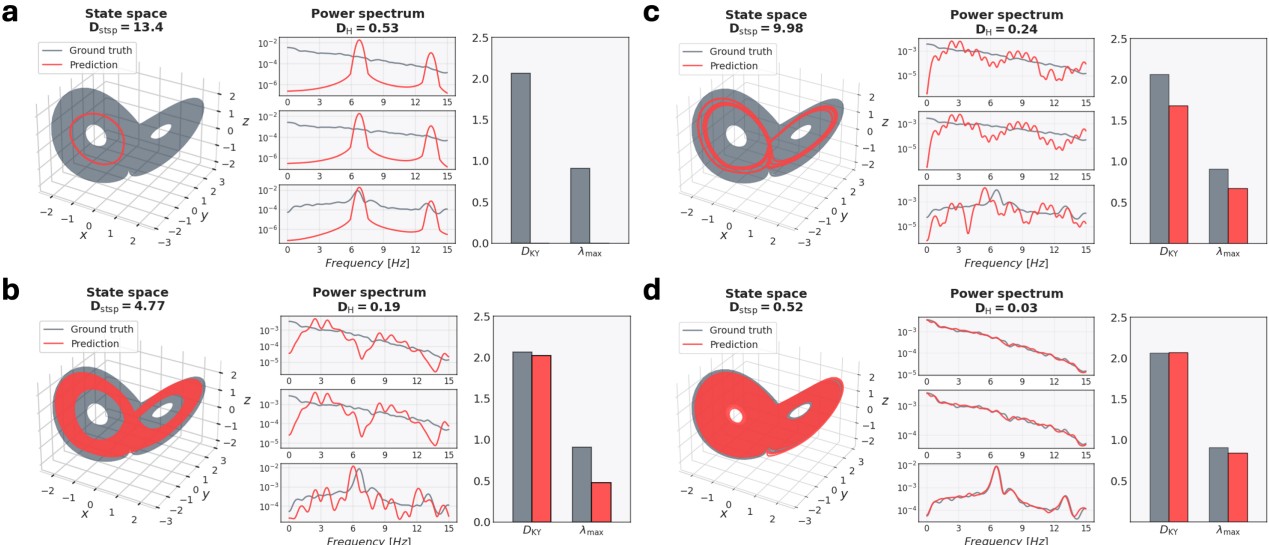

*Figure 8.* Illustration of DSR measures: Comparison of geometric (dis)agreement ($D_{\text{stsp}}$), power spectral distance ($D_{\text{H}}$), Kaplan-Yorke fractal dimension ($D_{\text{KY}}$), and max. Lyapunov exponent ($\lambda_{\max}$) on reconstructions of the chaotic Lorenz-63 system of different quality from very poor (**a**) to excellent (**d**), obtained by an AL-RNN assessed at different training stages.

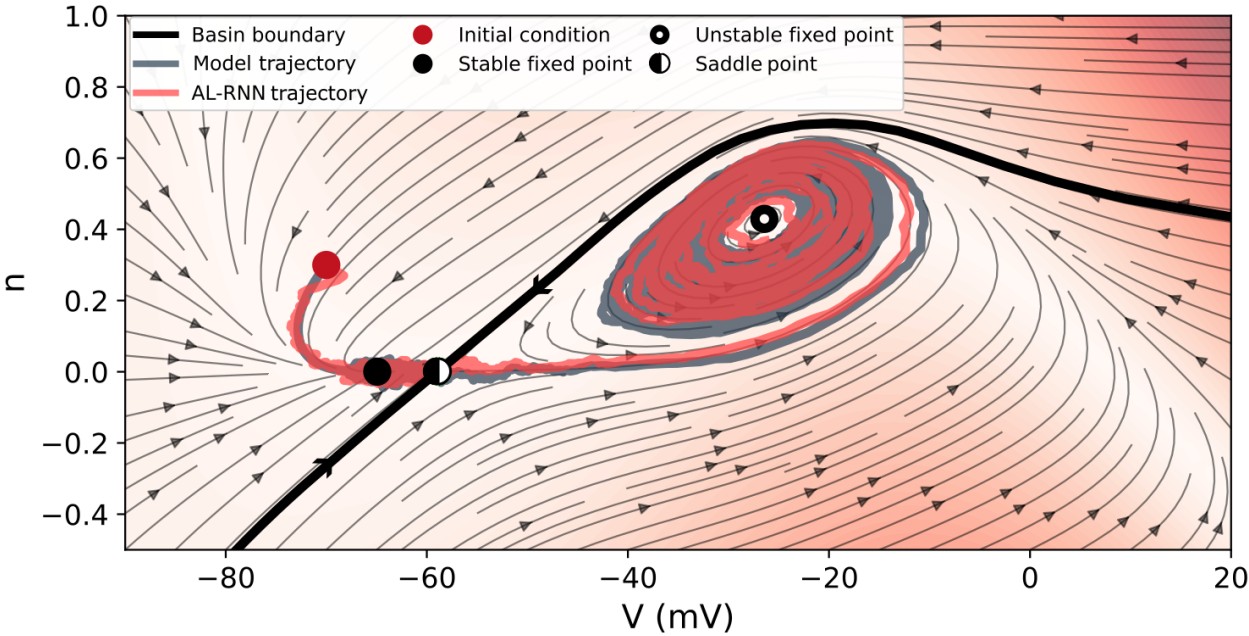

*Figure 9.* N-tipping in state space: Illustration of the basin boundary crossing of the noisy trajectory from Fig.1b.

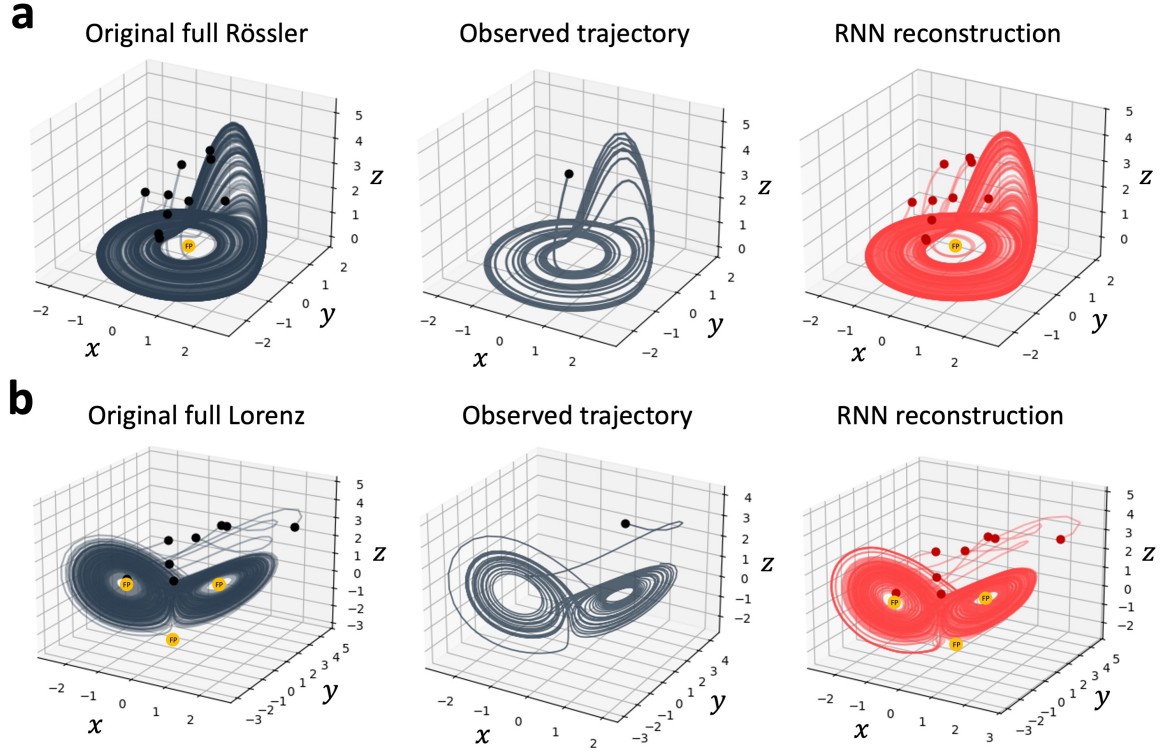

*Figure 10.* A DSR model trained on just one short trajectory (center) from the chaotic Rössler (**a**) or Lorenz (**b**) system (left) correctly reconstructs the full chaotic attractor (red trajectory), generalizes to novel initial conditions (red dots), and correctly infers the location of fixed points not shown in training (yellow dots). Part (a) based on Suppl. Fig. 4 of Durstewitz et al. (2023).

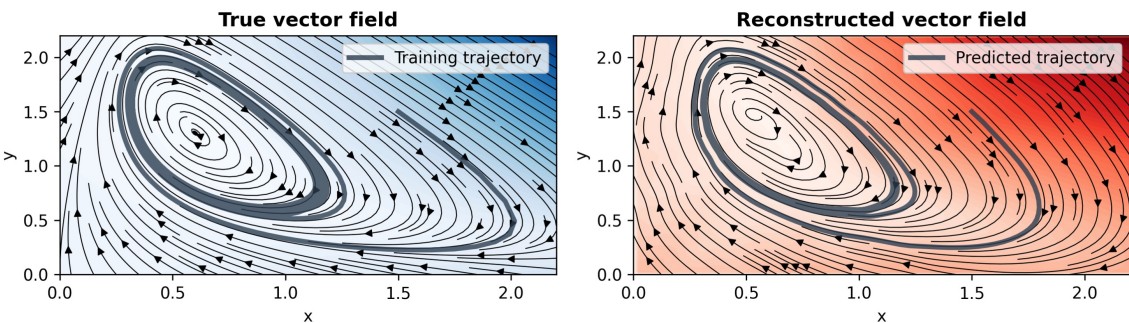

*Figure 11.* Although only trained on a short trajectory snippet (gray curve) from the Selkov system (left), the trained DSR model (shallow PLRNN (Hess et al., 2023); right) correctly infers properties of the surrounding vector field (darker shading = higher flow velocity), enabling it to generalize to new initial conditions.

### Rössler

$$\dot{x} = -y - z$$
$$\dot{y} = x + ay$$
$$\dot{z} = b + (x - c)z$$

### Newton-Leipnik

$$\dot{x} = -ax + y + 10yz$$
$$\dot{y} = -x - 0.4y + 5xz$$
$$\dot{z} = bz - 5xy$$

### Aizawa

$$\dot{x} = (z - b)x - dy$$
$$\dot{y} = dx + (z - b)y$$
$$\dot{z} = c + az - z^3/3$$
$$\qquad - (x^2 + y^2)(1 + ez) + fx^3 z$$

### Rikitake

$$\dot{x} = -\mu x + yz$$
$$\dot{y} = -\mu y - ax + xz$$
$$\dot{z} = 1 - xy$$

### Forced Brusselator

$$\dot{x} = a + x^2 y - (b + 1)x + f\cos(z)$$
$$\dot{y} = bx - x^2 y$$
$$\dot{z} = w$$

*Figure 12.* Illustration of a DSR training corpus. Time series (center) from different chaotic attractors (left), generated by the equations on the right. The top time series are from *non-autonomous* systems where a control parameter (in light blue on the right) was systematically varied, while the bottom time series are from autonomous and stationary system simulations (135 such systems are collected in Gilpin (2022), from which through recombination $\approx 10^4$ systems are created in Lai et al. (2025)).

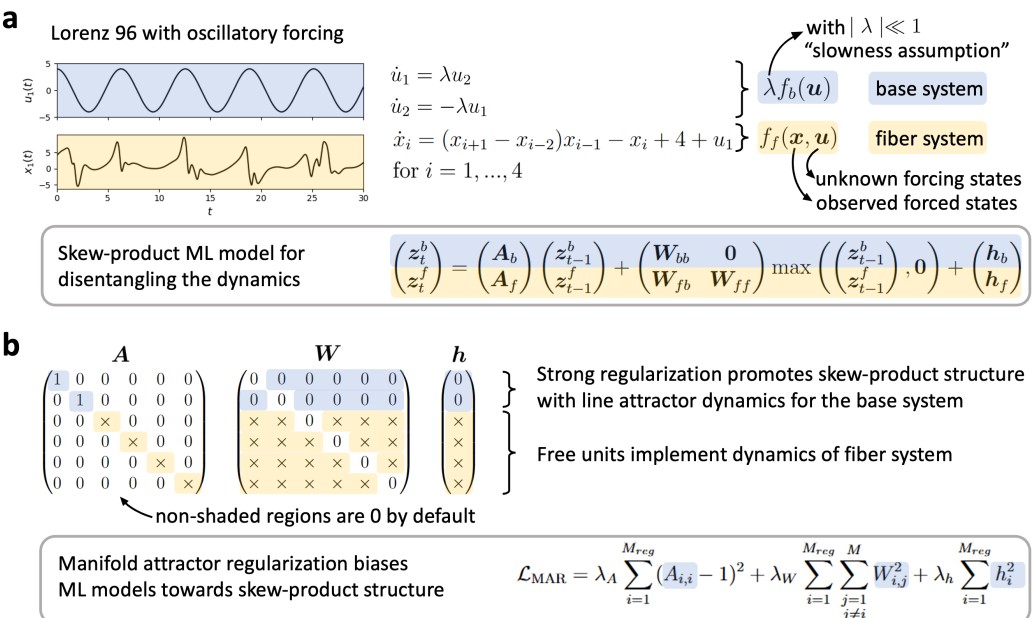

*Figure 13.* **a**) Example of generic skew-product form of a dynamical system that implements non-autonomous DS by separating a slow base (driver) system from a fast fiber (forced, dependent) system (Kloeden & Yang, 2020). **b**) Specific example (using a piecewise-linear RNN) of how such a form may be enforced in training through a particular regularization (inductive bias) on parameters which encourages the RNN to segregate its latent space into base and fiber system, inspired by Schmidt et al. (2021).

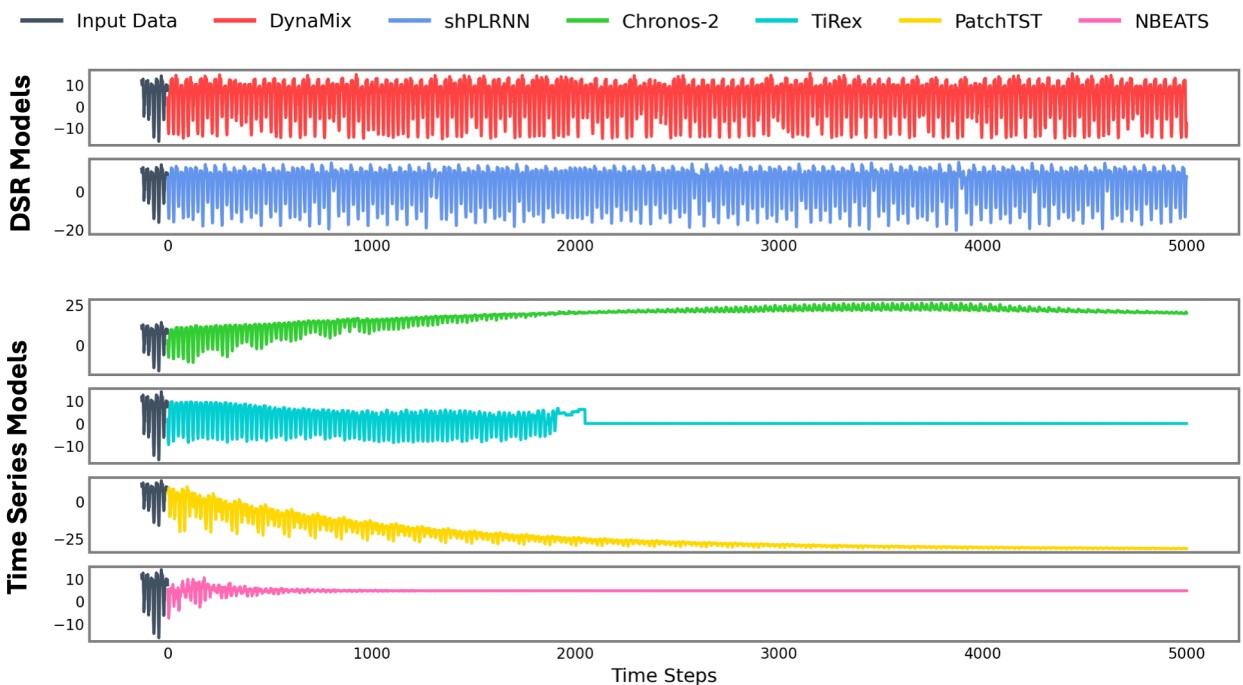

*Figure 14.* Long-term forecasts of ETTh1 dataset for different DSR and TS models.

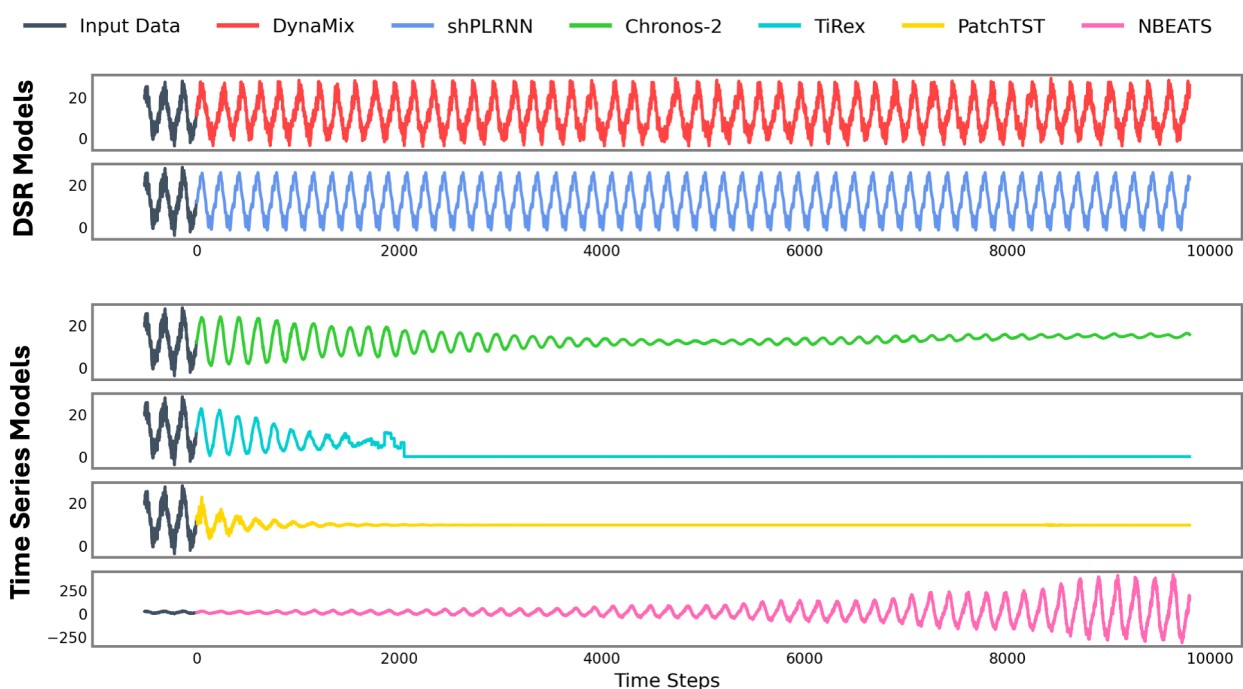

*Figure 15.* Long-term forecasts of weather temperature dataset for different DSR and TS models.

*Table 1.* Performance comparison of custom trained DSR and TS models on empirical time series in terms of geometrical divergence ($D_{stsp}$), long-term temporal distance ($D_H$), and forecast error (MASE). Best in red, second-best in blue.

| System | ALRNN | | | shPLRNN | | | RC | | | AutoARIMA | | | PatchTST | | | NBEATS | | |
|---|---|---|---|---|---|---|---|---|---|---|---|---|---|---|---|---|---|---|
| | $D_{stsp}$ | $D_H$ | MASE | $D_{stsp}$ | $D_H$ | MASE | $D_{stsp}$ | $D_H$ | MASE | $D_{stsp}$ | $D_H$ | MASE | $D_{stsp}$ | $D_H$ | MASE | $D_{stsp}$ | $D_H$ | MASE |
| ETTh1 | 0.46±0.34 | 0.16±0.02 | 3.09±0.07 | 0.21±0.13 | 0.18±0.01 | 1.61±0.08 | 0.35±0.01 | 0.17±0.01 | 1.09±0.00 | 3.92 | 0.22 | 1.46 | 3.85±2.66 | 0.16±0.05 | 4.78±1.82 | 3.14±0.57 | 0.38±0.13 | 3.77±0.42 |
| Traffic | 1.14±0.57 | 0.14±0.03 | 1.90±0.04 | 0.25±0.08 | 0.09±0.01 | 2.35±0.06 | 0.10±0.02 | 0.13±0.02 | 1.26±0.27 | 8.98 | 0.41 | 2.16 | 1.23±0.68 | 0.24±0.10 | 0.98±0.08 | 1.54±0.12 | 0.13±0.03 | 0.92±0.06 |
| Cloud Requests | 1.28±0.12 | 0.12±0.00 | 3.02±0.05 | 1.14±0.10 | 0.12±0.00 | 2.88±0.03 | 0.73±0.00 | 0.74±0.01 | 23.12±2.06 | 0.61 | 0.10 | 1.90 | 3.21±2.04 | 0.15±0.03 | 2.40±0.37 | 2.10±0.34 | 0.12±0.02 | 4.05±0.79 |
| Weather (Temp.) | 0.73±0.02 | 0.11±0.00 | 1.80±0.04 | 0.71±0.28 | 0.12±0.00 | 1.62±0.12 | 0.07±0.02 | 0.12±0.03 | 2.53±0.28 | 0.11 | 0.11 | 2.60 | 1.04±0.87 | 0.12±0.03 | 3.19±1.25 | 1.22±0.83 | 0.11±0.02 | 2.09±0.12 |
| Weather (Press.) | 0.23±0.09 | 0.29±0.02 | 4.66±0.22 | 0.25±0.03 | 0.30±0.02 | 4.80±0.24 | 1.90±1.05 | 0.89±0.05 | 4.68±0.62 | 13.26 | 1.00 | 4.63 | 5.90±0.38 | 0.45±0.01 | 5.63±0.54 | 5.42±2.45 | 0.49±0.18 | 8.47±3.98 |
| Human fMRI | 0.27±0.09 | 0.21±0.04 | 5.38±0.10 | 0.26±0.01 | 0.21±0.01 | 4.88±0.31 | 0.24±0.03 | 0.20±0.01 | 4.20±0.27 | 10.48 | 1.00 | 2.59 | 6.08±1.42 | 0.25±0.05 | 3.60±0.64 | 8.15±3.01 | 0.42±0.26 | 2.87±0.46 |
| Human EEG | 0.39±0.23 | 0.35±0.18 | 2.78±0.61 | 0.22±0.02 | 0.25±0.02 | 1.95±0.11 | 0.23±0.01 | 0.21±0.02 | 4.44±0.49 | 12.36 | 1.00 | 2.37 | 4.86±3.81 | 0.41±0.11 | 3.50±0.27 | 5.16±0.39 | 0.46±0.04 | 2.33±0.28 |
| Air Passengers | 2.78±1.63 | 0.18±0.03 | 5.55±1.54 | 4.39±1.86 | 0.21±0.12 | 9.01±5.02 | 9.06±0.17 | 0.28±0.01 | 5.50±0.15 | 1.15 | 0.05 | 1.25 | 1.96±1.27 | 0.13±0.03 | 6.31±0.34 | 1.98±0.03 | 0.11±0.01 | 2.41±0.11 |
| # Trainable Param. | $\mathcal{O}(10^2)-\mathcal{O}(10^3)$ | | | $\mathcal{O}(10^2)-\mathcal{O}(10^3)$ | | | $\mathcal{O}(10^4)$ | | | $\mathcal{O}(10^1)$ | | | $\mathcal{O}(10^4)-\mathcal{O}(10^6)$ | | | $\mathcal{O}(10^6)$ | | |

*Table 2.* Performance comparison of DSR and TS foundation models on empirical time series in terms of geometrical divergence ($D_{stsp}$), long-term temporal distance ($D_H$), and forecast error (MASE). Best in red, second-best in blue.

| System | DynaMix | | | Chronos-t5-base | | | Chronos-2 | | | TiRex | | | Panda | | |
|---|---|---|---|---|---|---|---|---|---|---|---|---|---|---|---|
| | $D_{stsp}$ | $D_H$ | MASE | $D_{stsp}$ | $D_H$ | MASE | $D_{stsp}$ | $D_H$ | MASE | $D_{stsp}$ | $D_H$ | MASE | $D_{stsp}$ | $D_H$ | MASE |
| ETTh1 | 0.60±0.01 | 0.15±0.00 | 1.61±0.41 | 1.58±0.23 | 0.20±0.01 | 1.30±0.03 | 3.44 | 0.16 | 1.71 | 9.20 | 0.29 | 1.46 | 8.18 | 0.69 | 2.86 |
| Traffic | 0.86±0.02 | 0.18±0.02 | 0.98±0.10 | 0.78±0.24 | 0.11±0.01 | 0.62±0.01 | 0.55 | 0.13 | 0.65 | 3.41 | 0.81 | 0.59 | 4.47 | 0.45 | 0.99 |
| Cloud Requests | 0.67±0.00 | 0.07±0.01 | 2.62±0.02 | 3.72±1.91 | 0.53±0.08 | 2.43±0.02 | 2.10 | 0.14 | 1.66 | 3.06 | 0.89 | 1.44 | 7.20 | 0.23 | 2.38 |
| Weather (Temp.) | 0.06±0.00 | 0.07±0.01 | 1.89±0.04 | 8.92±3.75 | 0.44±0.24 | 7.24±0.40 | 2.27 | 0.17 | 1.59 | 3.00 | 0.90 | 1.91 | 1.15 | 0.14 | 1.60 |
| Weather (Press.) | 0.18±0.01 | 0.26±0.01 | 4.21±0.02 | 3.23±0.67 | 0.54±0.00 | 4.76±0.16 | 11.16 | 0.47 | 4.94 | 7.29 | 0.75 | 4.75 | 10.23 | 0.68 | 4.72 |
| Human fMRI | 0.23±0.01 | 0.19±0.00 | 2.03±0.10 | 9.05±0.40 | 0.31±0.01 | 2.66±0.07 | 10.18 | 0.22 | 2.61 | 8.44 | 0.38 | 2.53 | 5.29 | 0.53 | 5.12 |
| Human EEG | 0.18±0.00 | 0.23±0.01 | 3.00±0.44 | 11.99±0.18 | 0.68±0.07 | 3.07±0.04 | 8.91 | 0.41 | 2.76 | 8.71 | 0.74 | 2.46 | 6.79 | 0.54 | 2.38 |
| Air Passengers | 0.78±0.04 | 0.11±0.00 | 2.83±0.11 | 8.29±0.71 | 0.16±0.07 | 1.56±0.08 | 6.55 | 0.29 | 1.64 | 8.64 | 0.11 | 1.43 | 9.25 | 0.13 | 2.48 |
| # Parameters | $\mathcal{O}(10^4)$ | | | $\mathcal{O}(10^8)$ | | | $\mathcal{O}(10^8)$ | | | $\mathcal{O}(10^7)$ | | | $\mathcal{O}(10^7)$ | | |

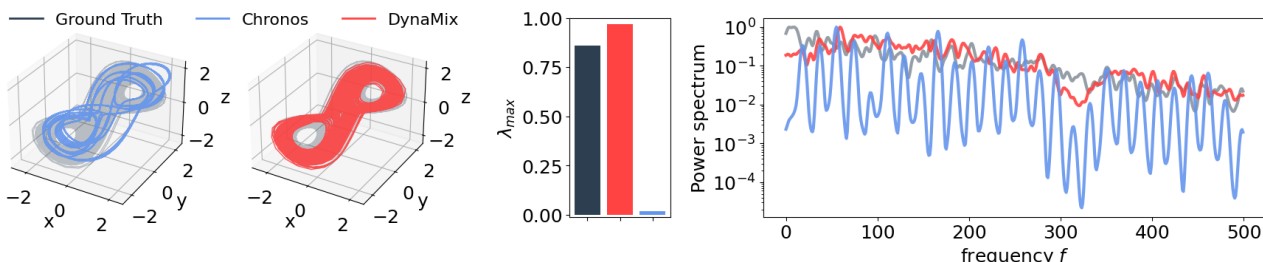

*Figure 16.* Example forecast of the chaotic Lorenz-63 system using DynaMix and Chronos-t5-base. Chronos' tendency to parrot its context input leads to a cyclic repetition of a fixed pattern, and Chronos therefore fundamentally fails to capture the truly aperiodic, chaotic behavior. This manifests as cyclic trajectories in the delay-embedded (cf. Appx. A) state space (left), the close-to-zero maximum Lyapunov exponent (center) and by a sharply peaked (instead of the truly broad) power spectrum (right). Results reproduced from Hemmer & Durstewitz (2025).

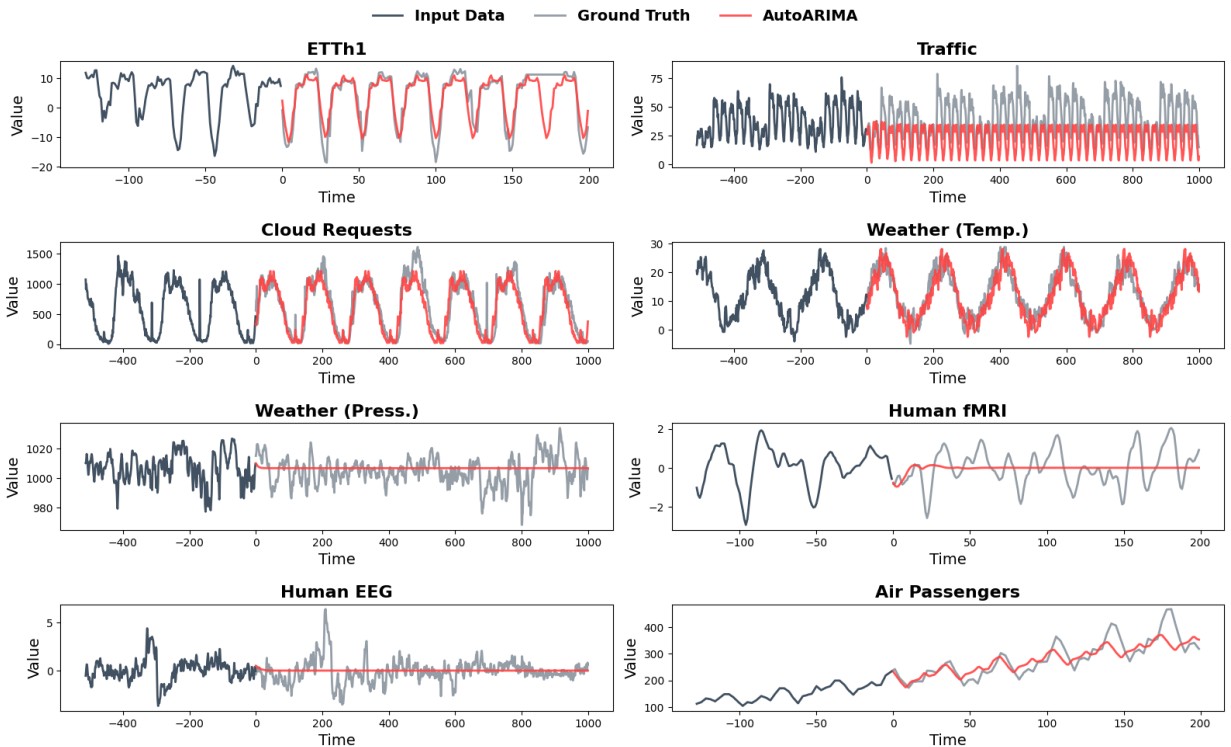

*Figure 17.* TS forecasts using custom trained AutoARIMA.

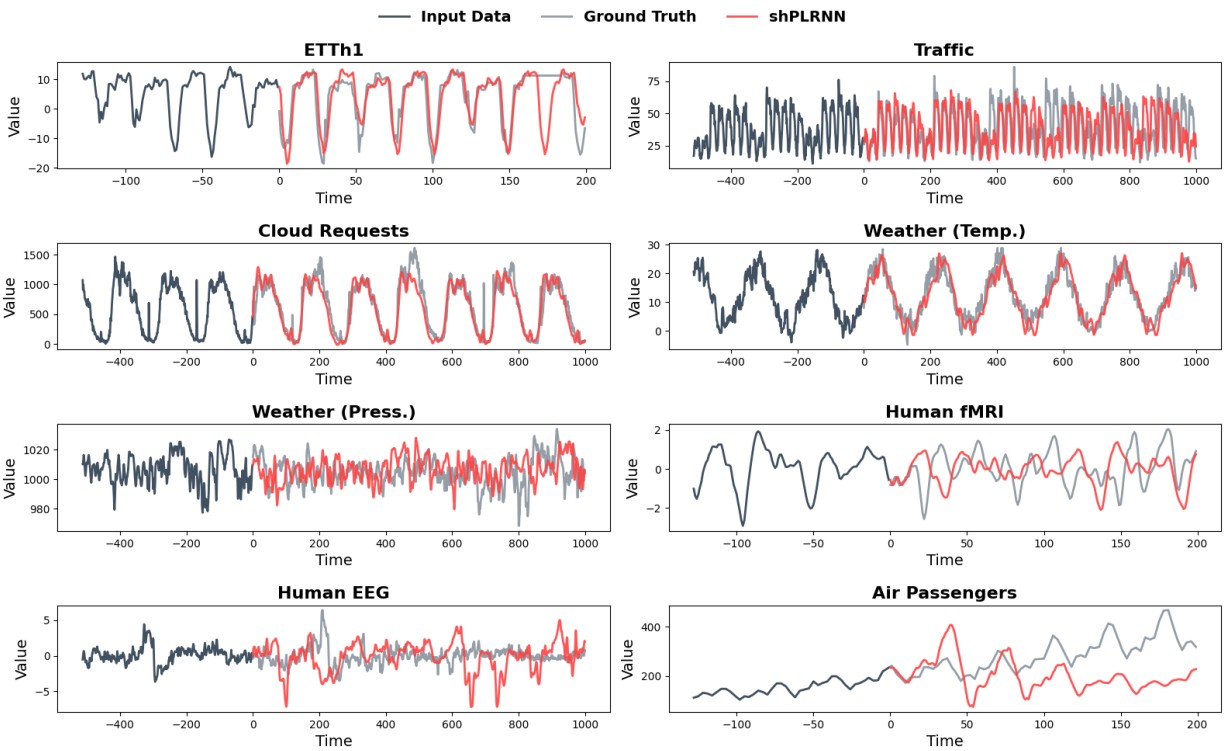

*Figure 18.* TS forecasts using custom trained shallow PLRNN.

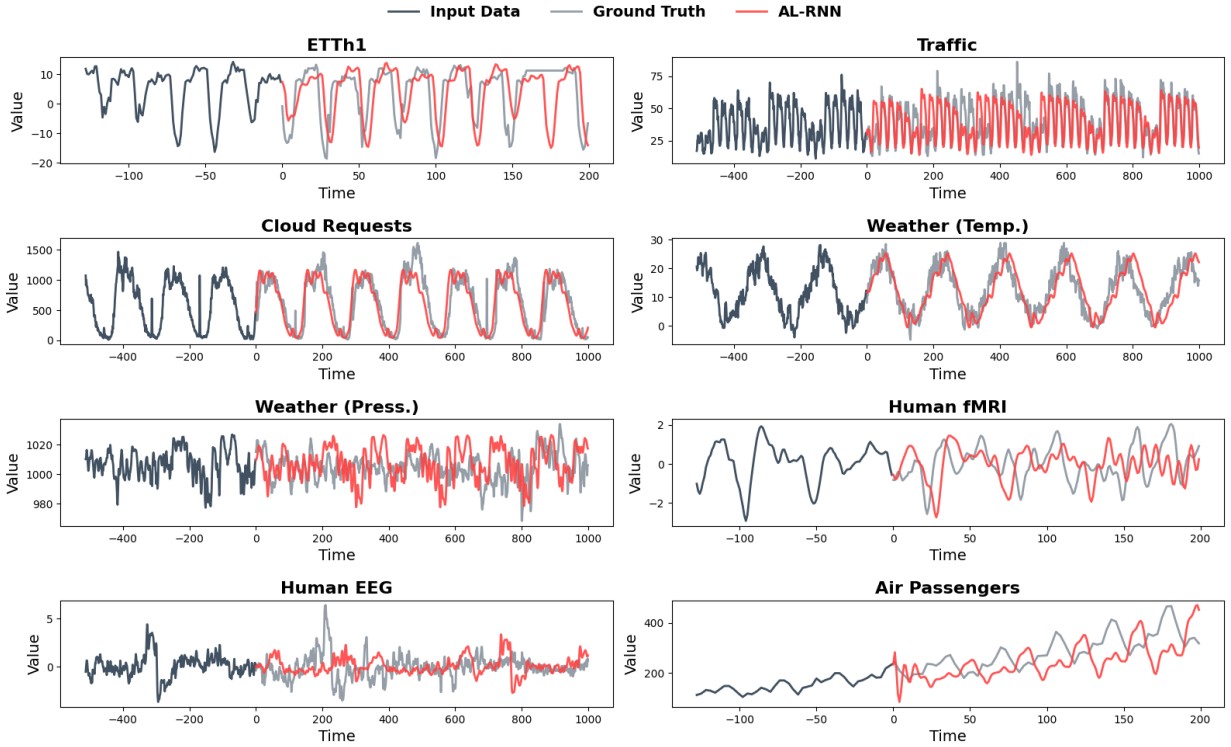

*Figure 19.* TS forecasts using custom trained AL-RNN.

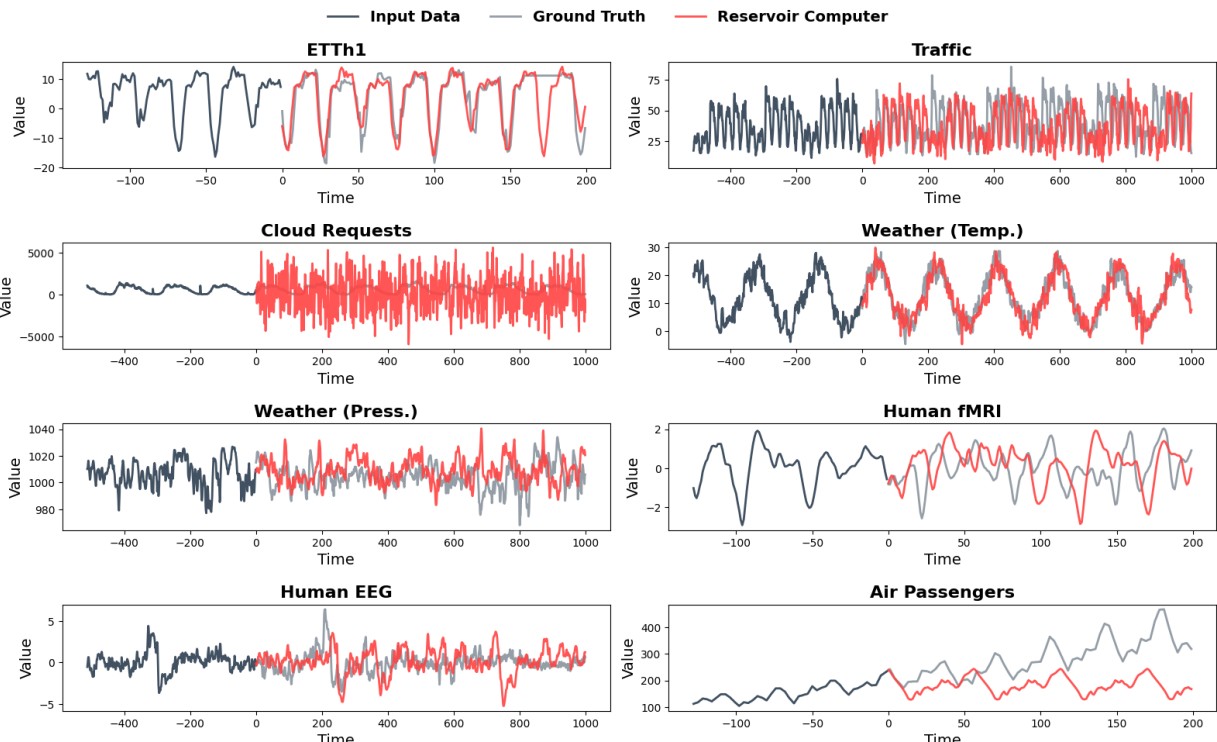

*Figure 20.* TS forecasts using custom trained Reservoir Computers.

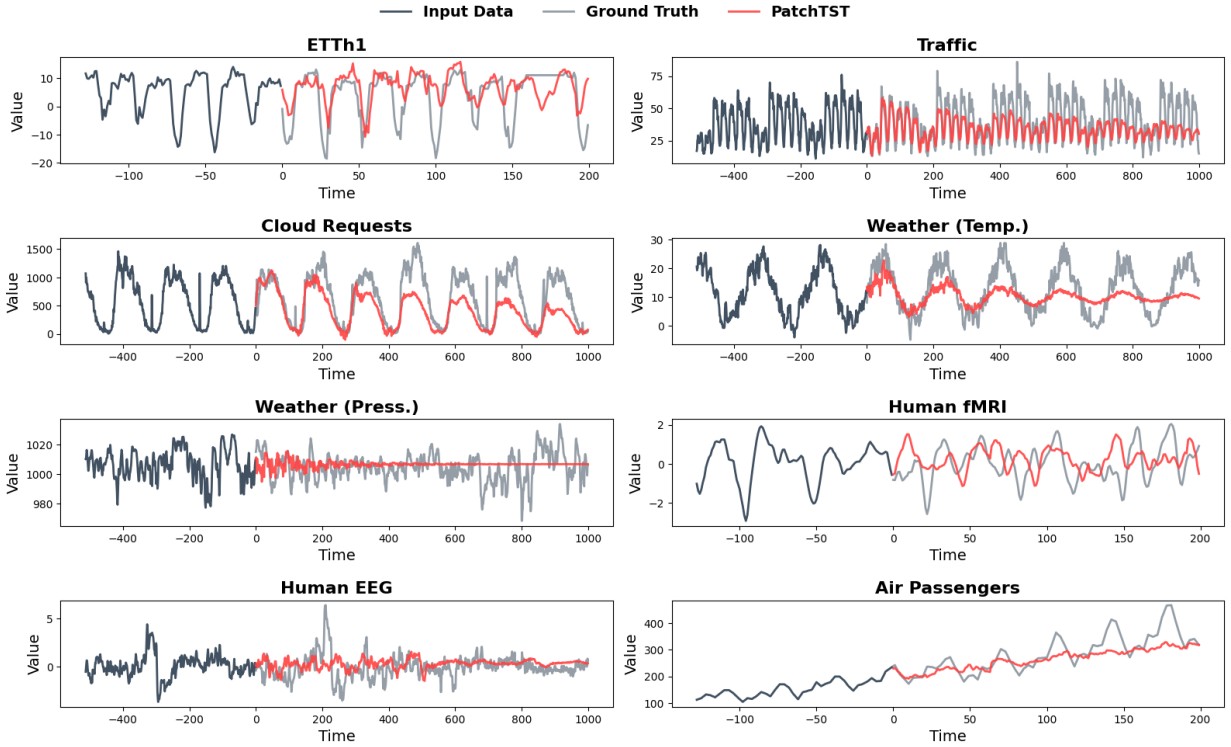

*Figure 21.* TS forecasts using custom trained PatchTST.

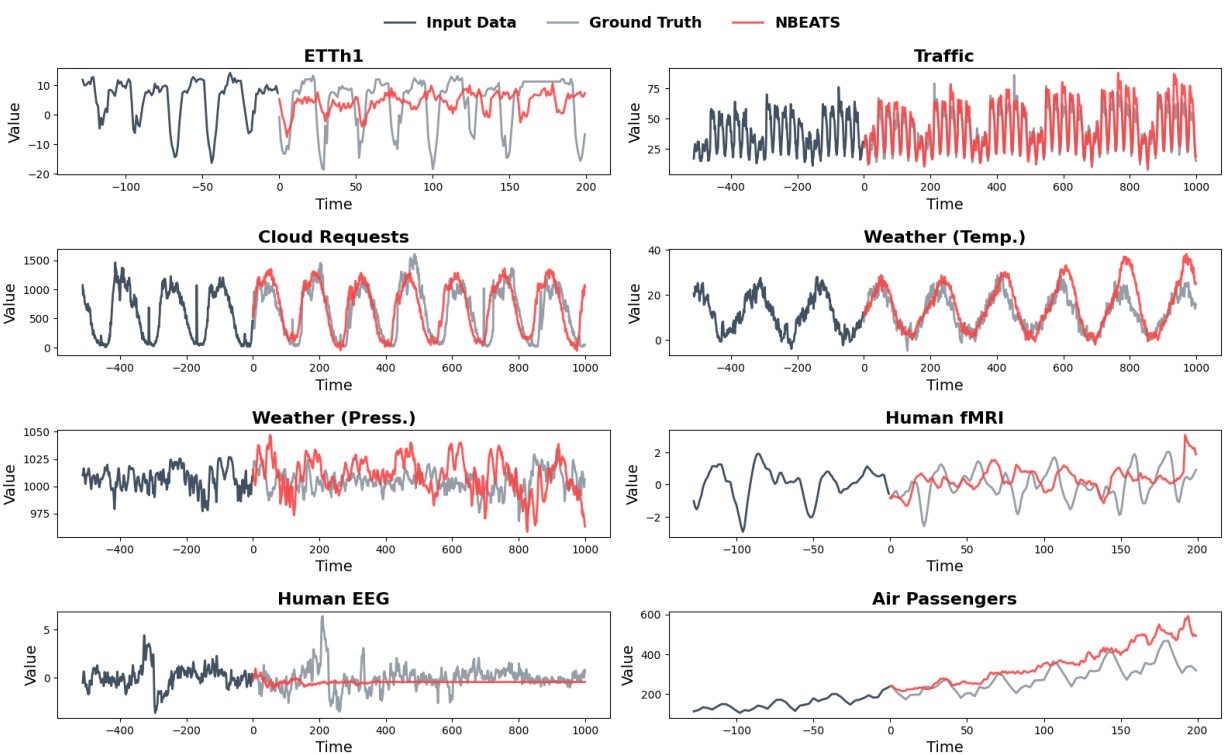

*Figure 22.* TS forecasts using custom trained NBEATS.

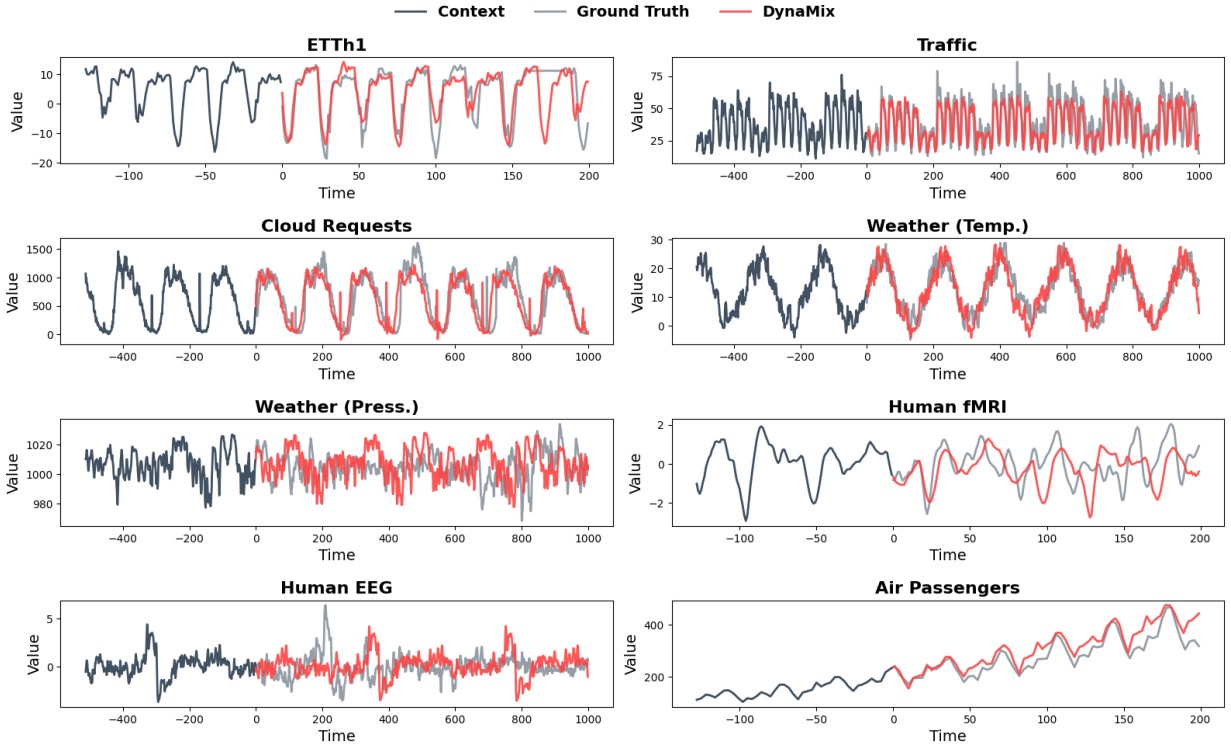

*Figure 23.* Zero-shot TS forecasts using DynaMix FM.

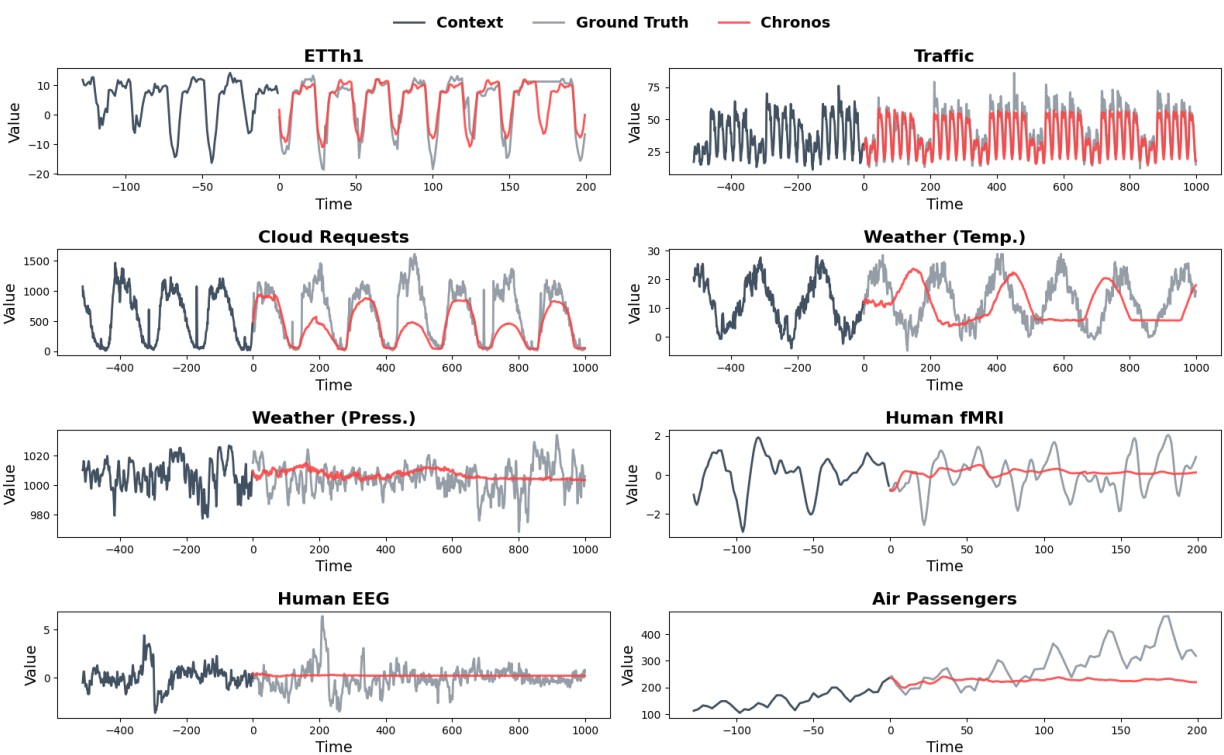

*Figure 24.* Zero-shot TS forecasts using Chronos-t5-base FM.

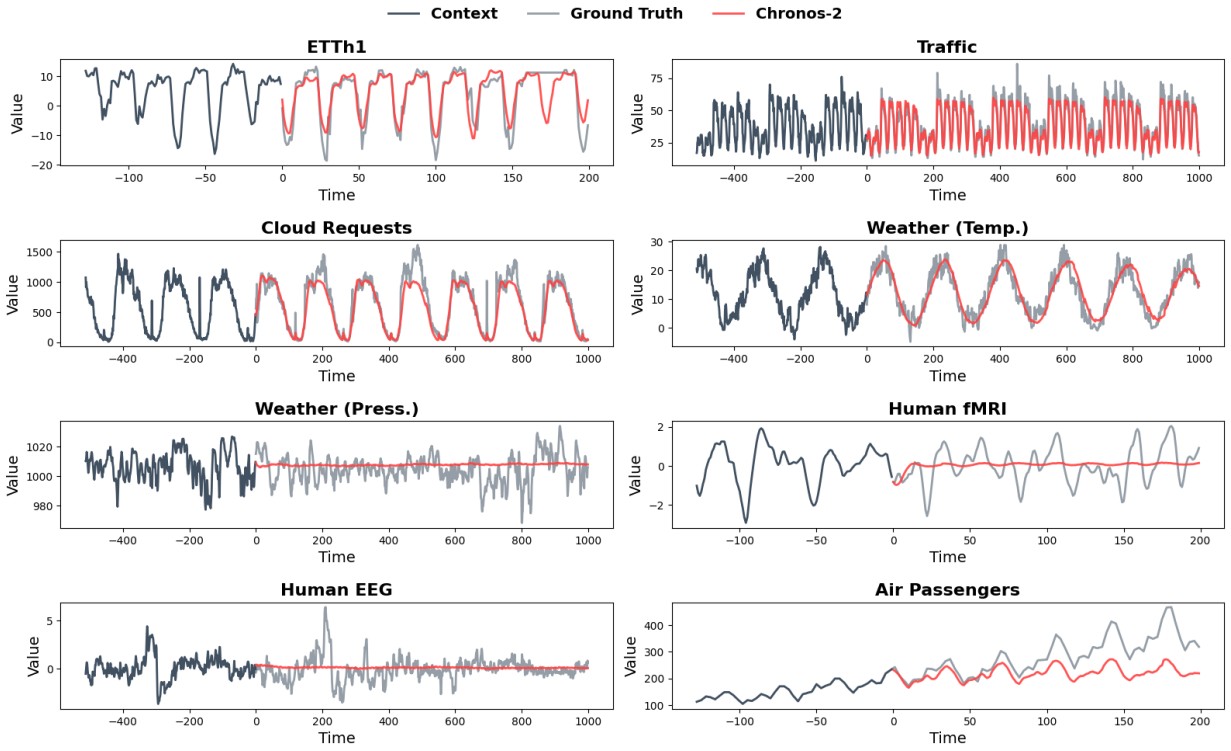

*Figure 25.* Zero-shot TS forecasts using Chronos-2 FM.

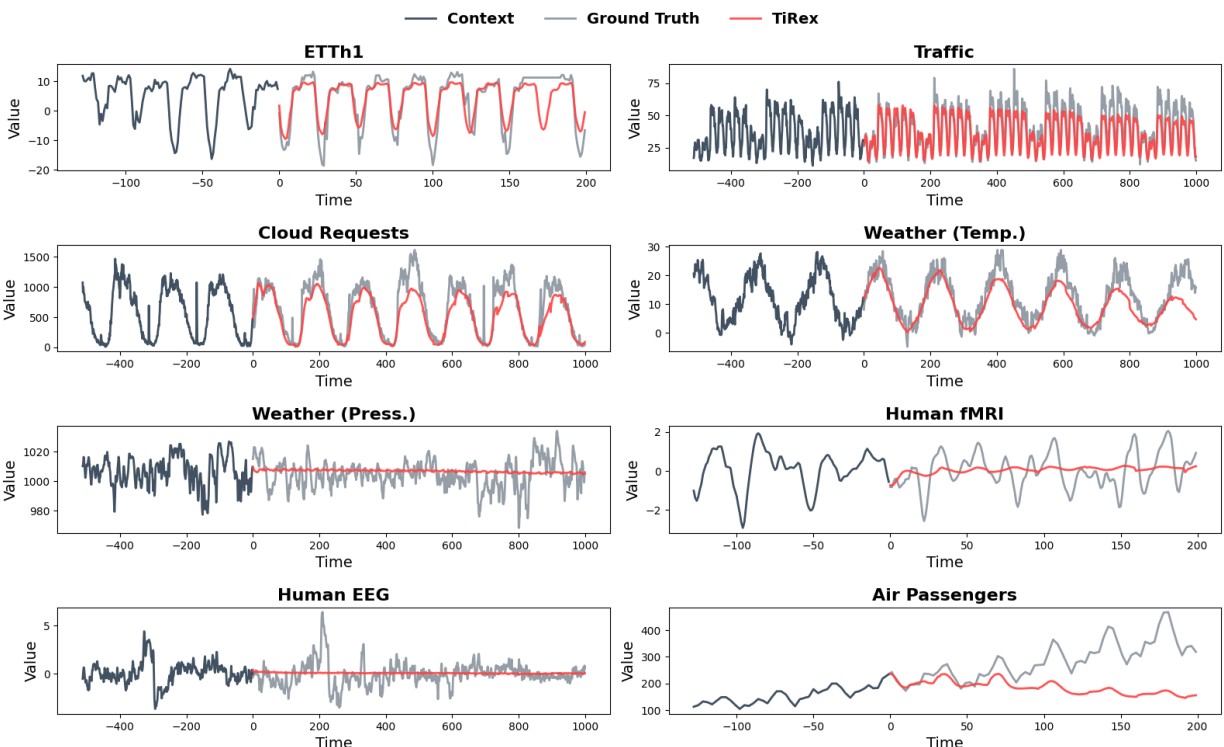

*Figure 26.* Zero-shot TS forecasts using TiRex FM.

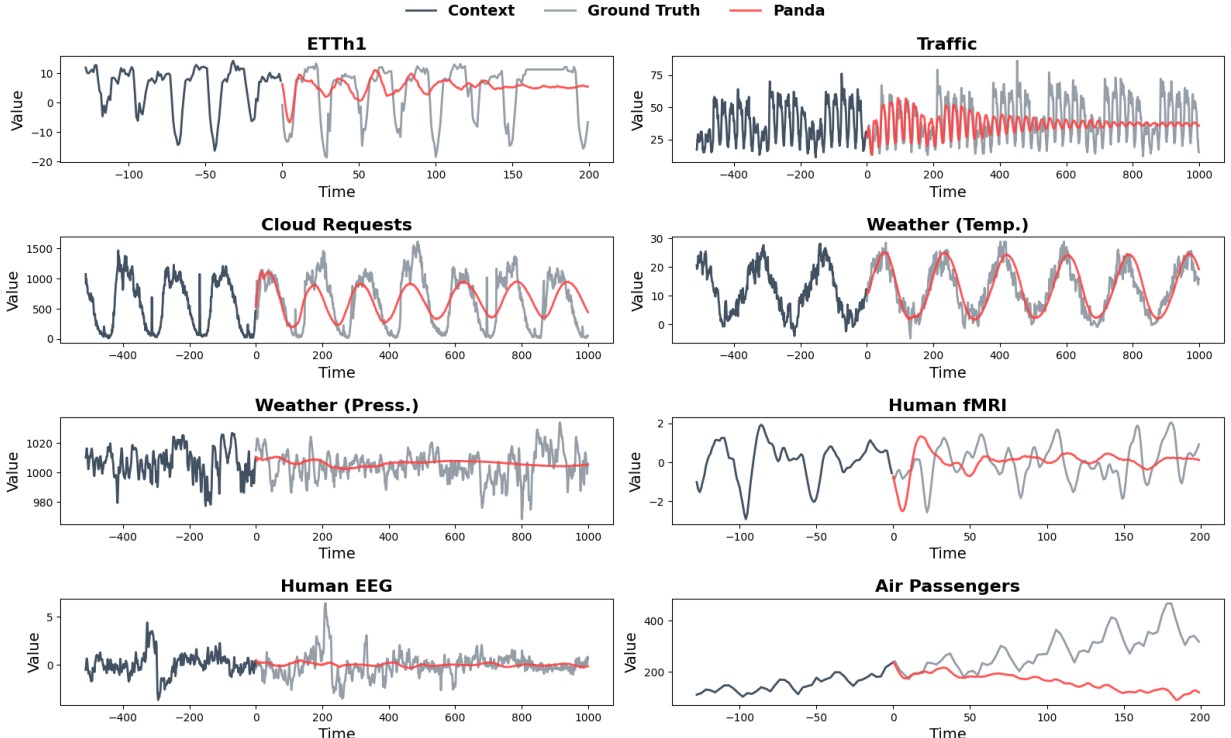

*Figure 27.* Zero-shot TS forecasts using Panda FM.

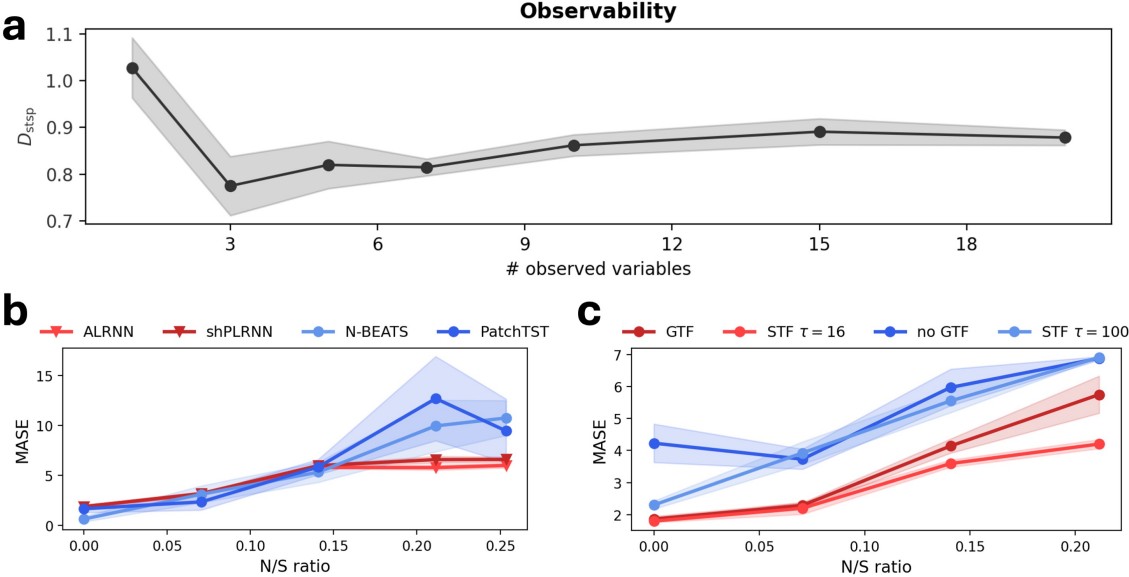

*Figure 28.* Impact of observability, noise, and DSR-specific training methods on short-term prediction. **a**) The number of observed variables has only little influence on reconstruction quality (as measured by $D_{\text{stsp}}$), as illustrated here for an ALRNN (Brenner et al., 2024a) trained by STF (Mikhaeil et al., 2022) on the $20d$ Lorenz-96 system. **b**) Short-term prediction error (MASE on 64 step forecasts) increases with noise level (N/S ratio), but does so more strongly for TS models (blue) than for DSR models (red), for stochastic Lorenz-63. **c**) Optimal DSR training improves short-term prediction (tested on stochastic Lorenz-63): Training a shPLRNN by GTF (Hess et al., 2023) or STF (Mikhaeil et al., 2022) with dynamically optimal forcing interval ($\tau = 16$) yields lower forecast errors than with suboptimal forcing ($\tau = 100$) or without GTF; see Mikhaeil et al. (2022) and Hess et al. (2023) who observed the same for DSR performance. Error bars = MAD.

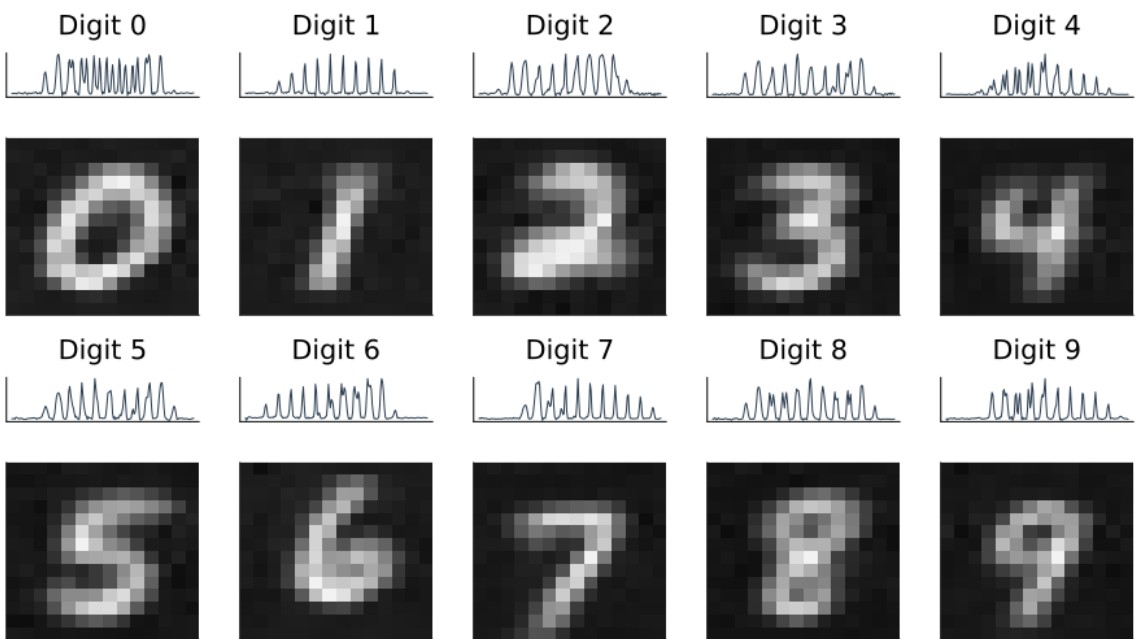

*Figure 29.* Predictions from a shallow PLRNN (Hess et al., 2023) trained via sparse teacher forcing (Mikhaeil et al., 2022) on sequential MNIST with the digit label provided as an external input. For each digit, a forecast was produced (top panels) and then reshaped to reconstruct the corresponding image (bottom panels).

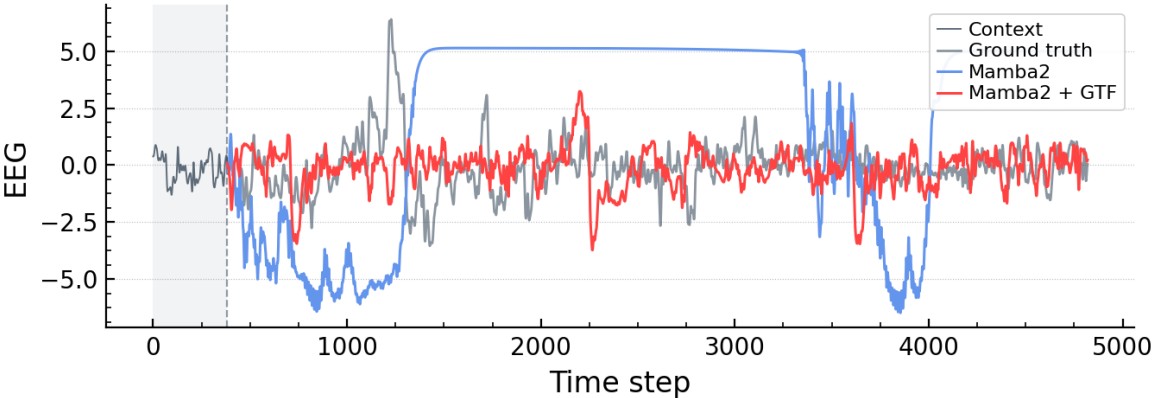

*Figure 30.* Proof-of-concept example that other types of architecture than typically used in the DSR literature could also profit from DSR training techniques, in this case a Mamba-2 block (Dao & Gu, 2024) followed by a 1-hidden-layer MLP trained with (red) or without (blue) GTF (Hess et al., 2023) on human EEG data (gray). Both long-term performance measures ($D_{\mathrm{stsp}}$ with vs. w/o GTF: 4.86 vs. 11.45, $D_{\mathrm{H}}$: 0.13 vs. 0.42) as well as the short-term prediction error (MASE(20): 10.1 vs. 14.2) are improved by GTF.

*Table 3.* Acronyms used throughout the main text.

| Acronym | Definition |
|---|---|
| AI | Artificial Intelligence |
| AL-RNN | Almost-Linear Recurrent Neural Network |
| AR | Auto-Regressive |
| ARMA | Auto-Regressive Moving-Average |
| AutoARIMA | Automatic Auto-Regressive Integrated Moving Average |
| DS | Dynamical System(s) |
| DSR | Dynamical Systems Reconstruction |
| EEG | Electroencephalogram |
| ETTh1 | Electricity Transformer Temperature (hourly, dataset 1) |
| FM | Foundation Model |
| fMRI | functional Magnetic Resonance Imaging |
| GPU | Graphics Processing Unit |
| GTF | Generalized Teacher Forcing |
| LLM | Large Language Model |
| MA | Moving Average |
| MAD | Median Absolute Deviation |
| MAPE | Mean Absolute Percentage Error |
| MASE | Mean Absolute Scaled Error |
| ML | Machine Learning |
| MLP | Multi-Layer Perceptron |
| MSE | Mean Squared Error |
| NBEATS | Neural Basis Expansion Analysis for Time Series |
| NLP | Natural Language Processing |
| ODE | Ordinary Differential Equation |
| OOD | Out-of-Domain |
| PatchTST | Patch Time Series Transformer |
| RC | Reservoir Computer / Reservoir Computing |
| RNN | Recurrent Neural Network |
| shPLRNN | Shallow Piecewise-Linear Recurrent Neural Network |
| SINDy | Sparse Identification of Nonlinear Dynamics |
| SOTA | State-of-the-Art |
| STF | Sparse Teacher Forcing |
| TF | Teacher Forcing |
| TS | Time Series |
| TSA | Time Series Analysis |
| TSF | Time Series Forecasting |
| xLSTM | Extended Long Short-Term Memory |

