# OpenReview forum: "Position: A Dynamical Systems Perspective is Needed to Advance Time Series Modeling"
_ICML.cc/2026/Position_Paper_Track — ICML 2026 Position Paper Track regular_

### Official Review · Reviewer_fZ2H · 2026-02-23

**Significance:** 4
**Argument Clarity:** 4
**Rating:** 5
**Confidence:** 4

**Questions:**

Related to my points above :

- How does stochasticity interact with chaotic behavior ? For instance I see many papers working on stochastic systems by incorporating noise to a deterministic but chaotic system (typically, Lorenz-63). Does this make sense from a modeling point of view ? Doesn’t adding noise this way kill chaos by smoothing the attractor into a true 3D attractor (not fractal anymore)?
- How important, according to the authors, is it for time series prediction systems trained on real data to provide a notion of confidence and uncertainty in their predictions, and how to evaluate it ?
- This is more of a curiosity, but how much of the basic theory of dynamical systems can be extended to time series with constrained values, e.g. on the sphere for trajectories on a sphere (common for instance in oceanographic/atmospheric data) or other types of constraints (for instance EEG data is commonly represented as time series of covariance matrices) ?

As a minor comment, I would suggest to adjust the title to "Positon: A Dynamical Systems Perspective is Needed
to Advance Time Series Modeling" (removing "why") to make it more direct, but this may simply be a matter of taste.

**Alternative Views Section:**

Yes

**Compliance With Llm Reviewing Policy A Conservative:**

Affirmed.

**Discussion Potential:**

3

**Final Justification:**

The rebuttal provided interesting discussion points, that I hope will be incorporated somehow in the final paper. I stand by my positive evaluation (5) of the paper.

**Paper Summary:**

This paper defends the idea that time series modeling using machine/deep learning should systematically use ideas and concepts from the theory of dynamical systems in order to progress and produce sensible models.

The authors propose a self contained primer on dynamical systems theory, illustrated using several low dimensional examples and trained neural surrogate models to explain concepts such as equilibrium points and their type, attractors, bifurcations, chaos, etc. Then they present a number of desirable properties that a surrogate model for time series should possess. They emphasise that the effor should not only put on their short term prediction capabilities, nor on their long term prediction capability, which is not even meaningful for chaotic systems, but rather on more longer term properties such as geometric and statistical consistency of the model. They review and suggest quantitative metrics to make this concrete, insisting onf their computation, applicability and limitations. Then they classify and review a number of architectures for time series modeling, drawing a line between dynamical system inspired models (which somehow learn the flow of a dynamical system) and others (such as e.g. transformer based models). Ad hoc and foundation models are discussed in both cases. Then they also note that training criteria is an important feature for the models, regardless of architectural choices. They also point out the interpretability provided by dynamical systems compared to other approaches.

The next part is devoted to research perspectives made possible by seeing time series modeling under the lens of dynamical systems, putting forward the universality of this type of representation, the inherent limiations of time series forecasting due to chaotic phenomena, which, according to them, should be an incentive to focus on long term properties and consistency/coherence of the forecasts rather than pure predictive performance. The authors highlight the potential to model and capture regime shifts and bifurcations (and give examples of areas where this is of interest). These concepts are illustrated by experiments on several standard datasets of the literature, showing how dynamical system-inspired models logically outperform other types of models on long term consistency (non dynamical models often fail to capture long term behavior and often degenerate into limit cycles or spurious equilibrium points), and only perform marginally worse on short term prediction, often without having been trained explicitly for this task.

The authors then present alternative view to their position : 1) the question of whether precise dynamical system models are necessary for partially observed systems, where conditioning by the right auxiliary variable goes a long way to obtain good performance, while a pure dynamical system will necessarily be limited since it doesn’t have access to all the variables of the problem. 2) the difficulty of high dimensional analysis which is much harder than in small dimensions where insights from a dynamical model are straightforward to obtain. 3) The fact that combining dynamical systems with competing approaches (ergodic theory, LLM based modeling) is possible.

The conclusion is organized in a call for several concrete actions to advance the field, including sticking to good practices for training and evaluation of dynamical models, and outline perspectives to address critical problems such as tipping point modeling using dynamical systems theory, or to use models as a way to mechanically describe and understand time series.

Further details on each part are provided in several appendices.

**Position:**

Yes

**Position In Title:**

Yes

**Related Work:**

3

**Strengths And Weaknesses:**

Strengths :

- Excellent self-contained introduction to key concepts of dynamical systems theory, with clear exposition, nice illustrations on low dimensional simulated systems or trained models, with an extensive body of relevant references. The main text is sufficient to understand most of the concepts and follow the discussion, the appendices provide more details and background and are well organized.

- A clear separation is drawn between models based on dynamical systems and more "ad hoc" models such as transformer based ones. The latter may seem appealing due to their superiority in NLP, but the authors question this view under the lens of the lack of explicit temporal representation capabilities (though they can still be powerful for short term forecasting as the authors point out).

- Informative experimental comparisons of several types of models (dynamical/non dynamical, foundation/non foundation) on a number of classical models from the literature, providing insight on the advantages and limitations of each type of methods, shedding light on the position that is put forward. Quantitative and qualitative behaviors of the models are systematically discussed.

- Links between technical contents and actually relevant problems of the literature in several application domains that should be adressed in the future: for instance tipping points, climate vs weather forecasting, etc.


Weaknesses :

- The paper does not discuss stochastic dynamical systems (for instance given by SDEs) and where they stand in terms of usefulness or capacity to answer the challenges in time series modeling. This is however an active area of research, e.g. in the weather forecasting or financial data analysis literature. For instance the question of uncertainty in the predictions could also have been addressed (see below). This criticism is mitigated by the fact that the authors do mention briefly stochastic and non autonomous systems in the Alternative views section, as well as concepts from ergodic theory which arguably provide a link between chaos and randomness.

- Among other topics (that I consider less important than the one above) that may have been addressed, we can mention specificities of spatio-temporal data (e.g. PDE in geosciences, or video data in vision), and also handling irregularly sampled data (in time, possibly also in space). But again this would require a lot more exposition than what is already in the paper.

**Support:**

4

---

> ### Author Rebuttal · Authors · 2026-03-30
>
> We thank the referee for the careful reading and strong support of our work, and for raising several interesting points!
>
> Please find add. rebuttal material here: https://drive.google.com/file/d/1nnVXjgacnS-oXY-PRnPZ3zyiw_aMZjWR/preview
>
> **Weaknesses**
>
> **W1 (discussion on stochastic DS):** Thanks for this suggestion; we will expand the discussion on stochastic DSR models that explicitly account for process/dynamical noise (Tzen & Raginsky 2019) in our revision, and introduce them right from the start on in sects. 2 & 3 (rather than only in passing in sect. 5). Essentially all SOTA DSR models can deal well with noise in both the process and the observations, as all the noisy empirical TS used in creating Fig. 5 highlight (and as further illustrated in new Fig. R4b). However, only DSR models that explicitly include a stochastic formulation in the dynamical equations yield an explicit uncertainty estimate for the latent states, which is of practical relevance in some applications. We will stress this point in the revised manuscript, where we will dive in a bit more detail into both Neural SDE-based dynamical models (e.g. Kim et al. 2021 *NeurIPS*; Haussmann et al. 2021 *AISTATS*; O'Leary et al. 2022 *J Comp Physics*) and discrete time DS with noise terms (e.g. Kramer et al., 2022 *ICML*; Pals et al. 2024 *NeurIPS*; Vermani et al. 2025 *ICLR*), and will discuss the relevance of obtaining state (or parameter) uncertainty estimates in particular scenarios like weather forecasts or the prediction of clinical patient trajectories, where the likelihood of extreme events may be more important than the (conditional) mean.
>
> **W2 (spatio-temporal systems and irregularly sampled TS):** Yes, we completely agree, both models for spatio-temporal data and irregularly sampled TS are important topics. We had deliberately omitted them for space reasons and to keep the discussion focused, but it is probably a good idea to at least mention these settings and provide some pointers to relevant literature (e.g. Rubanova et al. 2019 *NeurIPS*, Li et al. 2021 *ICLR*).
>
> **Questions**
>
> **Q1 (introducing noise into chaotic DS):** This is an excellent question (with no simple or straightforward answer)! Introducing noise into deterministic DS just for testing a DSR approach makes complete sense in our minds (see also Fig. 4 or our newly produced Fig. R4, for instance). However, the referee is right that noise can alter the dynamics beyond just perturbing the system’s state in an otherwise deterministic framework. This can go either way – there are examples where noise amplifies or even induces chaotic dynamics (e.g. Crutchfield et al. 1982, *Physics Reports*; Lin & Young 2008, *Nonlinearity*), but also those where, on the contrary, noise *stabilizes* or synchronizes a chaotic attractor (e.g. Matsumoto & Tsuda 1983, *J Stat Physics*; Pikovsky & Kurths 1997, *Phys Rev Lett*; textbook by Arnold 1998). This is in itself a broader topic, however – we will remark on this in sect. 5 in the revision, pointing to some of the relevant lit., but will otherwise keep this brief.
>
> **Q2 (uncertainty estimates):** Please see our reply to W1 above. In some scenarios, where decisions need to be made based on the predicted behavior of an observed system, obtaining uncertainty estimates will definitely be important (e.g. Fechtelpeter et al. 2026, *npj digital med*). For instance when treating a patient, medical decisions may want to put the emphasis on avoiding the worst case outcomes, rather than focusing on the average trajectory. For such scenarios, DSR models which infer an explicit process noise term and thus yield uncertainty estimates for the latent states or model parameters (see cit. above) may be preferred. Again, this is worth mentioning in the context of expanding the discussion of stochastic systems in sects. 2 & 3.
>
> **Q3 (DS with constrained values):** In general the answer likely depends on the type of constraints. For instance, if the dynamics evolves on a certain geometry like a sphere or torus, then DS variables can often be equivalently rewritten in the corresponding coordinate system (e.g. polar or phase coordinates) and no restrictions apply (Perko 2001). In other cases, saturating terms (like a softmax) may enforce restrictions (e.g. in systems where values are bounded by the physical properties of the DS). In general, we are not aware of any particular limitations in applying DS theory that arise from such constraints, but there is also an entire branch of DS theory which explicitly deals with DS with hard thresholds as, e.g., in billard (Filippov theory and impact systems, e.g. textbooks by Filippov 1998, di Bernardo et al. 2008). Whether one describes the time series itself, or some derived quantity like covariance matrices, by dynamical rules, in general depends more on the specific research question one would like to answer and is not limited by DS theory, we would say.
>
> **Minor:** Agreed, we will change the title as suggested.

---

> > ### Author Rebuttal · Reviewer_fZ2H · 2026-04-02
> >
> > Thanks for the rebuttal and for answering my questions.
> > I understand space is limited, but I believe giving a bit more weight to stochastic dynamics and uncertainties will provide more breadth to the paper.

---

### Official Review · Reviewer_FpyB · 2026-03-12

**Significance:** 3
**Argument Clarity:** 3
**Rating:** 4
**Confidence:** 3

**Questions:**

1. The paper suggests that DSR-based training techniques may simplify models and reduce computational costs even for short-term forecasting tasks. What are the limitations of this approach? Can it be applied broadly across different types of time-series models, or is it mainly applicable to specific model classes?
2. For high-dimensional systems with strong stochastic components, the authors suggest reconstructing a high-dimensional latent dynamical system. However, one of the main challenges in applying dynamical systems theory is the potential curse of dimensionality. How might the proposed approach address this issue in practice, particularly in scenarios with low signal-to-noise ratios?
3. Traditional statistical metrics such as MSE or RMSE often provide strong performance for short-term forecasting tasks. While the introduction of dynamical-system-based metrics may improve long-term stability or structural consistency, it would be helpful to clarify whether these metrics also provide advantages for short-term prediction performance.
4. As discussed above, it is not always clear whether every time series can be meaningfully associated with an underlying dynamical system. Are there practical criteria or diagnostic tools that could help determine whether a given time series is suitable for dynamical-system-based modeling? If some time series problems lack such structure, how would the proposed framework compare with existing statistical approaches?

**Alternative Views Section:**

Yes

**Compliance With Llm Reviewing Policy A Conservative:**

Affirmed.

**Discussion Potential:**

3

**Paper Summary:**

The authors argue that current time series modeling approaches face inherent limitations when dealing with some of the most challenging generalization problems. In particular, purely statistical fitting of observed data may fail to maintain long-term predictive accuracy.

The paper suggests that incorporating dynamical systems reconstruction (DSR) theory could help capture the underlying dynamical structure of data-generating processes by recovering chaotic information from observed time series.

Based on this perspective, the authors challenge the so-called Alternative View that dynamical systems (DS) theory has limited practical relevance for time series forecasting (TSF). Instead, they argue that a dynamical-systems perspective can provide both theoretical insights and practical guidance for improving time series modeling.

**Position:**

Yes

**Position In Title:**

Yes

**Related Work:**

3

**Strengths And Weaknesses:**

Strengths

1. The manuscript presents a technically informed discussion and articulates its arguments clearly. The theoretical background is well developed, and the paper raises several thought-provoking perspectives on an important topic.
2. The idea of using DSR-based measures to evaluate model behavior across systems with different levels of dynamical complexity is particularly interesting and may provide useful directions for evaluating time-series models beyond traditional statistical metrics.
3. The paper emphasizes the importance of mathematical foundations in time-series modeling and provides a relatively comprehensive overview of the theoretical tools required when applying dynamical-systems perspectives to time-series models.


Weaknesses

1. Limited discussion of practical feasibility.
While the paper argues that dynamical systems perspectives may offer a more principled framework for time-series modeling, the manuscript provides relatively limited discussion on how these ideas can be concretely incorporated into the training or design of practical time-series models.
Additional discussion or examples illustrating how DS-based insights could be integrated into existing modeling pipelines would help clarify the practical impact of the proposed perspective.

2. Implicit assumption about dynamical-system structure.
Some arguments in the manuscript appear to rely on an implicit assumption that most time series originate from an underlying dynamical system. However, this assumption may not always hold in practice.
For example, many real-world time series may represent only partial observations of a complex system, or may be heavily influenced by stochastic processes and measurement noise. In such cases, it is less clear whether dynamical-system reconstruction would still provide the advantages suggested in the paper.
Clarifying the scope of applicability of the proposed framework, particularly for systems with weak or unclear dynamical structure, would strengthen the overall argument.

**Support:**

3

---

> ### Author Rebuttal · Authors · 2026-03-30
>
> We thank the referee very much for the supportive and constructive feedback!
>
> Please find add. rebuttal material here: https://drive.google.com/file/d/1nnVXjgacnS-oXY-PRnPZ3zyiw_aMZjWR/preview
>
> **Weaknesses**
>
> **W1 (practical feasibility):** We had provided specific suggestions in sect. 6, pt. 1-5, but agree these could be spelled out in more detail and will do so by 1) exemplifying how DSR training methods like STF/GTF can easily be implemented with any RNN/SSM-based TSM (*new Fig. R6 shows a proof-of-concept for Mamba*), 2) providing details on what a DS training corpus specifically would look like that addresses pt. 2 & 4 in sect. 6 (*we now created examples for this in new Fig. R2*), 3) illustrating how an integration of RNNs/SSMs into Transformers could work, based on recent FMs for phys. TS which do this (eg Ryoo et al. 2025 *NeurIPS*), 4) specifying some of the domain-agnostic assumptions or inductive priors, like separation-of-timescales (Schmidt et al. 2021) or skew-product systems (Kloeden et al. 2020), that could be used to equip any TSM with means to predict across tipping points (*see new Fig. R3*).
>
> **W2 (DS assumption):** Indeed, almost *all* real-world DS will only be partially observed and noisy, but this per se does not limit the scope of DSR methods as pointed out in the 2nd-to-last pg. of sect. 5. The "classical" answer to partial obs. are the delay embedding theorems (Sauer et al. 1991) which ensure that attractors, in principle, can be retrieved from just scalar measurements. DSR models often perform implicit “delay embeddings” of observed TS in their latent spaces (eg https://arxiv.org/html/2508.21522v1). Hence, partial obs. are not per se a problem, as evident from examples in Fig. 5 which are all on *empirical* data from noisy systems with only a tiny fraction of all system variables observed, *and as more explicitly shown now in new Fig. R4a*. Many DSR models also explicitly account for both process & measurement noise (see cit. in sect. 5), and as shown in Fig.5 and in the lit. (Hess et al. 2023; Hemmer et al. 2025) DSR methods may still bring an advantage here. In *new Fig. R4b we further show DSR methods may even outperform TSMs in short-term predic. as noise levels increase*. By focusing too much on low-d noise-free examples in sects. 2 & 3, we may have inadvertently given a wrong impression here.
>
> **Questions**
>
> **Q1 (limits of DSR training):** Please see W1: DSR-specific training techn. can basically be applied to any TSM that has a recursive form, and thus potentially also to recent TS FMs which integrate RNNs/SSMs into Transformers (eg Ryoo et al. 2025 *NeurIPS*). Even pure Trsf. may potentially be optimized for long-term properties, encouraged e.g. via specific loss terms as in Platt et al. (2023) or Schiff et al. (2024). Other suggestions, like expanding the pretraining corpus of TS FMs by DS data (see new Fig. R2), could likely be implemented with any architecture.
>
> **Q2 (applic. to high-d noisy systems):** Please see W2: The curse of dim. and noisiness of real-world data is, we think, not per se a *principle* problem with *training* DSR models, maybe even less so than for some TSMs as new Fig. R4b suggests. The difficulties occur more at the level of later *model analysis*, which is often an explicit goal in DSR. However, there has been substantial progress in recent years also with analysis algos that efficiently scale up to high-d (Eisenmann et al. 2023, 2026).
>
> **Q3 (MSE vs. DSR loss in short-term prediction):** First, note that an MSE loss is almost always also included in training DSR models, only that for DSR either further loss terms emphasizing long-term properties are added or training is guided in a special way by techniques like STF or GTF. *Short-term pred.* (using MASE) were compared for DSR vs TS models in Fig. 5b, Tab. 1 & 2, suggesting the model classes are about on par here (although short-term pred. is not an explicit goal in DSR, and TSMs have *orders of magn.* more params., cf. Tab.1&2). In new Fig. R4c we now also prepared *an explicit example showing that DSR training may even improve short-term pred*.
>
> **Q4 (practical crit. for suitability of DSR):** We would claim that almost any TS from a natural or engineered system derives from some underlying DS. What are the alternatives? These could be either *purely* stochastic proc. which are genuinely unpredictable by *any* TSM beyond the (time-independent) moments of the distribution. Or it could be data with an artificial construction rule, like seq-MNIST, which are not truly TS but just brought into this form. We now tested DSR on seq-MNIST, and, surprisingly, it even works here, see new Fig. R5. There are indeed also stat. tests, e.g. based on phase-randomization bootstraps (Kantz & Schreiber 2004), that probe how much deterministic & nonlin. structure there is in obs. TS, which we will add to sect. 3. Pragmatically, one would probably just try out several DSR and TS models and see which works best.

---

> > ### Author Rebuttal · Reviewer_FpyB · 2026-04-03
> >
> > Thank you for your response. The questions I raised have all been addressed.

---

### Official Review · Reviewer_QcSo · 2026-03-12

**Significance:** 3
**Argument Clarity:** 2
**Rating:** 4
**Confidence:** 3

**Questions:**

See Weakness

**Alternative Views Section:**

Yes

**Compliance With Llm Reviewing Policy A Conservative:**

Affirmed.

**Discussion Potential:**

3

**Paper Summary:**

This paper proposes the position that time series modeling should reintroduce and strengthen the dynamical systems perspective. The authors argue that most time series originate from underlying dynamical systems; therefore, modeling should focus on the system dynamics rather than merely predicting values. In many cases, modern deep learning models lack explicit modeling of dynamical structures. Incorporating dynamical systems theory can potentially improve pattern interpretability and enhance generalization capability.

**Position:**

Yes

**Position In Title:**

Yes

**Related Work:**

3

**Strengths And Weaknesses:**

S: The paper clearly identifies an important problem, pointing out that current time series research relies heavily on black-box models and often lacks an understanding of the underlying generative mechanisms.

W1: The argument of the paper largely remains at a conceptual level. It would be more convincing if the authors provided deeper technical reasoning to address the following question: Why would adopting a dynamical systems (DS) perspective necessarily improve time series (TS) modeling?

W2: Although the paper advocates introducing a dynamical systems perspective, it does not outline concrete future research directions. Further discussion on potential research avenues would make the paper more complete.

**Support:**

2

---

> ### Author Rebuttal · Authors · 2026-03-30
>
> We thank the referee for the supportive and constructive feedback!
>
> Please find additional material prepared for the rebuttal here: https://drive.google.com/file/d/1nnVXjgacnS-oXY-PRnPZ3zyiw_aMZjWR/preview
>
> **Weaknesses**
>
> **W1 (why would DS concepts improve TS modeling):** TS modeling focuses on *prediction*, while dynamical systems reconstruction (DSR) models focus on *approximating the underlying governing equations* (or flow operator). If these dynamical rules underlying the observed system’s behavior were truly known, then (similarly as with a good scientific theory) …
>
> 1) using these would enable to forecast the system’s future temporal evolution (esp. long-term) naturally better than would any other way of modeling the system’s temporal structure less directly (as empirically confirmed in Figs. 5, 9, 10 & new Figs. R1,R4, and formally established in e.g. Kaszás & Haller 2020, *Chaos*, or as entailed by the def. of proper scoring rules, sect. 2.1 in Gneiting & Raftery 2007, *J Am Stat Assoc*);
>
> 2) it allows us to infer properties of the underlying system not directly covered by the observed data, like their long-term statistics (as shown in Figs. 5, 9, 10 & new Fig. R1), the geometry of attractors the system ultimately would converge to (Fig. 10 & new Fig. R1), other geom. properties of the system’s state space like location of fixed points (new Fig. R1a), their stable & unstable manifolds (Eisenmann et al. 2026 *ICLR*), or the system’s vector field more generally (new Fig. R1b). Knowledge of these properties, in turn, enables predicting the behavior of the system under conditions not directly observed, e.g. for novel initial conditions (Fig. R1a);
>
> 3) it further enables predicting the behavior of the system in completely novel scenarios & dynamical regimes, e.g. the type of dynamics beyond a tipping point (Fig. 2b); in fact, theoretically, if the true flow operator were exactly known, the system’s behavior could be predicted *anywhere* across its state and parameter space (Goering et al. 2024);
>
> 4) it enables to analyze the system in depth, understand it causally and mechanistically (Eisenmann et al. 2023 *NeurIPS*, 2026 *ICLR*), and explore new empirical scenarios, perturbations, and suitable control strategies by model simulation (Fechtelpeter et al. 2026).
>
> Standard TSM, in contrast, does not offer any of these advantages. Of course, in practice this will depend on the quality of the approximation of the flow operator. However, since DSR models *explicitly aim for such an approximation*, while TSMs do not, they will come with some of the advantages that true knowledge about the governing equations would have (Kaszás & Haller 2020, Gneiting & Raftery 2007). In the revision we will make sure to work out these points and their underlying reasoning very explicitly.
>
> **W2 (future research avenues):** Future concrete research steps we had tried to provide in sect. 6 – these include (see p.8 for more details): 1) As training methods like GTF designed for DSR are generic and work with almost any TSM, we recommend using them also for training TSMs; 2) incorporating TS data explicitly produced by *simulations of a variety of chaotic DS* into the training corpus of TS FMs; 3) shifting away from Transformer-based architectures back to RNN/SSM-based models, or at least include RNNs/SSMs as part of the architecture; 4) focus on hard problems like topological shifts by, for instance, a) incorporating explicitly designed DS simulations in the training data that feature transitions among dynamical regimes and b) building methods for extracting control parameters from the data jointly with the dynamics (e.g. similarly as in Brenner et al., 2025, *ICLR*, or Huh et al., 2025, *ICML*); 5) incorporate a more dynamical understanding into TSM design, e.g. through domain knowledge represented in LLMs.
>
> However, we agree that these suggestions could be spelled out in more detail. We will do this in the revision by 1) exemplifying how DSR training methods like GTF (Hess et al. 2023) can be implemented with any RNN/SSM-based TSM (**new Fig. R6 provides a proof-of-concept for Mamba**), 2) providing more details on what a training corpus specifically should look like that acknowledges points 2 & 4 in sect. 6 (**we now created examples for this in new Fig. R2**), 3) illustrating how an integration of RNNs/SSMs into Transformers could work, based on recent FMs for physiological data which do this (Ryoo et al. 2025 *NeurIPS*, Ma et al. 2026 *ICLR*), 4) specifying some of the domain-agnostic assumptions or inductive priors (like the separation-of-timescales principle (Schmidt et al. 2021 *ICLR*) or the skew-product-system conceptualization (Kloeden et al. 2020)) that could be used to equip any TS model with mechanisms to predict across tipping points (**see new Fig. R3**). We will also discuss the idea of combining TS/DSR models with LLMs/ reasoning models to infuse mechanistic-dynamical domain knowledge into model construction.

---

> > ### Author Rebuttal · Reviewer_QcSo · 2026-04-04
> >
> > Thanks for the response, my concerns are addressed.

---

### Official Review · Reviewer_iyWQ · 2026-03-13

**Significance:** 3
**Argument Clarity:** 3
**Rating:** 4
**Confidence:** 4

**Questions:**

1. It's stated that the DRS methods allows one to "to understand its behavior even outside of the immediate data regime." How is this possible? I.e., why can I trust my (approximate) flow operator outside of the data regime (without making any assumptions that, e.g., we're still within a basin of attraction)?
2. Given the recent success of SDE-based models in ML, could the authors comment more about how stochastic systems fit into their position?

**Alternative Views Section:**

Yes

**Compliance With Llm Reviewing Policy A Conservative:**

Affirmed.

**Discussion Potential:**

3

**Final Justification:**

The rebuttal has addressed my main concerns with respect to presentation. The promised changes seem reasonable in scope for revisions, and I think will improve the clarity of the paper considerably. My main reasons for recommending a 4 instead of a 5 is the lack of discussion of stochastic systems, and the size of promised revisions that will not be reviewable.

**Paper Summary:**

This paper argues that a dynamical systems perspective is necessary to further advance time series analysis/forecasting. It begins with an overview of dynamical systems theory, then asserts a few specific claims. In particular: (1) most real-world phenomena arise from a deterministic, autonomous, and chaotic DS; (2) that chaos bounds the long-term performance of time series forecasts; (3) that DS-based models will be more reliable at learning long-term statistics than non-DS models; (4) that DSR models may result in comparable short-term performance with many less parameters; and (5) that DS-based models may help in predicting tipping points.

**Position:**

Yes

**Position In Title:**

Yes

**Related Work:**

4

**Strengths And Weaknesses:**

### Strengths
- The paper is very exhaustive, spanning 43 pages with several illustrative experiments and technical appendices.
- The presentation is well-done; I found the figures, in particular, to be well-thought out and helpful.
- The experiments, in particular, show strong evidence for the paper's claims. This includes comparisons to "non-DS" time series models, including state-of-the-art time series foundation models. The comparison on geometric misalignment paints a very nice picture that DSR-based models can be both more efficient and more accurate at preserving the properties of a DS.

### Weaknesses
- The submission is very dense for a position paper.
	1. The paper begins with four pages of textbook-style review of dynamical systems. This is fairly dense, mentioning topics like "topological conjugacy." There are also dozens of pages of technical appendices, and many acronyms.
	2. This sometimes comes at the cost of clarity in the main manuscript. For example, a motivating, informal definition of $D_{stsp}$ is important for understanding Figure 5a, but the reader is instead pointed to a technical definition in Appendix B.
	3. I think these contents were well-written, but I fear this lack of accessibility harms the potential for useful discussions to come out of the paper. For example, a prime audience would seemingly be someone who has been building foundation models for time series forecasting, with limited formal knowledge of dynamical systems theory. I think this paper would be very intimidating to that person.
	4. It would greatly benefit the paper, in my opinion, to substantially reduce the complexity of the material presented to be more approachable, even if this limits technical depth.
- While the exhaustiveness of the paper is in some ways nice, many aspects of this submission read more as a review than "position paper."
- The alternative views section generally seems good, though leaves out an obvious alternative viewpoint: "I only care about short-term rollouts, so DS is useless to me." This is partially discussed/address throughout the rest of the text, but would also be useful as an alternative view.
- While a trivial fix, the paper does not current state its position in the title (I think removing the word "why" would be sufficient here).
### Editorial Remarks
- "Geometrical" is used throughout, but seems non-standard to me (compared to "geometric").
- \[Line 182, right column\] The authors write "control-theoretic training methods like sparse (STF)"; I think they mean "sparse teacher forcing (STF)".

**Support:**

3

---

> ### Author Rebuttal · Authors · 2026-03-30
>
> We thank the referee very much for the constructive comments, providing a helpful "general reader’s" perspective!
>
> Please find add. rebuttal material here: https://drive.google.com/file/d/1nnVXjgacnS-oXY-PRnPZ3zyiw_aMZjWR/preview
>
> **Weaknesses**
>
> **W1 (technical presentation):** Of course our goal here is to stimulate a useful discussion, not to repel potential readers by too much technical detail! It was important to us, however, to clarify some techn. points; first, because DS terminology is increasingly used in the TSM lit., but often in a ‘hand-waving’, sometimes misleading, way that may obscure rather than clarify important diff. between model classes and capabilities; second, because we think some of the methodolog. promises of DSR can only be fully appreciated with a more techn. understanding.
>
> This being said, we suggest moving sect. 2 and the more math. aspects in sect. 3 to the Appx., and replace it instead with a shorter ‘lay-style’ summary (pt. 1). We will use this space to expand on points brought up by this and other referees, like providing a more intuitive explanation of $D_{stsp}$ (pt. 2). Pt. 3 & 4 are well taken, and we will skim through the whole text and defer more technical & math. aspects to the Appx. (but would like to keep them there for the reasons above). We will also provide a list of acronyms and always spell out some of the less frequent ones (eg PL or FM).
>
> **W2 (many aspects review-like):** For sect. 2 and some of sect. 3 this may be true, but we believe one can only properly develop this position if all readers are "brought to the same page"; some background in DST/DSR is necessary to fully grasp its impact on TSM. But see our suggestions on sect. 2 & 3 above - and other than these two, we believe most of the main paper (sect. 4-6) is indeed devoted to developing this DS perspective on TSM and its potential benefits, or would the ref. disagree?
>
> **W3 (interest in short-term prediction):** Agreed! We will include this as a defendable alternative view in sect. 5; but see Figs. 5b & new R4b which suggest DSR benefits also for short-term pred.
>
> **W4 (title):** Thanks for pointing out, will be changed.
>
> **Editorial remarks:** Thanks for spotting, we will amend this.
>
> **Questions**
>
> **Q1 (inferences outside immediate data regime):** This is an important question, the answer to which may have been too implicit in the current presentation. There are different forms of generalization and analysis opportunities in DSR models that go beyond what standard TSM can offer, rooted in the fact that DSR models explicitly aim to approximate the underlying flow operator while TSMs do not.
>
> First, all current SOTA DSR models can generalize to new initial conditions within a given basin of attraction and reconstruct the full attractor geometry from just a short trajectory snippet, see *new Fig. R1a*. They also often infer add. geometrical properties of the state spaces like fixed points (Fig. R1a) or their un-/stable manifolds (Eisenmann et al. 2026 ICLR), even though not explicitly represented in the training data, and reproduce properties of the vector field surrounding the training regime (*new Fig. R1b*). These are capabilities that standard TSMs lack, and that require some sort of inference beyond the immediate data regime. Second, many recent DSR models for *beyond tipping point dynamics* (e.g. Patel & Ott 2023) or DSR FMs go further and predict behavior even in completely novel dynamical regimes (what we called a "topological shift"). How DSR FMs manage this is not well understood, but custom DSR models explicitly designed to capture *non-autonomous dynamics* often achieve this through certain inductive biases (see new Fig. R3) or implicitly exploiting behavior of trajectories near basin boundaries or bifurcations (Patel & Ott 2023).
>
> Beyond these empirical indications, whether one *really* has approximated well enough the true system’s flow operator, is something which one cannot establish with certainty without explicitly testing new predictions of the model – but this is true for *any* model, even human-created scientific theories! DSR models enable, however, to probe new conditions and perturbations by *simulation*, hence providing *empirically testable predictions*, something that standard TSMs don’t easily allow for. We will make these points more explicit in sect. 3, incl. Figs. R1,R3.
>
> **Q2 (SDE-based models):** DSR models with stochastic terms, whether in continuous time (i.e., Neural SDEs) or discrete time (e.g. Pals et al. 2024), were briefly touched upon in the 2nd/4th pg. of sect. 5. Empirically observed systems will, of course, almost always be noisy/stochastic, and all DSR methods we discuss have extensively been tested on stochastic systems (see Fig. 5, which exclusively contains noisy, partly highly stochastic, real world data, plus new Fig. R4b). We agree this was not clear enough and will make it more explicit by adding a pg. on stochastic DSR models to sect. 3 on p.4.

---

> > ### Author Rebuttal · Reviewer_iyWQ · 2026-04-02
> >
> > I thank the authors for their rebuttal. I leave a few brief remarks below, but consider my concerns resolved, and have updated my score accordingly.
> >
> > > This being said, we suggest moving sect. 2 and the more math. aspects in sect. 3 to the Appx., and replace it instead with a shorter ‘lay-style’ summary (pt. 1). We will use this space to expand on points brought up by this and other referees, like providing a more intuitive explanation of D_{stsp} (pt. 2).
> >
> > That sounds great! Thanks for the receptiveness w.r.t. presentation.
> >
> > > and other than these two, we believe most of the main paper (sect. 4-6) is indeed devoted to developing this DS perspective on TSM and its potential benefits, or would the ref. disagree?
> >
> > I agree with the authors that Sections 4-6 substantially advance the position (though, I note that with the lengthy Sections 2 & 3, these sections are <1/2 the main body). Given the suggested revisions to Sections 2 and 3, I would consider this point resolved.

---

### Decision · Program_Chairs · 2026-04-30

**Decision:**

Accept (regular)

**Comment:**

This paper argues that a dynamical systems perspective is needed to advance time series modeling, particularly beyond short-term forecasting. The reviewers are overall positive and agree that the topic is timely and important, and that the paper presents a meaningful and well-supported position. The strongest review recommends acceptance, while the others are borderline accept.

The main concerns relate to presentation rather than substance, including the manuscript's density, its partially review-like character, and the need for clearer practical framing and broader discussion of stochastic and noisy settings. The rebuttal convincingly addresses these issues, with concrete plans to improve clarity, accessibility, and scope. All reviewers indicated that their concerns were resolved.

Overall, the paper presents a clear and non-trivial position that is likely to stimulate useful discussion in the community. The remaining weaknesses are fixable and do not undermine the core contribution.